# Remote sensing of canopy nitrogen at regional scale in Mediterranean forests using the spaceborne MERIS Terrestrial Chlorophyll Index

Yasmina Loozen[1], Karin T. Rebel[1], Derek Karssenberg[2], Martin J. Wassen[1], Jordi Sardans[3,4], Josep Peñuelas[3,4] and Steven M. De Jong[2]

[1]Copernicus Institute of sustainable development, Faculty of Geosciences, Utrecht University, Utrecht, The Netherlands
[2]Physiscal geography, Faculty of Geosciences, Utrecht University, Utrecht, The Netherlands
[3]CSIC, Global Ecology Unit CREAF-CSIC-UAB, 08913 Bellaterra, Catalonia, Spain
[4]CREAF, 08913 Cerdanyola del Vallès, Catalonia, Spain

*Correspondence to*: Yasmina Loozen (y.m.a.loozen@uu.nl)

**Abstract.** Canopy nitrogen (N) concentration and content are linked to several vegetation processes. Therefore, canopy N concentration is a state variable in global vegetation models with coupled carbon (C) and N cycles. While there is ample C data available to constrain the models, widespread N data are lacking. Remotely sensed vegetation indices have been used to detect canopy N concentration and canopy N content at the local scale in grasslands and forests. Vegetation indices could be a valuable tool to detect canopy N concentration and canopy N content at larger scale. In this paper we conducted a regional case-study analysis to investigate the relationship between the Medium Resolution Imaging Spectrometer (MERIS) Terrestrial Chlorophyll Index (MTCI) time series from ESA ENVISAT at 1 km spatial resolution and both canopy N concentration (%N) and canopy N content (N g m$^{-2}$ of ground area) from a Mediterranean forests inventory in the region of Catalonia, NE of Spain. The relationships between the datasets were studied after resampling both datasets to lower spatial resolutions (20 km, 15 km, 10 km and 5 km) and at the original spatial resolution of 1 km. The results at higher spatial resolution (1 km) yielded significant log-linear relationships between MTCI and both canopy N concentration and content, $r^2 = 0.32$ and $r^2 = 0.17$, respectively. We also investigated these relationships per plant functional type. While the relationship between MTCI and canopy N concentration was strongest for deciduous broadleaf and mixed plots ($r^2 = 0.24$ and $r^2 = 0.44$, respectively), the relationship between MTCI and canopy N content was strongest for evergreen needleleaf trees ($r^2 = 0.19$). At the species level, canopy N concentration was strongly related to MTCI for European Beech plots ($r^2 = 0.69$). These results present a new perspective on the application of MTCI time series for canopy N detection.

**Keywords**: vegetation index, MERIS, foliar nitrogen concentration, foliar nitrogen content, plant functional types, Mediterranean forest, remote sensing

## 1    Introduction

Canopy nitrogen (N) concentration is an essential state variable in regional (Ollinger and Smith, 2005) and global vegetation models including both the carbon (C) and the N cycles (such as Zaehle and Friend, 2010;Smith et al., 2014). This variable has

been linked to several vegetation traits and processes at the leaf and canopy levels. At the leaf level, leaf N concentration, which represents the leaf N status expressed as a percentage of leaf dry matter (%N, N g 100g$^{-1}$ DM), has been related to photosynthetic capacity (Evans, 1989;Reich et al., 1995;Reich et al., 1997;Reich et al., 1999;Wright et al., 2004), specific leaf area, leaf life span (Reich et al., 1999;Wright et al., 2004) and light use efficiency (Kergoat et al., 2008). Leaf N concentration expressed on

a leaf area basis, also called leaf N content (N g m$^{-2}$), has also been linked with chlorophyll content, Rubisco content (Evans, 1989) and photosynthetic capacity (Evans, 1989;Reich et al., 1995). At stand scale, canopy nitrogen concentration, which represents the leaf N concentration averaged over the stand canopy, has also been found to correlate with above ground Net Primary Productivity (NPP) (Reich, 2012), while canopy N content has been linked with the canopy light use efficiency (Green et al., 2003).

Given their links to many vegetation processes, leaf and canopy N variables could be used to constrain N cycle modules in global vegetation models. At the global scale, ample data is available to constrain models for the C cycle; however, data to constrain the N cycle are limited. Currently, canopy N data are not widely available and canopy N sampling campaigns are time-consuming and thus expensive tasks. Moreover, upscaling from local sampling campaign measurements represents an additional limitation. In this perspective, local, regional or even global remotely sensed canopy N estimates will be a valuable

addition, enabling us to collect information in a less time intensive and expensive manner than traditional on-field sampling campaigns. Such near global canopy N estimates will be beneficial as input in global vegetation models or to calibrate and validate these models.

Remote detection of foliage N status has been extensively studied at the leaf scale (Hansen and Schjoerring, 2003;Ferwerda et al., 2005;Li et al., 2014) and few studies have investigated the processes underlying the relationships between vegetation indices and foliar N (Pacheco-Labrador et al., 2014). Detection of foliage N status with vegetation indices is attributed to the strong

link between foliar nitrogen and chlorophyll content (Schlemmer et al., 2013) and is often based on the NIR and red-edge region of the spectrum, hence similar to the ones used for chlorophyll detection (Filella and Penuelas, 1994;Dash and Curran, 2004;Clevers and Gitelson, 2013). At canopy level, however, spectral reflectance is a complex function of vegetation cover, plant activity, water content, illumination angle, viewing angle and atmospheric composition (Kumar et al., 2006) and it is not

straightforward to disentangle the influence of nitrogen from other contributions in the spectra. It is thus not clear how the relationships observed at the leaf level translate at the canopy level. The mechanisms possibly modifying the remote sensing of foliage N status at the canopy scale are still not clearly understood (Ollinger, 2011). High correlation between canopy N and both NIR reflectance and albedo has been reported in boreal forests (Ollinger et al., 2008). However, the mechanism behind these findings is still controversial. Knyazikhin et al. (2013) argued that the observed correlation solely resulted from canopy

structural differences between broad and needleleaf forests and was thus spurious. Other authors, although agreeing that canopy structure was a confounding factor to account for, stated that the NIR – canopy N relationship was not necessarily spurious and stemmed from an association between canopy N and structural traits (Ollinger et al., 2013;Townsend et al., 2013). Canopy traits are interrelated (Wright et al., 2004) and have been known to covary due to evolutionary convergence, as stated by Ollinger (2011).

Different remote sensing techniques have been applied to detect canopy N in terrestrial vegetation. Imaging spectrometry has proven efficient in improving N sensing capabilities at the local scale. Imaging spectrometry images are acquired from either airborne or spaceborne sensors and are analysed with different methods, including partial least squares regression (PLS), continuum removal, spectral unmixing or vegetation indices (Smith et al., 2003;Ollinger et al., 2008;Huber et al., 2008;Martin et al., 2008;Schlerf et al., 2010;Wang et al., 2016). Among other techniques, ratios or normalized differences of reflectance

bands in the Red and Near Infrared (NIR) regions of the spectrum, the so called vegetation indices (VI) (Glenn et al., 2008), are one of the most straightforward methods for canopy N detection. Combined with in situ hyperspectral devices, vegetation indices have been extensively used for leaf or canopy N detection in agricultural systems (Peñuelas et al., 1994;Filella et al., 1995;Hansen and Schjoerring, 2003;Tian et al., 2011;Schlemmer et al., 2013;Li et al., 2014). Vegetation indices have also been applied to airborne or spaceborne acquired imagery in natural environments (Ramoelo et al., 2012;Wang et al., 2016).

A particular vegetation index, the MERIS Terrestrial Chlorophyll Index (MTCI) has been proposed for detecting canopy N (Clevers and Gitelson, 2013). MTCI was originally computed from three reflectance bands from the Medium Resolution Imaging Spectrometer (MERIS) aboard the European Space Agency (ESA) ENVISAT satellite at a spatial resolution of 1 km. However, it can also be obtained from other sensors' reflectance data and a similar product will be available from the ESA Sentinel-2 satellite mission (Drusch et al., 2012). It was first developed to estimate chlorophyll content (Dash and Curran, 2004,

2007). Regarding canopy N detection, most studies were carried out in agricultural crops using MTCI values computed from in situ hyperspectral reflectance data (Tian et al., 2011;Clevers and Gitelson, 2013;Li et al., 2014). A few were directed towards sensing N concentration in natural environments using airborne data, e.g. in temperate forests (Wang et al., 2016), or spaceborne data, for example in grasslands (Ramoelo et al., 2012;Ullah et al., 2012) or sub-tropical forests (Cho et al., 2013).

In this context, there are several knowledge gaps that we would like to address in this paper. First, although 1 km spatial

resolution spaceborne MTCI time series are available from the ESA, MTCI has mainly been employed to detect canopy N in agricultural applications with in situ devices and rarely in a broader range of natural ecosystems and scales using spaceborne data. Due to its almost global coverage, MTCI time series could be applied to estimate canopy N over a larger spatial extent Moreover, Mediterranean forests have specific functional characteristic due to their great forest ecosystems diversity, influenced by contrasting climatic and topographic conditions, and their high tree species richness (Vilà-Cabrera et al., 2018). However,

to our knowledge, limited research has been conducted to sense canopy N in Mediterranean ecosystems (Serrano et al., 2002) and even more so in Mediterranean forests. The relationship between MTCI and both N concentration ($N_{[\%]}$, %N) and canopy N content ($N_{[area]}$, g m$^{-2}$) has been studied separately (Clevers and Gitelson, 2013;Wang et al., 2016), but very few analyses (Mirik et al., 2005;Ullah et al., 2012) have compared the ability to detect canopy N concentration and canopy N content simultaneously, especially in forest ecosystems.

The objective of our study is thus to investigate the relationship between the spaceborne MTCI remote sensing product and canopy N in Mediterranean forests at the regional scale. More specifically, the relationships between MTCI and both canopy N concentration and canopy N content are investigated and compared. We then also examine these relationships per PFT and at the species level.

Remote sensing of canopy N is often limited to local scale studies due to the spatial restrictions associated with N data acquisition in the field and treatment of high spatial resolution remote sensing imagery with limited spatial coverage (Lepine et al., 2016). Our case-study exploits the broadly and readily available MTCI time series at 1 km spatial resolution from the ESA ENVISAT mission and combines it with canopy N data, both concentration and content, from 846 forest plots measured between 1988 and 2001 by the Catalonian National Forest Inventory (Gracia et al., 2004). First, we develop a methodology to overcome the time discrepancy between our two sets of data. Next, both data sets are resampled to the same, lower, spatial resolutions, i.e. 5 km, 10 k, 15 km and 20 km, in order to overcome the initial spatial discrepancy between MTCI spatial resolution (1 km) and the size of the forest plots (6 m). Subsequently, we analyse the relationship between MTCI and both canopy N concentration and canopy N content variables, both at the resampled and initial spatial resolutions. The relationships at the initial spatial resolution are then stratified according to the PFT of the plots. The results are presented and discussed. Finally, we address the implications for future research and draw a conclusion.

## 2 Material and methods

### 2.1 Study area

Our study area corresponds to the region of Catalonia (Fig. 1) which is located in north eastern Spain and has a spatial extent of 32,114 km$^2$ (Sardans et al., 2011). While the region is characterised by a Mediterranean climate, the presence of the Pyrenees to the northwest and the Mediterranean Sea to the east creates contrasting climate conditions with an altitudinal gradient from north to south and a continental gradient from west to east. Following this pattern, the mean annual temperature varies from 1 °C in the north to 17 °C in the south (Sardans et al., 2011). While mean annual precipitation (MAP) is 1400 mm in the Pyrenees, in the south, the MAP is lower than 350 mm (Sardans et al., 2011), leading to seasonal drought (Lana and Burgueño, 1998) and fires (González and Pukkala, 2007), impacting the vegetation (Liu et al., 2015).

### 2.2 Data collection

#### 2.2.1 Canopy N data

The canopy N data used in this research was collected by the Ecological and Forestry Applications Research Centre (CREAF), Universitat Autònoma de Barcelona. The data included 2300 closed canopy forest plots sampled between 1988 and 2001 by the Catalonian National Forest Inventory (Gracia et al., 2004).

The forest plots (Fig. 1) had a minimum diameter of 6 m, which varied depending on the tree density in order to include between 15 and 25 trees with a diameter at breast height (DBH) of at least 5 cm. The DBH was recorded for all the trees present on the plot with a DBH of minimum 5 cm. The plots were investigated for canopy N concentration ($N_{[\%]}$, %N) defined as g of N per 100 g of leaf dry matter. The leaf samples were collected from the upper central part of the crown using extensible loppers. All foliar cohorts present in the canopy were included in the leaf sample. Each leaf sample was constituted by the leaves of at least

three different trees of the dominant tree species in the canopy. The species dominance was determined by the tallest individual.

A proportion of 96% of the plots included in this analysis were monospecific (Sardans et al., 2011). 4% of the plots (n = 30) had two codominant species. For these plots, two leaf samples were collected, one for each of the codominant species found on the plots.

The leaf samples were dried and then ground using a Braun Mikrodismembrator-U (B. Braun Biotech International, Melsungen, Germany). They were analysed for foliar N concentration using the combustion technique coupled to gas chromatography using

a Thermo Electron Gas Chromatograph (model NA 2100, CE Instruments-Thermo Electron, Milan, Italy) (Gracia et al., 2004). To scale from leaf to canopy level, we used the leaf nitrogen concentration averaged over three individuals as the plot level value (Schlerf et al., 2010). We did not weight the average by species abundance (Smith and Martin, 2001) as only 4% of the plots had two different species.

Along with the canopy $N_{[\%]}$ data, we used foliar biomass data (dry matter g per square meter of ground area, g m$^{-2}$) acquired

during the same forest inventory (n = 2286). The foliar biomass data were obtained for each plot from allometric equations relating the diameter at breast height to the leaves dry weight. The allometric equations were species specific (Sardans et al. (2015), Table A1). The foliar biomass data were used to calculate canopy N content ($N_{[area]}$, g of N per m$^{-2}$ of ground) for each plot following Eq. (1):

$$\text{canopy } N_{[area]} = \frac{\text{canopy } N_{[\%]} * fbiom}{100}, \qquad (1)$$

where $canopy\ N_{[area]}$ is the canopy N content (N g per square meter of ground area, g m$^{-2}$), $canopy\ N_{[\%]}$ is the canopy N concentration (%N) and *fbiom* is the foliar biomass (g m$^{-2}$).

For the plots with two codominant species, the concentration measurements were done separately. The obtained foliar N concentration and biomass values were then averaged to obtain a single canopy $N_{[\%]}$ and canopy$_{[area]}$ value for each plot with two codominant species. Among these 30 plots with codominant species, 16 plots had codominant species from different PFT.

Their PFT is thus labelled as mixed while the plots with several codominant species from the same PFT are labelled according to their PFTs.

Catalonian forests include both deciduous and evergreen broadleaf as well as evergreen needleleaf tree species. These three PFTs are referred to as Deciduous Broadleaf Forest (DBF), Evergreen Broadleaf Forest (EBF) and Evergreen Needleleaf Forest (ENF), respectively. The main tree species are *Pinus halepensis* Mill., *Pinus sylvestris* L., *Quercus ilex* L., *Pinus uncinata*

Ramond ex DC., *Pinus nigra* J.F. Arnold, *Quercus suber* L., *Quercus cerrioides* Willk. & Costa., *Quercus petraea* Liebl. and *Fagus sylvatica* L. These species accounted for 92% of the sampled forest plots. The 15 tree species included in this analysis are listed in Table 1. Plots with a rare dominant tree species, i.e. species that were detected in only one single plot, were excluded from the analysis. This applied to plots with these dominant species: *Abies alba* Mill*., Fraxinus augustifolia* Vahl*, Fraxinus excelsior* L.*, Pinus radiata* D. Don*, Populus nigra* L.*, Populus tremula* L.*, Quercus robur* L.

## 2.2.2 MTCI product

The MERIS Terrestrial Chlorophyll Index (MTCI) was first developed to estimate chlorophyll content in canopies. MTCI is sensitive to high chlorophyll content while presenting low sensitivity to soil brightness (Curran and Dash, 2005). Its calculation, presented in Eq. (2), is based on three reflectance bands, located around the red edge point (REP) (Dash and Curran, 2004):

$$MTCI = \frac{R_{band10} - R_{band9}}{R_{band9} - R_{band8}} = \frac{R_{753.75} - R_{708.75}}{R_{708.75} - R_{681.25}} \qquad (2)$$

where $R_{band8}, R_{band9}$ and $R_{band10}$ represent the 8th, 9th and 10th bands of MERIS, respectively. Following MERIS standard bands settings, the centres of the bands were located at 681.25 nm, 708.75 nm and 753.75 nm on the electromagnetic spectrum. While the ESA ENVISAT satellite mission producing MERIS data came to an end in 2012, MERIS products and MTCI in particular are still relevant because the new ESA Sentinel-2 and Sentinel-3 satellite missions have improved band settings compared to those of MERIS. MTCI can be calculated from Sentinel-2 reflectance data with increased spatial resolution to 20 m (Drusch et al., 2012). The Sentinel-3 mission also releases a level 2 chlorophyll product, the OLCI Terrestrial Chlorophyll Index (OTCI), which calculation is directly based on MTCI. OTCI continues the time series already available for MTCI (Dash and Vuolo, 2010;Vuolo et al., 2012). In this study, we put emphasis on ENVISAT-MERIS as our field data are closer to the MERIS acquisition period.

MTCI level 3 imagery was obtained from the NERC Earth Observation Data Centre (NEODC, 2015) for the region of Catalonia between 2002 and 2012. The original data were provided by the European Space Agency and then processed by Airbus Defence and Space. The original MERIS reflectance images, following ENVISAT specifications, have a revisit time of three days and a spatial resolution of 300 m. Compared to the original reflectance images, the MTCI processed imagery has been corrected for atmospheric influences and cloud cover (Curran and Dash, 2005) and is available as an either weekly or monthly averaged product almost globally (Curran et al., 2007). The spatial resolution of the processed data is approximately 1 km. As there is no temporally averaged product available at full resolution, we chose to carry out this analysis with the MTCI monthly averaged processed imagery. This was done to decrease the uncertainty resulting from the use of single daily reflectance values. An MTCI time series of 10 years is available almost globally. One MTCI monthly averaged imagery product covering the entire study area was obtained for every month between June 2002 and March 2012, except for October 2003, when no valid product was available.

## 2.3 Data handling

### 2.3.1 Methodology to link canopy N data to MTCI values

There is a discrepancy between the timing of the ground truth sampling and the satellite image acquisition period. While the plot sampling campaigns were carried out between 1988 and 2001, the ENVISAT satellite mission was launched in 2002 and ended in 2012. To overcome the discrepancy, MTCI images were averaged by month over the 10 years of the satellite mission period. This process yielded twelve MTCI averaged images, one for each month. The averaged MTCI images were then linked

to the forest plots based on the forest plot coordinates and sampling month, as the exact sampling date was known for each plot. The period between the 1st of June and the 31st of October was determined to be the growing season after a pre-analysis, where we studied yearly temporal variation of MTCI in several locations and forest types in Catalonia. This extended period was chosen to encompass the different vegetation phenology types corresponding to the contrasted climate conditions in this region. The forest plots sampled outside of the growing season were excluded from the analysis. The inter-annual variation of canopy $N_{[\%]}$ data was analysed for each month included in the analysis to ensure that the ground data could be related with MTCI data (Figure A1). The Globcover 2009 land cover map was used to exclude forest plots for which the dominant vegetation type of the MTCI pixel did not correspond to natural vegetation. The Globcover map was created by ESA using MERIS reflectance data from 2009 (Bontemps et al., 2011). The Globcover map was downloaded from the ESA data user elements website (ESA, 2010). This map comprises 22 land cover classes and has a spatial resolution of 300 m. Using this map, we excluded forest plots that had undergone a land cover change since the sampling period and did not have a natural vegetation cover any more at the time of remote sensing image acquisition. To do so, the landcover map was first resampled to a spatial resolution of 1 km to be in accordance with MTCI spatial resolution. The resampling was done using the majority option, which ensured that the resampled landcover type was the most occurring landcover type in the MTCI pixel. Resampling the landcover map was done to exclude the plots located on heterogeneous MTCI pixels, i.e. pixels where the natural vegetation was not the dominant landcover type. Then, the plots located on land area classified as either rainfed cropland, mosaic between croplands and natural vegetation, sparse vegetation or artificial surfaces were excluded from the analysis.

### 2.3.2    Relationship between MTCI and canopy N data at lower spatial resolution

In a first step, the relationships between MTCI and canopy N data values were investigated after resampling both datasets to the same, lower, spatial resolution. The resampled spatial resolutions were 5 km, 10 km, 15 km, and 20 km. This was done because of the initial difference in support size between MTCI spatial resolution and the forest plots size (i.e. 1 km and 6 m, respectively). This enabled us to investigate the relationships between MTCI and canopy N data when the spatial discrepancy was accounted for. The statistical basis of this approach is that we bring both datasets (forest plots and MTCI values) to the same support size or representative area (Bierkens et al., 2000). By averaging out forest plot values within this support size, we calculate the mean of the canopy N value at that support size. By resampling the MTCI values to that same support size, the obtained result consist of a mean of the MTCI value at that support size. We then regressed the expected canopy N values (at the new support size) against the expected MTCI values (at the new support size).

The monthly averaged MTCI images obtained previously (section 2.3.1) were resampled successively to 5 km, 10 km, 15 km, and 20 km. Beforehand, the Globcover 2009 land cover map was used to exclude from the resampling computation the MTCI pixels located on land surface without natural vegetation cover. As for the forest plots, MTCI pixels whose land cover class corresponded to rainfed cropland, mosaic between croplands and natural vegetation, sparse vegetation or artificial surfaces were excluded from the upscaling analysis. Forest plots data were then averaged per month over the newly obtained pixel. The relationship between the resampled MTCI values and canopy N data was analysed using log-linear regression.

### 2.3.3 Relationship between MTCI and canopy N data at original spatial resolution (1 km)

In a second step, the relationships between MTCI and canopy N data, both canopy $N_{[\%]}$ and canopy $N_{[area]}$, were examined at the original spatial resolution of 1 km. This allowed us to investigate the influence of PFT and species on the relationships as this information was lost in the resampling process. The relationships between MTCI and canopy N at 1 km-spatial resolution were analysed with log-linear regression for the whole dataset, for each PFT separately as well as for individual species.

### 2.3.4 Statistical analysis

After applying the selection criteria as explained in the section 2.3.1, i.e. plots measured between June 1[st] and October 31[st], exclusion of plots with infrequent species and selection based on Globcover 2009, 846 forest plots were available for analysis, including 841 plots with foliar biomass and canopy N content information. Descriptive statistics of canopy $N_{[\%]}$, foliar biomass and canopy $N_{[area]}$ were produced for each of the tree species and PFT included in the analysis. The log-linear regressions between MTCI and canopy N were performed for both resampled and non-resampled datasets. Preliminary analysis showed that using a natural logarithm transformation (log) of the canopy N variables was necessary to fulfil linear regression model assumptions, namely normality and homogeneity of variance of the residuals. The minimum number of data points needed to carry out the regression analysis was fixed at 10. All the coefficients of determination ($r^2$) presented are the adjusted $r^2$ to account for the differences in sample sizes. We calculated the Relative Root Mean Square Error of cross-validation (RRMSEcv, %) using the leave-one-out cross validation method (Clevers and Gitelson, 2013). Its calculation is presented in Eq. (3) following (Yao et al., 2010):

$$RRMSEcv = \sqrt{\frac{1}{n} \times \sum_{i=1}^{n} (P_i - O_i)^2} \times \frac{100}{\overline{O}_i} \qquad (3)$$

where $P_i$ represents the predicted value, $O_i$, the observed value, $\overline{O}_i$ the mean of all observed value and n the total number of measurement. Resampling both datasets as well as linking the plots to the MTCI pixels was done with the PCRaster software (Karssenberg et al., 2010). The statistical analyses were performed in the R environment (R Development Core Team, 2014) and the ggplot2 package was used for the graphics (Wickham, 2009).

## 3 Results

### 3.1 Descriptive statistics

Descriptive statistical analysis of canopy $N_{[\%]}$, canopy $N_{[area]}$ and foliar biomass were performed for each tree species included in the dataset (Table 1). The four most abundant species (*Pinus halepensis*, *Pinus sylvestris*, *Quercus ilex* and *Pinus uncinata)* dominated 667 plots i.e. almost 80% of the plots. The cumulated abundance percentages of ENF, EBF and DBF species were equal to 66 %, 22 % and 9 %, respectively. From this data, it is clear that the forests plots were mainly dominated by ENF

species. On average, *Pinus uncinata* plots had the highest biomass values while *Quercus suber* plots showed the lowest mean value for this variable. Descriptive statistics were also analysed by PFT. The mean canopy $N_{[\%]}$ was lowest for ENF species, 0.97 %N, and highest for DBF trees, 2.17 %N (Fig. 2a). Canopy $N_{[\%]}$ value ranges were equal to 1.91 %N, 2.06 %N , 1.68 %N and 1.42 %N for DBF, EBF, ENF and mixed plots, respectively. The canopy $N_{[area]}$ statistics were analysed by PFT as well (Fig. 2b) and the averaged canopy $N_{[area]}$ values ranged from 1.82 g m$^{-2}$ to 4.61 g m$^{-2}$. A Pearson correlation matrix (Fig. 3) was computed between the variables for the whole dataset. The correlation between each pair of variables was significant and the correlation between canopy $N_{[area]}$ and foliar biomass was strongest (r = 0.88). This result was expected as the foliar biomass was included in the $N_{[area]}$ calculation. This matrix also shows distribution histograms of the three variables. As canopy $N_{[\%]}$ and canopy $N_{[area]}$ distributions are positively skewed, a logarithmic transformation was applied to these variables to fulfil linear model assumptions. Correlation matrices for each DBF, EBF and ENF plots are presented in the Appendix (Fig. A 2 – 4).

## 3.2 Relationship between MTCI and canopy N data at lower spatial resolution

The relationships between MTCI and both canopy $N_{[\%]}$ and canopy $N_{[area]}$ were studied after resampling both datasets to the same, lower, spatial resolution. This was done to investigate the relationship between MTCI and canopy N data when the initial spatial discrepancy between the two datasets was accounted for. The results showed that the log-linear relationships between MTCI and either canopy $N_{[\%]}$ or canopy $N_{[area]}$ were all highly significant (p<0.000). Moreover, the relationship between MTCI and canopy $N_{[\%]}$ was always stronger than the relationship for MTCI and canopy $N_{[area]}$ for each resampling factor. The r$^2$ values of the relationship between MTCI and canopy $N_{[\%]}$ were equal to 0.33, 0.37, 0.34 and 0.42 for 5 km, 10 km, 15 km and 20 km resampled spatial resolution, respectively. The r$^2$ values of the relationship between MTCI and canopy $N_{[area]}$ were equal to 0.20, 0.20, 0.19 and 0.17 at 5 km, 10 km, 15 km and 20 km spatial resolution. The relationship between MTCI and canopy $N_{[\%]}$ at 20 km spatial resolution is shown in Figure 4. Table 2 shows the number of plots per pixel for different pixel sizes (km). As expected, the number of plots per pixel increased with the pixel size, with a mean of 4.1 plots at 20 km spatial resolution. The descriptive statistics of the number of different PFT, species and sampling years per pixel spatial resolution are provided in the Appendix (Table A 2 – A 4).

## 3.3 Relationship between MTCI and canopy N data at original spatial resolution (1 km)

### 3.3.1 Relationship between MTCI and canopy N concentration

The relationships between MTCI and canopy N data were studied at the original spatial resolution (1 km). The results showed that the log-linear regression between MTCI and canopy $N_{[\%]}$ for the whole dataset (n = 846) was highly significant (p<0.000) and had an r$^2$ value of 0.32 and an RRMSEcv value of 18.7 % (Table 3, Fig. 5a). The relationship between MTCI and Canopy $N_{[\%]}$ was also investigated for each PFT individually (Fig. 5b-e). For DBF plots, the relationship between MTCI and canopy $N_{[\%]}$ had an r$^2$ value of 0.24 (n = 80) and was significant. However, although statistically significant, the r$^2$ of the relationship between MTCI and canopy $N_{[\%]}$ for EBF and ENF plots were lower and equal to 0.02 (n = 186) and 0.10 (n = 564), respectively.

The relationship between MTCI and canopy $N_{[\%]}$ was also significant for one individual species, *Fagus sylvatica*. The proportion of explained variance for this species was equal to 0.69 (n = 15). This result, although obtained on a restricted number of plots, showed that the significant relationships between MTCI and canopy $N_{[\%]}$ not only existed when all DBF plots were included but also held for one individual DBF species.

### 3.3.2 Relationship between MTCI and canopy N content

Significant relationships between MTCI and canopy $N_{[area]}$ were found for the whole dataset as for EBF and ENF plots (Table 3). The scatterplots between MTCI and canopy $N_{[area]}$ are presented in Figure 6. The proportion of explained variance was higher for ENF plots compared to the other PFTs and compared to the overall relationship across all plots. The relationship between MTCI and canopy $N_{[area]}$ was also investigated for 10 individual species and one of them showed significant relationships: *Quercus ilex* ($r^2 = 0.10$, p-value < 0.000, n = 160).

## 4 Discussion

### 4.1 Relationship between MTCI and canopy N data at lower spatial resolution

This pre-analysis was undertaken to study the MTCI-canopy N relationships when taking the discrepancy between MTCI original spatial resolution (1 km) and the size of the forest plots (diameter of 6 m) into account. By resampling both datasets to a lower spatial resolution, i.e. 5 km, 10 km, 15 km and 20 km, the obtained values were less impacted by small-scale variations because they were obtained by averaging several values over a larger area. The results showed that the relationship between MTCI and canopy N data was significant and consistent across the resampled spatial resolutions investigated: 5 km, 10 km, 15 km and 20 km. This, however, does not give any indication about the uncertainties resulting from the initial spatial discrepancy between both datasets and about the influence of such uncertainties on the MTCI-canopy N relationship.

### 4.2 Relationship between MTCI and canopy N data at original spatial resolution (1 km)

#### 4.2.1 Canopy N concentration

The overall relationship between MTCI and canopy $N_{[\%]}$ at 1 km spatial resolution for all the forest plots (n = 846) was significant and the $r^2$ value was equal to 0.32 (Table 3, Fig. 5). This result showed that canopy $N_{[\%]}$ could be related to MTCI in Mediterranean forests. The performance of the MTCI vegetation index to detect canopy $N_{[\%]}$ in Mediterranean vegetation was similar to the results obtained from previous studies using spaceborne MTCI at higher spatial resolution. For example, using MTCI computed from the spaceborne RapidEye sensor at 5 m spatial resolution, it was possible to detect canopy $N_{[\%]}$ in grassland savannah and sub-tropical forest with similar coefficients of determination, $r^2 = 0.35$ and $r^2 = 0.52$, respectively (Ramoelo et al., 2012;Cho et al., 2013). However, while there is a consensus regarding MTCI ability for in situ leaf or canopy $N_{[\%]}$ detection in a variety of crops using handheld spectrometers (Tian et al., 2011;Li et al., 2014), there is no general agreement

about MTCI ability for canopy $N_{[\%]}$ detection across vegetation and sensor types at larger scales. For example, MTCI computed from airborne data at 3 m spatial resolution could not be related to canopy $N_{[\%]}$ from a mixed temperate forest (Wang et al., 2016). In this context our finding brings new insight into MTCI $N_{[\%]}$ sensing capabilities at a much coarser spatial resolution (1 km) compared to what has been done before. In these comparisons, it should be taken into account that most previous studies were based on a short sampling campaign while our study incorporates canopy N data from a forest inventory that was carried out during the entire growing season and, therefore, includes differences in phenology.

Investigating the influence of the PFTS on the overall relationship highlighted the difference between DBF, EBF and ENF types of vegetation regarding canopy $N_{[\%]}$ detection by spaceborne MTCI. The relationships between MTCI and canopy $N_{[\%]}$ were significant for all the PFT taken separately (p-value<0.05). However, a higher proportion of variance was explained for DBF and mixed plots ($r^2 = 0.24$ and $r^2 = 0.44$ for DBF and mixed plots, respectively) compared to the other plant functional types ($r^2 = 0.10$ and $r^2 = 0.02$ for ENF and EBF trees, respectively) and the relationship between MTCI and canopy $N_{[\%]}$ was especially weaker for EBF plots. This indicates that the relationship observed for all the forest plots was mainly driven by DBF and mixed plots. This result is different from what was observed by Ollinger et al. (2008) in boreal forests, where canopy $N_{[\%]}$ was related to NIR reflectance for both broadleaf and needleleaf plots taken separately. Moreover, the results obtained for ENF tree species are surprising as previous studies investigating the relationship between foliar $N_{[\%]}$ and in situ measured spectra reported higher $r^2$ values, $r^2 = 0.59$ and $r^2 = 0.81$ in spruce and pine forest, respectively (Stein et al., 2014;Schlerf et al., 2010). The differences in scale and methodology might explain the divergent results compared to previous findings. Indeed, in our study, the analysis is carried out at a much coarser spatial resolution using spaceborne data compared to the fine spatial scale obtained with in situ devices. Moreover, most of these studies were carried out in temperate forests and studies investigating canopy $N_{[\%]}$ detection in Mediterranean regions are scarce. When investigating the relationship between canopy $N_{[\%]}$ and MTCI at the species level, we also found that it was significant for *Fagus sylvatica* plots ($r^2 = 0.69$).

In the literature, the relationship between MTCI and canopy $N_{[\%]}$ is often not stratified by PFT or species (Sullivan et al., 2013;Wang et al., 2016). In this study, we showed that investigating this relationship for each PFT taken separately yielded additional insight. Indeed, to our knowledge the difference in explained variance between DBF and other PFTs in MTCI and canopy $N_{[\%]}$ relationship has not been observed before. Moreover, the results observed for *Fagus sylvatica* plots (n = 15) were consistent with the stronger relationship observed for DBF plots.

### 4.2.2  Canopy N content

The relationship between MTCI and canopy $N_{[area]}$, which was obtained by combining canopy N concentration values with biomass data, was significant across all plots (n = 841) (Table 4, Fig. 6). Although the $r^2$ value was lower for the relationship between MTCI and canopy $N_{[area]}$ ($r^2 = 0.17$) than for the relationship between MTCI and canopy $N_{[\%]}$ ($r^2 = 0.32$), it is interesting to note that canopy $N_{[area]}$ can be related to spaceborne MTCI as remotely sensed detection of canopy $N_{[area]}$ is rarely investigated in forest environments (Mirik et al., 2005). In comparison, previous studies conducted in grasslands reported higher prediction accuracy e.g. by using spaceborne MTCI at 300 m spatial resolution or a simple ratio-type vegetation index computed from

airborne imagery at 1 m spatial resolution, canopy $N_{[area]}$ was detected with $r^2$ values equal to 0.29 and 0.66, respectively (Mirik et al., 2005;Ullah et al., 2012).

The relationship between MTCI and canopy $N_{[area]}$ was only significant for ENF and EBF plots (Fig. 6b-e), with a higher proportion of explained variance for ENF plots ($r^2 = 0.19$). However, when this relationship was investigated at the species scale, significant results were found for *Quercus ilex* (EBF) plots. This is accordance with a previous study examining the remote sensing of canopy $N_{[area]}$ in *Quercus ilex* trees by MTCI computed from in situ spectra ($r^2 = 0.43$) (Pacheco-Labrador et al., 2014).

## 4.3    Comparing results obtained for canopy N concentration and canopy N content

This analysis highlighted the difference between canopy N expressed as a percentage of leaf dry matter (canopy $N_{[\%]}$) and on an area basis (canopy $N_{[area]}$) regarding the log-linear relationship with MTCI for the different PFTs. Canopy $N_{[\%]}$ of DBF and mixed plots showed higher correlation with MTCI compared to EBF and ENF plots while the relationship between canopy $N_{[area]}$ of ENF plots with MTCI was stronger than for any other PFTs. These differences between the log-linear relationship between MTCI and either canopy $N_{[\%]}$ and canopy $N_{[area]}$ can be related to previous findings showing that canopy $N_{[area]}$ but not canopy $N_{[\%]}$ could be detected by MTCI in grassland (Ullah et al., 2012) and by a simple ratio index in heterogeneous rangelands (Mirik et al., 2005) at various spatial scales, 300 m and 1 m, respectively. In the literature, canopy $N_{[\%]}$ is more often used to detect N state of foliage in forest while canopy $N_{[area]}$ is regularly employed in grasslands but also in crops (Clevers and Gitelson, 2013;Schlemmer et al., 2013). Our results showed that, for ENF plots, when biomass was accounted for, as in canopy $N_{[area]}$, the relationship between MTCI and canopy $N_{[area]}$ was stronger compared to canopy $N_{[\%]}$. This suggests that biomass had an influence on and was a confounder of the MTCI-canopy N log-linear relationship.

## 4.4    Possible confounding factors of the MTCI canopy N relationship

The relationships between MTCI and both canopy $N_{[\%]}$ and canopy $N_{[area]}$ were influenced by the PFT of the plots. The relationship between MTCI and canopy $N_{[\%]}$ was stronger for DBF and mixed plots compared to EBF and ENF plots while the opposite was true for the MTCI-canopy $N_{[area]}$ relationship. In the ongoing discussion about the mechanisms underlying the remote sensing of canopy N, some authors argued that the difference in structural properties between different PFTs was a confounding factor of the observed relationship between canopy N and remote sensing data, rendering it spurious (Knyazikhin et al., 2013). Other authors suggested that the role of canopy structure as confounding factor can be explained by an indirect association between canopy N and canopy structure resulting from convergent adaptive processes (Ollinger et al., 2013;Townsend et al., 2013). In this context, our analysis showed that the PFTs of the plots and the biomass had an influence on the MTCI canopy N relationship in a specific type of ecosystem, namely Mediterranean forests. Other confounding factors associated with N availability that might affect the observed relationship possibly include biomass, biomass allocation, leaf area index (LAI), water availability, soil type. The data from the forest inventory used in this analysis, i.e. the Catalonian National Forest Inventory, were extensively studied, showing that water availability was the most limiting factor in this region. Water

availability was positively correlated with both the $N_{[area]}$ and $N_{[\%]}$ in leaves, as well as with foliar and total above-ground biomass through MAP (Sardans et al., 2011;Sardans and Peñuelas, 2013). The MAP also influenced the PFT distribution as DBF plots were located in wetter areas than EBF plots, which were found in wetter sites than ENF plots. Regarding the influence of PFT on the foliar biomass, DBF plots had on average 45% less foliar biomass than EBF or ENF plots (Sardans and Peñuelas, 2013).

## 4.5    Perspectives for future applications

The methodology applied in this paper is different from the usual methodology implemented to detect canopy N concentration in forests. Remote sensing of N in forest canopies by hyperspectral sensors is often coupled with intensive forest sampling measurements. This method has been effective at detecting canopy N concentration locally in a vast range of environments (Serrano et al., 2002;Smith et al., 2002;Townsend et al., 2003;Ollinger et al., 2008;Wang et al., 2016). Applying this technique at larger scales has already been explored. For example, Martin *et al.*, (2008) compiled 137 field plots data from previous studies in various forest types and investigated the possibility to find a common detection algorithm. However, due to the different treatments required as well as the limited swath width associated with the high spatial resolution (from 3 m to 30 m for Hyspex airborne and Hyperion spaceborne sensors, respectively, Wang et al., 2016;Smith et al., 2003), applying imaging spectrometry at a broader scale, although feasible, might reveal time-consuming. Depending on the sensors as well as on the extent of the study area, this might involve correcting the acquired images for atmospheric influences and cloud cover as well as combining several images into a larger scale image. A recent study in northern temperate forests explored the effect of spatial resolution on canopy $N_{[\%]}$ estimation. The results showed that, although the prediction accuracy was reduced compared to what was achieved using PLS regression at higher spatial resolution, it was still possible to estimate canopy $N_{[\%]}$ with $r^2$ between 0.34 and 0.81 using various vegetation indices computed from MODIS reflectance data at 500 m spatial resolution (Lepine et al., 2016). In this context, the methodology applied in this article could be a valuable alternative to explore remote sensing of canopy N at larger scale. Using published data from an extensive field plot inventory, we were able to relate both canopy $N_{[\%]}$ and canopy $N_{[area]}$ to MTCI at 1 km spatial resolution. Although the relationships found were modest, our study contributes to the ongoing discussion about how to remotely sense canopy N over larger area. As MTCI time series (1 km) are readily and almost globally available, it could eventually be possible to assess our approach at a broader scale in different types of biomes. The results obtained for DBF species and *Fagus sylvatica* in particular suggest that this method may be efficient at estimating canopy N in temperate forests. If the strength of the relationship between MTCI and canopy N can further be improved, this could lead to canopy N monitoring possibilities at regional scale. In this context, the new sensors OLCI, onboard Sentinel-3 satellite, and especially MSI, onboard Sentinel-2 satellite might be promising due to their higher spatial resolution, from 10 to 60 m for Sentinel-2. They have bands well positioned to compute the MTCI vegetation index. Although the OTCI, i.e. the successor of the MTCI for the OLCI sensor, is already included in the OLCI level 2b reflectance image, no level 3 product (mosaicked over larger areas and temporally averaged hence similar to the MTCI time series used in this analysis) is available yet. In addition to more detailed remote sensing data, supplementary ground based canopy N observations could better constrain the regression

models as well. Obtaining reliable ground based canopy N data over larger areas and for diverse and globally distributed vegetation types would also be necessary to calibrate and validate global vegetation models, as the model performance will depend on the ground data availability and distribution. Remotely sensed canopy N estimates would also the calibration of such models. In a recent study, the global vegetation model LPJ-Guess was able to simulate the differences in foliar nitrogen between different PFTs but not within one PFT (Fleischer et al., 2015). In this context, improving remotely sensed canopy N estimates

for homogeneous vegetation types would be a beneficial development for such models.

## 5     Conclusion

In this study, we investigated the relationship between spaceborne MTCI from ENVISAT and both canopy $N_{[\%]}$ and canopy $N_{[area]}$ at regional scale in Mediterranean forests. We found significant results across all plots both when the original data were resampled to 5 km, 10 km, 15 km and 20 km and for the original spatial resolution of 1 km. The relationship between MTCI

and canopy N data was also significant for some individual PFTs and species. The $r^2$ values were 0.32 and 0.17 for the overall relationships between MTCI and either canopy $N_{[\%]}$ or canopy $N_{[area]}$, respectively. We highlighted the differences between PFTs and both canopy $N_{[\%]}$ and canopy $N_{[area]}$: the relationship between MTCI and canopy $N_{[\%]}$ was stronger for DBF and mixed plots while canopy $N_{[area]}$ was more linked to MTCI for ENF plots. Such differences in relationships between MTCI and either canopy $N_{[\%]}$ or canopy $N_{[area]}$ were already observed in grasslands ecosystem. Our results showed that MTCI could be related

to canopy N for some individual PFTs, indicating an influence of the PFTs on the MTCI-canopy N relationship. The methodology developed in this study could be investigated at larger scales in different types of ecosystem. While this could already be undertaken using the ENVISAT MTCI 10 year time series as it is almost globally available, ESA new Sentinel-2 satellite launched on 23 June 2015 yields reflectance data at improved spatial and temporal resolution than ENVISAT-MERIS. Canopy N estimates collected through larger scales applications could be exploited in vegetation modelling studies including

both the C and N cycles.

## 6     Data availability

The canopy data used in this study can be obtained from the TRY Plant Trait Database (https://www.try-db.org/TryWeb/Home.php, dataset 91) or by directly contacting the authors.

## 7     Appendix A

This appendix presents the inter-annual variation of canopy $N_{[\%]}$ (Fig. A 1), the correlation matrices for DBF (Fig. A2), EBF (Fig. A3) and ENF plots (Fig. A3) as well as the tables representing the allometric relationships between foliar biomass and

diameter at breast height (DBH, Table A 1), the number of PFT (Table A 2), the number of species (Tables A 3) and the number of sampling years (Table A 4) per resampled pixel, by pixel spatial resolution

## 8    Competing interest

The authors declare that they have no conflict of interest.

## 9    Acknowledgements

This research was funded by The Netherlands Organisation for Scientific Research (NWO) under the project number NWO ALW-GO-AO/14-12. J. Sardans and J. Peñuelas were funded by the European Research Council Synergy grant SyG-2013-610028 IMBALANCE-P, the Spanish Government projects CGL2013-48074-P and the Catalan Government project SGR 2014-

274. We would like to acknowledge Scott Ollinger and Lucie Lepine for their valuable comments and discussions on our research project as well as Ton Markus for his help with the figures presented in this paper.

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

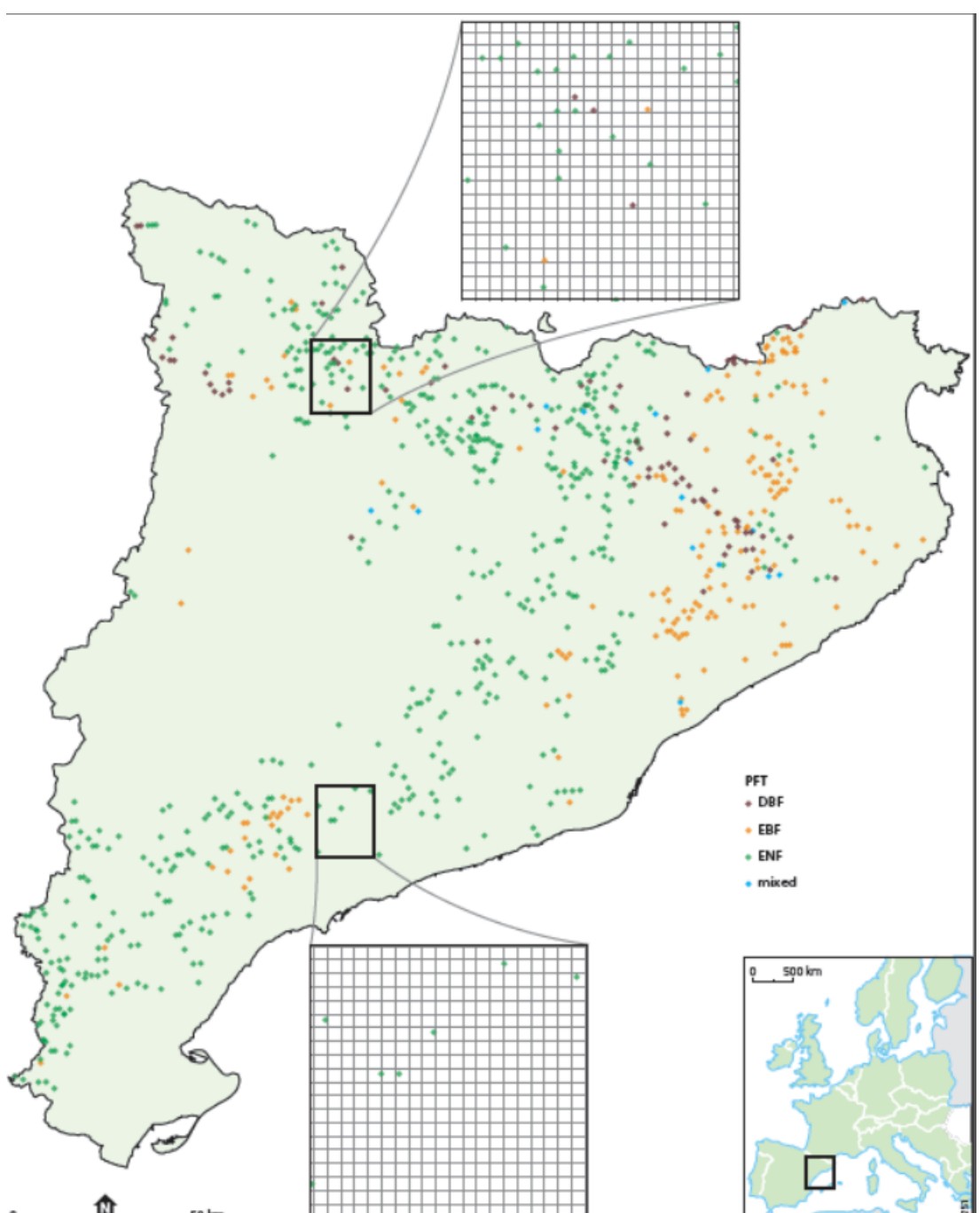

**Figure 1. Map showing the forest plots (n = 846) location in the region of Catalonia, north eastern Spain. Two zoom windows are included showing the density of the plots, one with high density and one with low density, relatively to the MTCI 1 km pixel grid. DBF = Deciduous Broadleaf Forest, EBF = Evergreen Broadleaf Forest, ENF = Evergreen Needleleaf Forest, mixed = mixed forest.**

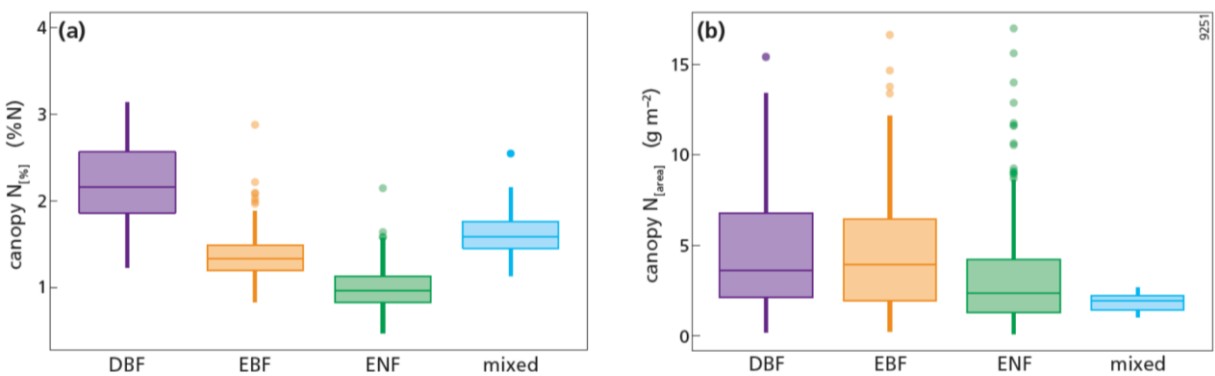

**Figure 2. Boxplot of (a) canopy nitrogen (N) concentration (canopy $N_{[\%]}$, %N) for Deciduous Broadleaf Forest plots (DBF, n = 80), Evergreen Broadleaf Forest plots (EBF, n = 186), Evergreen Needleleaf Forest plots (ENF, n = 564) and mixed forest plots (mixed, n = 16); (b) canopy N content (canopy $N_{[area]}$, g m$^{-2}$) for Deciduous Broadleaf Forest plots (DBF, n = 80), Evergreen Broadleaf Forest plots (EBF, n = 186), Evergreen Needleleaf Forest plots (ENF, n = 563) and mixed forest plots (mixed, n = 12);**

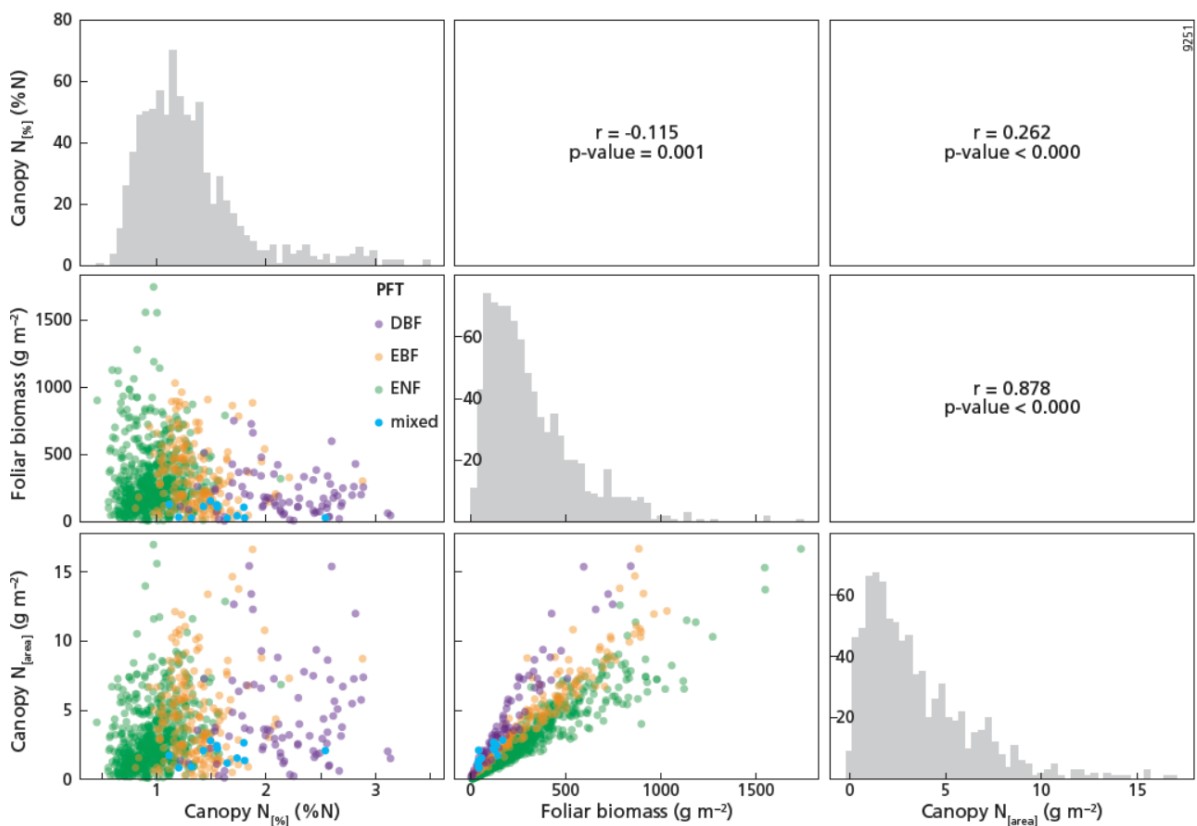


**Figure 3.** The upper right part of this figure shows the Pearson correlation matrix between canopy $N_{[\%]}$ (%N), canopy $N_{[area]}$ (g m$^{-2}$) and foliar biomass (g m$^{-2}$) variables for the whole dataset, n = 841. The diagonal presents the histograms of the variables on the x-axis, while the y-axis represents the number of counts. The lower left part of this figure represents the scatterplots between the variables. PFT = Plant Functional Type, DBF = Deciduous Broadleaf Forest, EBF = Evergreen Broadleaf Forest, ENF = Evergreen

Needleleaf Forest, mixed = mixed forest.

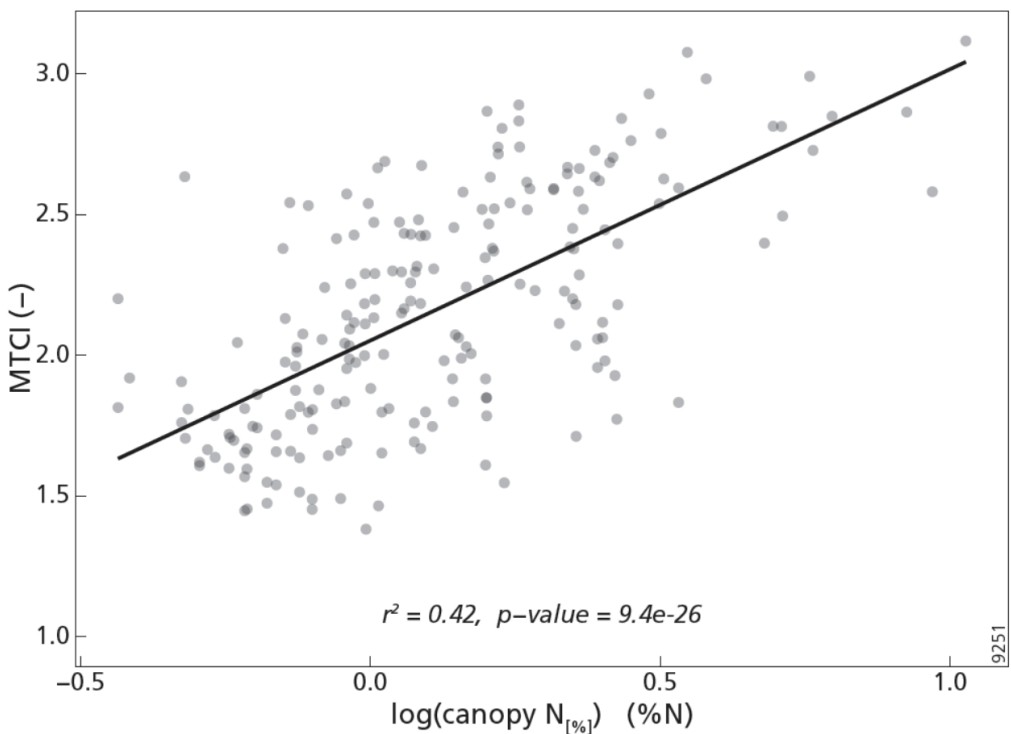

$r^2 = 0.42$, $p$−value $= 9.4e-26$

**Figure 4. Scatterplot between the MERIS Terrestrial Chlorophyll Index (MTCI) (-) and canopy nitrogen concentration (canopy N[%], %N) after resampling the datasets to 20 km-spatial resolution (n = 204).**

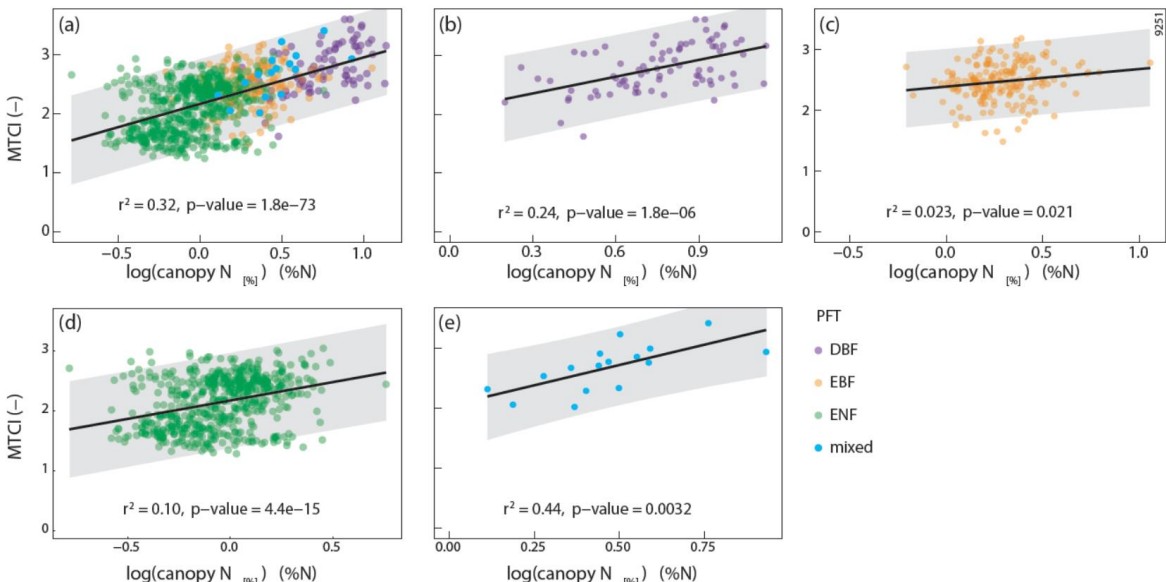

**Figure 5. Scatterplot and log-linear regression line between the MERIS Terrestrial Chlorophyll Index (MTCI) (-) and canopy nitrogen (N) concentration (canopy $N_{[\%]}$, %N) for (a) whole dataset (n = 846); (b) Deciduous Broadleaf Forest plots (DBF, n = 80); (c) Evergreen Broadleaf Forest plots (EBF, n = 186); (d) Evergreen Needleleaf Forest plots (ENF, n = 564); (e) mixed forest plots (n = 16). PFT = Plant functional type. The grey shading represents the prediction intervals (95 %). Canopy $N_{[\%]}$ variable was log transformed to fulfil linear model assumptions.**


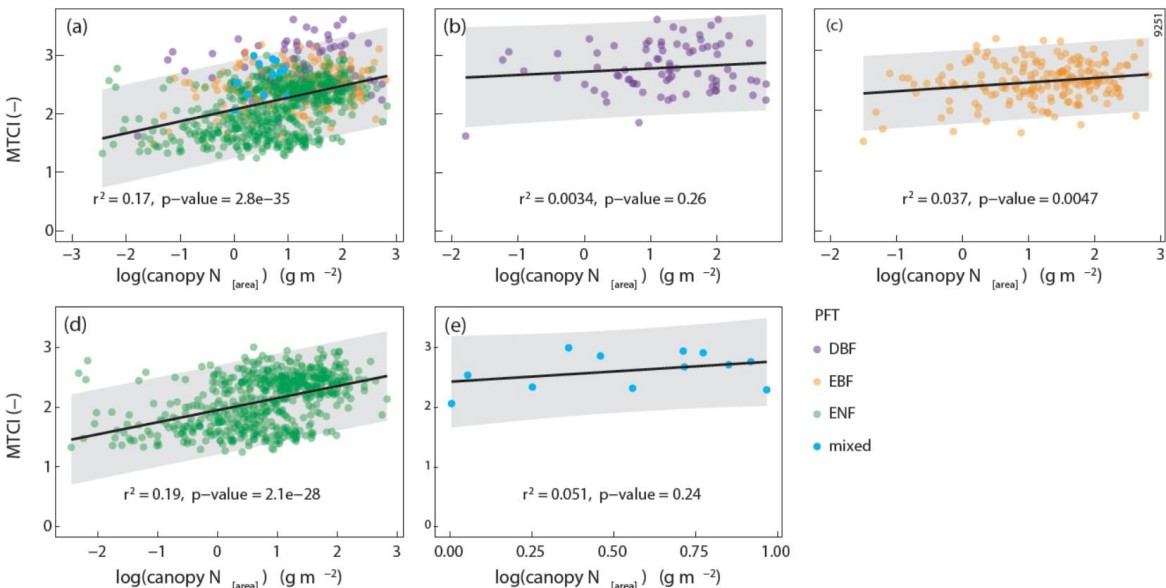


**Figure 6. Scatterplot and log-linear regression line between the MERIS Terrestrial Chlorophyll Index (MTCI) (-) and canopy N content (canopy $N_{[area]}$, g m$^{-2}$) for (a) whole dataset (n = 841); (b) Deciduous Broadleaf Forest plots (DBF, n = 80); (c) Evergreen Broadleaf Forest plots (EBF, n = 186); (d) Evergreen Needleleaf Forest plots (ENF, n = 563); (e) mixed forest plots (n = 12). PFT = Plant functional type. The grey shading represents the prediction intervals (95 %). Canopy $N_{[area]}$ variable was log transformed**
**to fulfil linear models assumptions.**

**Table 1. Descriptive analysis of canopy nitrogen (N) concentration (N$_{[\%]}$, g 100g$^{-1}$), foliar biomass (g m$^{-2}$) and canopy N content (N$_{[area]}$, g m$^{-2}$) by tree species. PFT = Plant Functional Type, DBF = Deciduous Broadleaf Forest, EBF = Evergreen Broadleaf Forest, ENF = Evergreen Needleleaf Forest, mixed = mixed forest, min = minimum, max = maximum, mean = average, sd = standard deviation, [a] codominant plots refer to the plots where two tree species were dominant in the canopy, [b] foliar biomass data was lacking for five of the plots. Foliar biomass and canopy N content statistics are thus measured on a restricted number of plots.**

| Species | PFT | Number of plots | Abundance (% of total number of plots) | Canopy N$_{[\%]}$ (g 100g$^{-1}$) | | | | Foliar biomass (g m$^{-2}$) | | | | Canopy N$_{[area]}$ (g m$^{-2}$) | | | |
|---|---|---|---|---|---|---|---|---|---|---|---|---|---|---|---|
| | | | | min | max | mean | sd | min | max | mean | sd | min | max | mean | sd |
| *Castanea sativa* | DBF | 14 | 1.7 | 1.62 | 2.81 | 2.08 | 0.36 | 18.13 | 425.90 | 203.46 | 123.49 | 0.40 | 11.99 | 4.25 | 2.89 |
| *Fagus sylvatica* | DBF | 15 | 1.8 | 1.22 | 3.13 | 2.28 | 0.61 | 49.94 | 279.86 | 173.54 | 68.70 | 1.21 | 7.40 | 3.96 | 1.95 |
| *Pinus halepensis* | ENF | 240 | 28.4 | 0.56 | 1.57 | 0.90 | 0.19 | 9.58 | 827.80 | 197.23 | 145.54 | 0.09 | 7.29 | 1.77 | 1.33 |
| *Pinus nigra* | ENF | 37 | 4.4 | 0.56 | 1.28 | 0.89 | 0.19 | 32.25 | 923.98 | 294.29 | 224.32 | 0.23 | 8.87 | 2.67 | 2.18 |
| *Pinus pinaster* | ENF | 5 | 0.6 | 0.82 | 1.08 | 0.93 | 0.13 | 271.75 | 718.87 | 501.67 | 211.53 | 2.30 | 7.69 | 4.75 | 2.25 |
| *Pinus pinea* | ENF | 5 | 0.6 | 0.75 | 1.06 | 0.95 | 0.14 | 103.28 | 275.50 | 179.74 | 66.80 | 1.08 | 2.91 | 1.71 | 0.75 |
| *Pinus sylvestris* | ENF | 198 | 23.4 | 0.67 | 2.14 | 1.11 | 0.20 | 10.48 | 828.63 | 326.44 | 181.20 | 0.10 | 12.86 | 3.65 | 2.22 |
| *Pinus uncinata* | ENF | 69 | 8.2 | 0.46 | 1.33 | 0.87 | 0.19 | 183.59 | 1744.50 | 687.22 | 345.21 | 1.41 | 16.97 | 5.92 | 3.25 |
| *Quercus canariensis* | DBF | 3 | 0.4 | 1.97 | 2.78 | 2.25 | 0.46 | 122.11 | 197.85 | 160.32 | 37.87 | 2.41 | 5.51 | 3.71 | 1.61 |
| *Quercus faginea* | DBF | 4 | 0.5 | 1.49 | 2.11 | 1.82 | 0.31 | 10.34 | 419.14 | 233.47 | 187.01 | 0.17 | 8.83 | 4.64 | 4.09 |
| *Quercus humilis* | DBF | 9 | 1.1 | 1.53 | 3.11 | 2.41 | 0.42 | 56.12 | 337.33 | 142.65 | 92.11 | 1.21 | 8.64 | 3.33 | 2.19 |
| *Quercus cerriodes* | DBF | 17 | 2.0 | 1.44 | 2.80 | 2.07 | 0.37 | 12.97 | 834.68 | 262.24 | 237.49 | 0.29 | 15.42 | 5.06 | 4.31 |
| *Quercus ilex* | EBF | 160 | 18.9 | 0.81 | 2.87 | 1.32 | 0.26 | 16.63 | 1033.31 | 378.23 | 238.61 | 0.22 | 16.61 | 4.95 | 3.23 |
| *Quercus petraea* | DBF | 17 | 2.0 | 1.37 | 2.70 | 2.21 | 0.41 | 20.45 | 741.42 | 279.96 | 229.78 | 0.32 | 15.37 | 5.98 | 4.66 |
| *Quercus suber* | EBF | 23 | 2.7 | 1.25 | 2.08 | 1.55 | 0.21 | 26.26 | 219.05 | 110.49 | 55.65 | 0.40 | 4.34 | 1.72 | 0.96 |
| *Codominant*[a] | mixed | 30 (25)[b] | 3.5 | 0.92 | 2.54 | 1.45 | 0.41 | 23.45 | 342.58 | 153.70 | 77.39 | 0.33 | 5.74 | 2.06 | 1.02 |

**Table 2. Descriptive statistics of the number of plots per pixel, for different spatial resolution (km, pixel length). min = minimum, max = maximum, mean = average, sd = standard deviation.**

| Spatial resolution (km) | Number of plots per pixel | | | |
|---|---|---|---|---|
| | min | max | mean | sd |
| 5 | 1 | 6 | 1.44 | 0.77 |
| 10 | 1 | 11 | 2.19 | 1.53 |
| 15 | 1 | 15 | 3.11 | 2.59 |
| 20 | 1 | 22 | 4.09 | 3.74 |


**Table 3. Observed log-linear regression equations between the MERIS Terrestrial Chlorophyll Index (MTCI) (-) and canopy nitrogen concentration (CN$_{[\%]}$, %N) for different subgroups. Number of plots (n), determination coefficient (r$^2$), p-value and Relative Root Mean Square Error of cross-validation (RRMEcv). PFT = Plant Functional type, DBF = Deciduous Broadleaf Forest, EBF = Evergreen Broadleaf Forest, ENF = Evergreen Needleleaf Forest, mixed = mixed forest.**

| group | n | log-linear regression | 95% confidence interval intercept | 95% confidence interval slope | r$^2$ | p-value | RRMSEcv |
|---|---|---|---|---|---|---|---|
| overall | 846 | MTCI = 2.18 + 0.79 log(CN$_{[\%]}$) | [2.15, 2.20] | [0.71, 0.87] | 0.32 | < 0.000 | 17.0 |
| DBF | 80 | MTCI = 2.07 + 0.95 log(CN$_{[\%]}$) | [1.78, 2.36] | [0.59, 1.32] | 0.24 | < 0.000 | 12.7 |
| EBF | 186 | MTCI = 2.39 + 0.29 log(CN$_{[\%]}$) | [2.31, 2.48] | [0.04, 0.54] | 0.02 | 0.021 | 12.4 |
| ENF | 564 | MTCI = 2.13 + 0.61 log(CN$_{[\%]}$) | [2.10, 2.17] | [0.46, 0.76] | 0.10 | < 0.000 | 19.2 |
| mixed | 16 | MTCI = 2.05 + 1.35 log(CN$_{[\%]}$) | [1.63, 2.46] | [0.53, 2.17] | 0.44 | 0.003 | 12.4 |


**Table 4. Observed log-linear regressions equations between the MERIS Terrestrial Chlorophyll Index (MTCI) (-) and canopy nitrogen content ($CN_{[area]}$, g m$^{-2}$) for different subgroups. Number of plots (n), determination coefficient ($r^2$), p-value and Relative Root Mean Square Error of cross-validation (RRMEcv). PFT = Plant Functional type, DBF = Deciduous Broadleaf Forest, EBF = Evergreen Broadleaf Forest, ENF = Evergreen Needleleaf Forest, mixed = mixed forest.**

| group | n | log-linear regression | 95% confidence interval intercept | 95% confidence interval slope | $r^2$ | p-value | RRMSEcv |
|---|---|---|---|---|---|---|---|
| Overall | 841 | MTCI = 2.08 + 0.20 log($CN_{[area]}$) | [2.04, 2.12] | [0.17, 0.23] | 0.17 | <0.000 | 18.7 |
| DBF | 80 | MTCI = 2.72 + 0.06 log($CN_{[area]}$) | [2.58, 2.87] | [-0.04, 0.15] | 0.003 | 0.263 | 14.7 |
| EBF | 186 | MTCI = 2.39 + 0.07 log($CN_{[area]}$) | [2.32, 2.46] | [0.02, 0.12] | 0.04 | 0.005 | 12.4 |
| ENF | 563 | MTCI = 1.94 + 0.20 log($CN_{[area]}$) | [1.91, 1.99] | [0.17, 0.24] | 0.19 | <0.000 | 18.2 |
| mixed | 12 | MTCI = 2.43 + 0.34 log($CN_{[area]}$) | [2.05, 2.82] | [-0.26, 0.95] | 0.05 | 0.236 | 12.8 |

## 11   Appendix

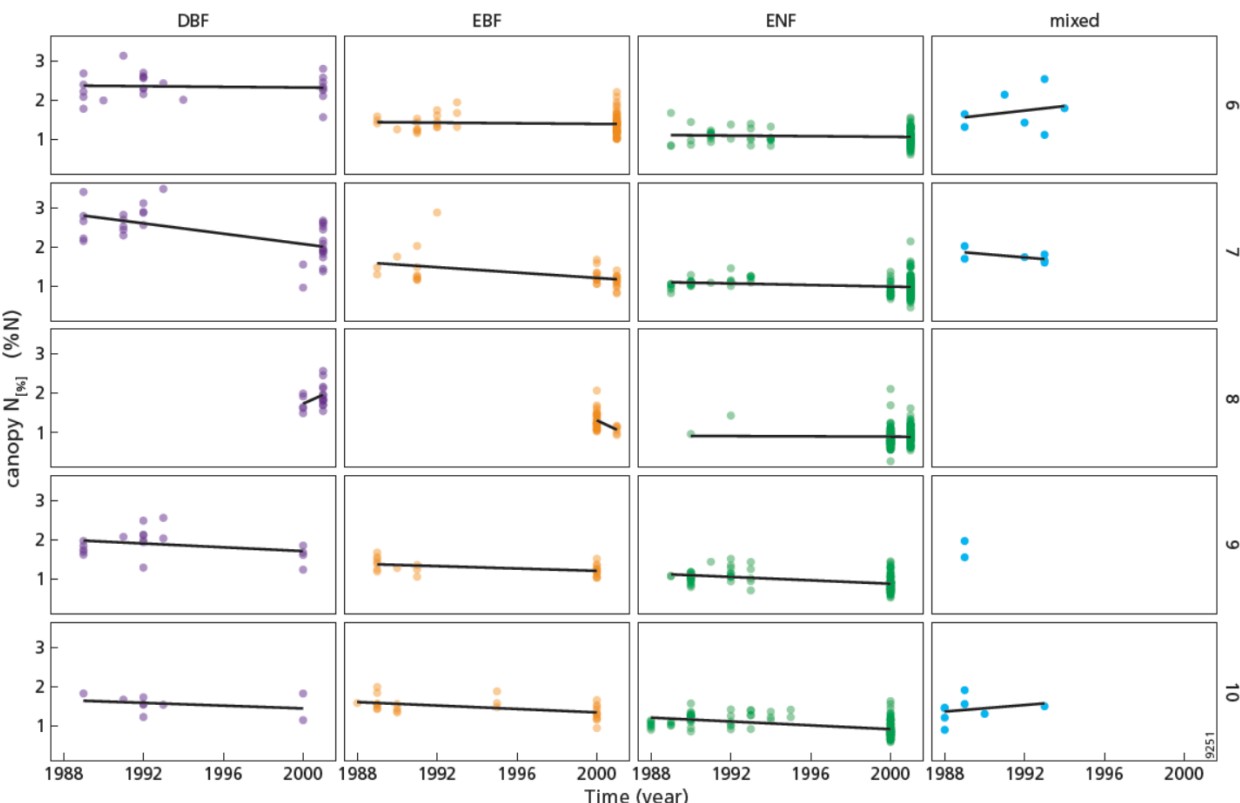

**Figure A1. Inter-annual variation of canopy N[%] (%N) for each month included in the analysis. The numbers 6 – 10 (right side of the figure, row numbers) refer to the month of June, July, August, September and October, respectively. DBF = Deciduous Broadleaf Forest, EBF = Evergreen Broadleaf Forest, ENF = Evergreen Needleleaf Forest, mixed = mixed forest. Each point represents an observation at a forest plot. Note that the forest plots were not sampled multiple times, hence the inter-annual variation encompasses both temporal variation and spatial variation.**

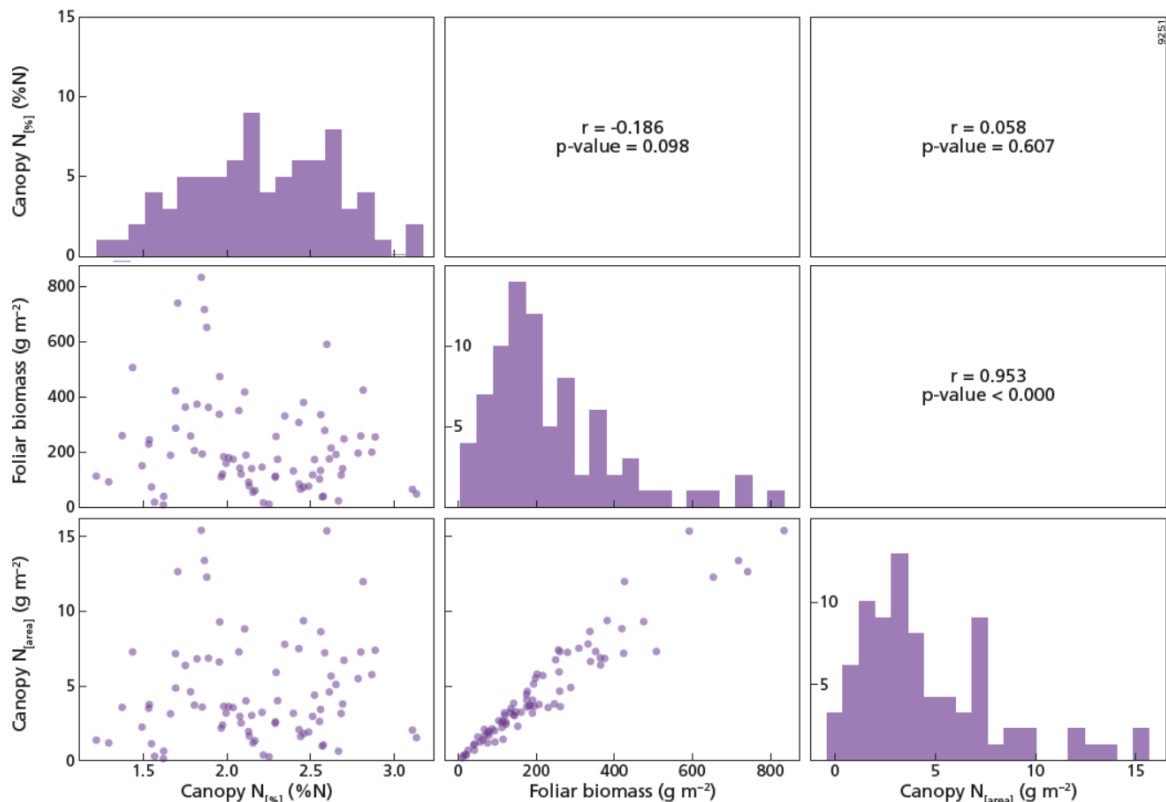

**Figure A 2. The upper right part of this figure shows the Pearson correlation matrix between canopy $N_{[\%]}$ (%N), canopy $N_{[area]}$ (g m-2) and foliar biomass (g m-2) variables for deciduous broadleaf forest plots (DBF), n = 80. The diagonal presents the histogram of the variable on the x-axis, while the y-axis represents the number of counts. The lower left part of this figure represents the scatterplots between the variables.**


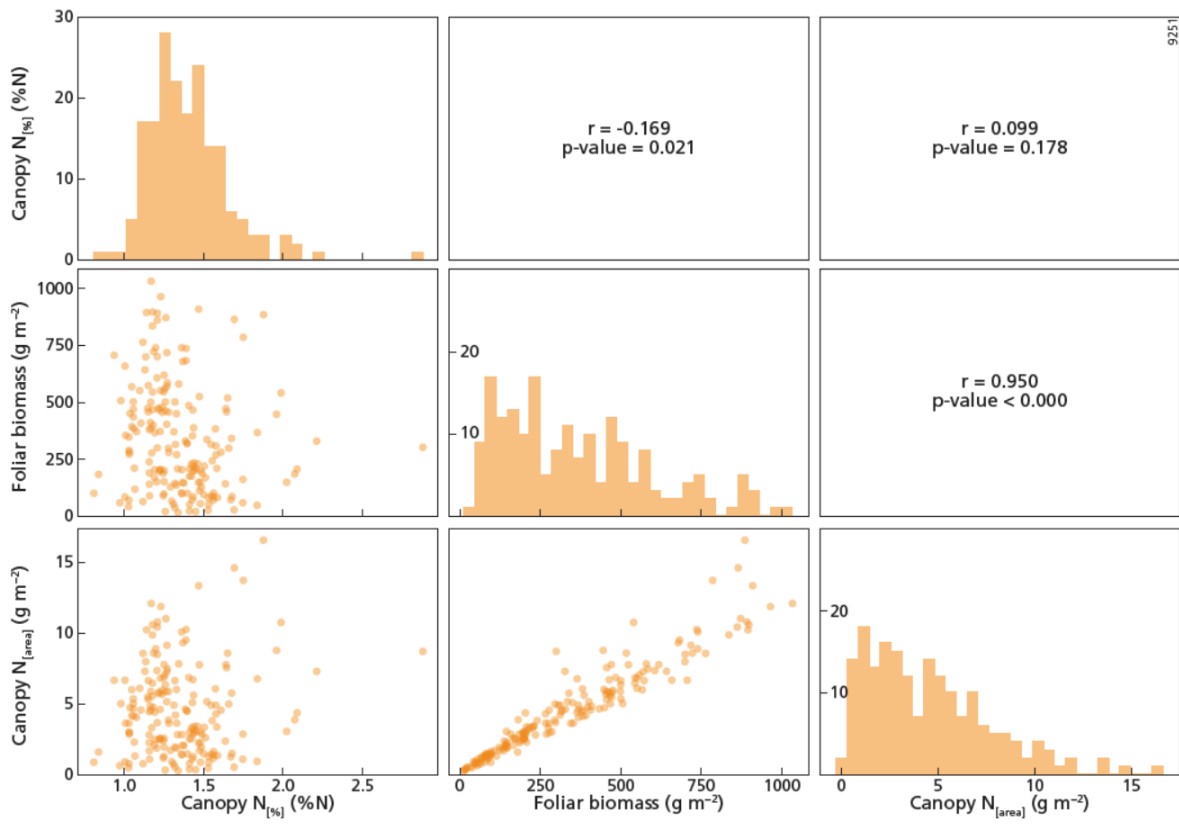

**Figure A 3. The upper right part of this figure shows the Pearson correlation matrix between canopy N[%] (%N), canopy N[area] (g m⁻²) and foliar biomass (g m⁻²) variables for evergreen broadleaf forest (EBF) plots, n = 186. The diagonal presents the histogram of the variable on the x-axis, while the y-axis represents the number of counts. The lower left part of this figure represents the scatterplots between the variables.**

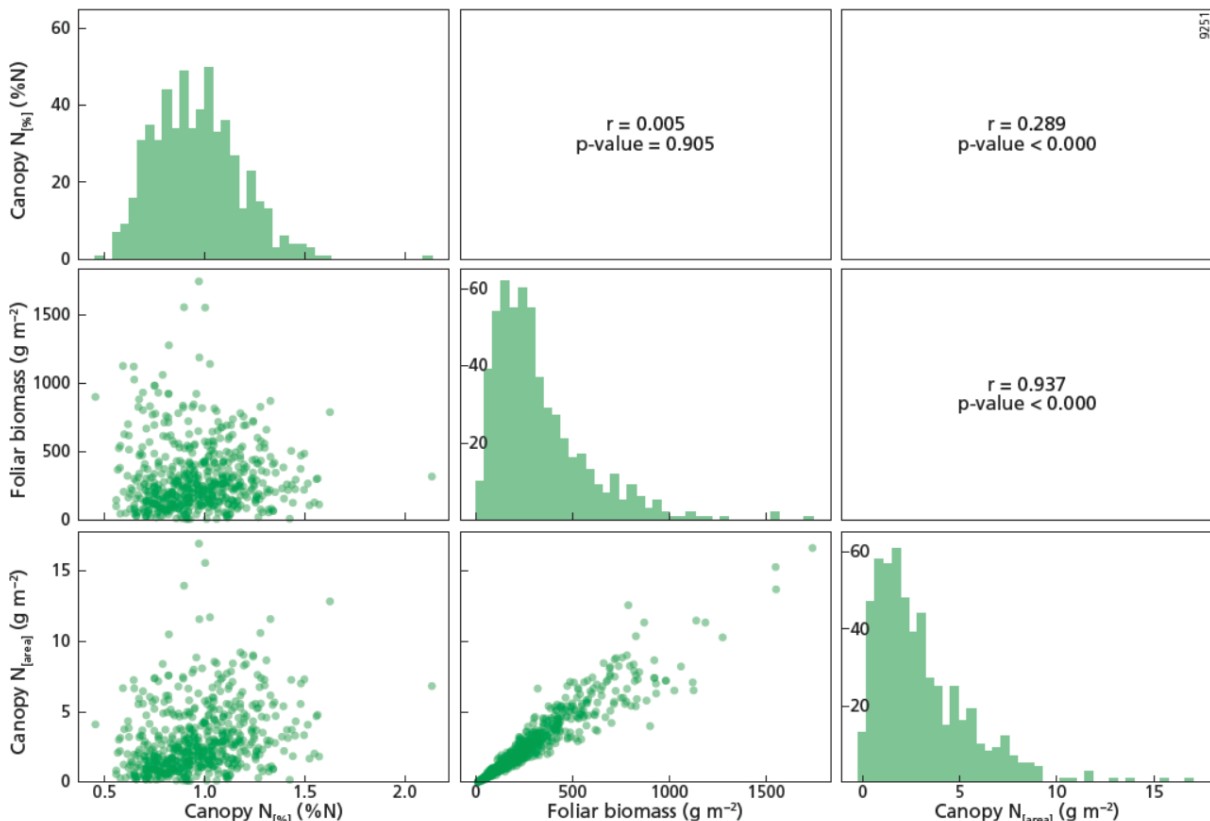

**Figure A 4. The upper right part of this figure shows the Pearson correlation matrix between canopy N[%] (%N), canopy N[area] (g m⁻²) and foliar biomass (g m⁻²) variables for evergreen needleleaf forest (ENF) plots, n = 563. The diagonal presents the histogram of the variable on the x-axis, while the y-axis represents the number of counts. The lower left part of this figure represents the scatterplots between the variables.**


**Table A1. Allometric relationships between foliar biomass and DBH for the different species included in this analysis. DBH = Diameter at breast height (cm). Adapted from (Sardans and Peñuelas, 2015).**

| Species | Foliar biomass = a · DBH$^b$ | | | |
|---|---|---|---|---|
| | a | b | n | r² |
| *Castanea sativa* | 0.032 | 1.669 | 86 | 0.49 |
| *Fagus sylvatica* | 0.026 | 1.546 | 285 | 0.66 |
| *Pinus halepensis* | 0.037 | 1.656 | 2420 | 0.65 |
| *Pinus nigra* | 0.022 | 1.870 | 1641 | 0.65 |
| *Pinus pinaster* | 0.034 | 1.848 | 169 | 0.67 |
| *Pinus pinea* | 0.014 | 2.029 | 335 | 0.72 |
| *Pinus sylvestris* | 0.036 | 1.651 | 2755 | 0.66 |
| *Pinus uncinata* | 0.087 | 1.410 | 770 | 0.62 |
| *Quercus canariensis* | 0.120 | 1.322 | 36 | 0.57 |
| *Quercus faginea* | 0.197 | 0.943 | 170 | 0.40 |
| *Quercus humilis* | 0.047 | 1.462 | 595 | 0.59 |
| *Quercus cerrioides* | 0.023 | 1.805 | 138 | 0.73 |
| *Quercus ilex* | 0.063 | 1.576 | 2151 | 0.60 |
| *Quercus petraea* | 0.014 | 1.888 | 121 | 0.73 |
| *Quercus suber* | 0.026 | 1.446 | 314 | 0.55 |

**Table A 2. Descriptive statistics of the number of plant functional types (PFT) per pixel, by pixel spatial resolution (km). min = minimum, max = maximum, mean = average, sd = standard deviation.**

| Spatial resolution (km) | Number of PFT per pixel | | | |
|:---:|:---:|:---:|:---:|:---:|
| | min | max | mean | sd |
| 5 | 1 | 3 | 1.08 | 0.29 |
| 10 | 1 | 4 | 1.22 | 0.48 |
| 15 | 1 | 4 | 1.34 | 0.61 |
| 20 | 1 | 4 | 1.45 | 0.69 |

**Table A 3. Descriptive statistics of the number of species per pixel, by pixel spatial resolution (km). min = minimum, max = maximum, mean = average, sd = standard deviation.**

| Spatial resolution (km) | Number of species per pixel | | | |
|:---:|:---:|:---:|:---:|:---:|
| | min | max | mean | sd |
| 5 | 1 | 4 | 1.14 | 0.41 |
| 10 | 1 | 4 | 1.38 | 0.67 |
| 15 | 1 | 4 | 1.58 | 0.85 |
| 20 | 1 | 6 | 1.79 | 1.07 |


**Table A 4. Descriptive statistics of the number of sampling years per pixel, by pixel spatial resolution (km). min = minimum, max = maximum, mean = average, sd = standard deviation.**

| Spatial resolution (km) | Number of sampling years per pixel | | | |
|:---:|:---:|:---:|:---:|:---:|
| | min | max | mean | sd |
| 5 | 1 | 2 | 1.02 | 0.15 |
| 10 | 1 | 3 | 1.07 | 0.26 |
| 15 | 1 | 3 | 1.10 | 0.33 |
| 20 | 1 | 3 | 1.14 | 0.40 |
