# Peer review of "Remote sensing of canopy nitrogen at regional scale in Mediterranean forests using the spaceborne MERIS Terrestrial Chlorophyll Index"

_Biogeosciences, 2017_

## Referee Comment (RC1) · Anonymous Referee #1 · 5 Sep 2017

The manuscript shows an interesting study on the use of MERIS data to analyze empirical relationships between MTCI and ground measurements of forest canopy N content and concentration. Foliar N influences a variety of important ecosystem processes so it is clear the interest of exploring the capacity for remote detection of canopy N at regional scales from space-based platforms and the potential of new generation of sensors such as those included in the Copernicus program. However, direct estimation of N in fresh vegetation using remote sensing data is challenging due to its weak effect on leaf reflectance so the influence of structural properties of the canopy and other potential confounding factors related with the input data are key issues to be explored.

The paper is well written and also well-structured and the research questions addressed are relevant and clearly fall within the scope of Biogeosciences. However, my main concern about this work is that, at some point, the paper could be read as a search for correlations without a thoughtful discussion on the different confounding factors that could potentially affect to the observed relationship between satellite and ground data and how these factors could impact the results. A key element in this study is related with the intrinsic limitations of the input data: spatial and temporal mismatch but also, for example, the method used to scale from leaf to canopy N using field sampling strategies. In this work allometric equations are used to relate the diameter of the branches to the leaves dry weight in order to estimate canopy N content. It would be interesting to discuss the accuracy of this method compared to others proposed in the literature to estimate canopy foliar mass per species at the stand level. It would be also interesting to know what is the inter-annual variation of N (ground measurements) in the study region in order to evaluate how this can affect to the discrepancy between timing of ground and satellite data.

Another important issue in this work is the lack of assessment of robustness of empirical models applied using either independent data or statistical techniques (bootstrap). This may be critical when the relationships found could depend on the covariance with other variables as is typically the case in the canopy N estimation from remote sensing. Finally, I also miss in the discussion how the authors consider the results could be potentially useful for monitoring canopy N at regional scale considering the strength of the relationships found and the estimation errors (not analyzed in the paper). Specific comments addressing particular scientific/technical/formal issues follow:

Page 5 line 139. Complementary o alternative reference on methodology applied?

Page 5 line 143. Correct . . ..foliar biomass (N g per square meter. . ..

Page 5 line 153. Reword to clarify content and avoid repetitions

Page 6 line 180. Why the MERIS 300m full resolution product was not used instead?

Page 7 lines 197-199. What about other land cover changes as those caused by forest fires (quite frequent in the study region), where they investigated and filtered?

Page 7 sections 2.3.2 and 2.3.3. Would be interesting to know the number of plots per pixel (average, min and max) at the different spatial resolutions.

Page 8 line 238. Foliar biomass is used in the calculation of canopy N content so the correlation is obviously strongest

Page 9 line 254. Higher instead of lower

Page 9 line 269. R2 for Quercus ilex?

Page 9 section 4.1. Could the authors elaborate here on how this could affect to the regional estimation of canopy N using new generation Sentinel-2 and 3 with improved spatial resolutions?

Page 11 line 315. Any hypothesis on the stronger relationship found for DBF plots? Further investigation on the proportion of the variance explained by other potential confounding factors would be desirable (same in lines 329 and 341)

Page 11 lines 332-335. This has been already stated in the results sections. This apply for other paragraphs in this section, authors should avoid to repeat the results and focus on the discussion

Page 12 lines 152-153. I would recommend to include the analysis in this paper using information acquired in the forest inventory used for the study.

Page 20 FIGURE 1. Please clarify if the plots represented in the map are all the forest inventory plots (2300?) or 1075 (after temporal and spatial filtering) or 846/841 finally used in the analysis. I would recommend including only the plots used in the analysis.

---

## Referee Comment (RC2) · Anonymous Referee #2 · 20 Oct 2017

General comments: The paper aims to investigate the potential of using MTCI to map regional variations in canopy nitrogen (N). The study uses field measurements of canopy N for a large number of forest plots situated across Catalonia to derive empirical relationships between N and MTCI data across a range of spatial resolutions (1 - 20 km). The study also aims to identify the influence of plant functional type on the observed relationships. Whilst the premise of the work may be interesting, there are a number of questions and comments, some of which are fairly fundamental, which I feel need addressing before this manuscript can be considered for publication. The comments are provided in the hope that they may help improve the manuscript and its subsequent impact.

[Figure]

Specific comments: I am not entirely convinced of the justification for reducing the spatial resolution of the MTCI data. Why degrade the 1 km product? The MERIS sensor on board ENVISAT is no longer operational (which the author's should note). The authors do note that a variation of the MTCI can be calculated from Sentinel-2 but this is a sensor with a higher spatial resolution then MERIS so what is the justification for making the data worse? Especially since the forest plots were substantially smaller than the original 1 km pixel size in the first instance. Averaging 6 m plots over a 1 km grid would "reduce small-scale variations (line 279)" so why 5, 10, 15 and 20 km also? Without this information the paper appears to be more of an academic exercise as opposed to addressing a tangible issue.

One of the main justifications for the study is that "limited research has been conducted to sense canopy N in Mediterranean ecosystems and even more so in Mediterranean forests", yet there is no discussion of the importance of these ecosystems, or their N content. More information should be included to justify the significance of this sentence.

More information is required on how the forest plot data are deemed suitable for comparison with the MTCI data. There are several questions here:

1. Is the year of data collection an issue for the correlation? Perhaps colouring the points in figure 4 based on year of in situ collection may be useful e.g. were there any climatically anomalous years that could have influenced the MTCI relationships?

2. How well do the 6m forest data plots represent the 1, 5, 15 and 20 km grid scales? There isn't any information as to how many points were included in each grid square when the data were resampled at each resolution. What was the distribution of values (mean, SD)?

3. Can homogenous species plots be observed from satellite imagery at 5 – 20 km resolution? Surely the plots are going to be mixed species at this scale?

The results presented, whilst statistically significant have quite low r2 values, which

indicates that the precision with which N can be predicted will be low, even though there is a statistically significant relationship between the two variables. The authors do not comment on this but I think they should as this has practical implications for their suggested approach.

It would be useful for the authors to suggest possible reasons why the reported statistically significant regressions are only explain 20 - 30% of the variation at best. The authors indicate that these r2 values are somewhat lower than MODIS so why not just use MODIS?

Technical corrections: The first sentence of the abstract is quite long. Consider fragmenting and re-wording to improve impact.

Line 11 and throughout – Data "is" should be changed to data "are" since data are plural

Line 13 etc. – The abstract should include some justification as to why the work is important. This could be more clearly explained in the abstract as opposed to simply saying x did this and we are doing that. The key question is why?

Line 31: Delete "," after processes

Line 35: Insert "," after (N g m-2)

Line 48: Delete "Currently"

Line 49: Insert "," either side of from and sensors

Line 48 – 52: This is a very long sentence. Consider fragmenting.

Line 53: No need for a new paragraph

Line 68: "were aimed" is an odd choice of words. Consider re-wording

Line74: Do the authors mean "a few studies" or "few studies"? It's not clear as no references are referred to.

Line 83: "stated that the NIR – canopy N relationship was not necessarily spurious as plant traits have been known to covary along the leaf economic spectrum" This statement needs further explanation. What is meant by the leaf economic spectrum?

Line 89: "MTCI time series could be applied to estimate canopy N at a larger scale" Be careful with the terms scale here. Do you mean over a larger spatial extent?

Line 106: Suggests that there are 1075 forest plots but line 123 suggest that there are 2300 and in line 2017 there are 846 plots. Were some removed from the sample?

Line 110: What are the re-sampled resolutions and what is the justification for this?

Line 117: duplicate word "create"

Line 150: "Several (up to two times) " does not make sense. Several suggests three or more. Consider re-wording.

Line 200: MTCI was not re-sampled as the product was already a 1 km product

Line 303-204: "This enabled us to investigate the relationships between MTCI and canopy N data independently of differences in initial support size." I don't entirely agree. Just because they now match on a spatial grid does not mean that the difference in sampling support size no longer matters. The crucial point is how well do the 6 m forest data represent the 1 km grid scale? Anything can be re-sampled. Whether it makes sense to do so is a different question.

Section 2.3.3. It seems a bit odd to investigate relationships at a lower resolution before you investigate it at the original spatial resolution.

Line 215: Refer to section numbers as opposed to "explained above"

Lines 219 and 220: delete the word "then"

Line 223: "The spatial analyses were done with the PCRaster software" It is not clear what spatial analyses were "done". Consider re-wording.

[Figure]

Figure3: I am not sure what the purpose of this figure is since some of the variables being correlated are actually included in the calculation of others e.g. biomass and N concentration are both used to calculate N content – they are bound to be correlated. Hence line 238 is not really a finding.

Line 282: I don't understand what this sentence means I'm afraid "This shows that, when the influence of the discrepancy between the original datasets was taken into account, MTCI and canopy N data were linked" what discrepancies were observed?

Line 294 "there is no general agreement about MTCI ability for canopy N[%] detection across vegetation and sensor types" Can the authors bring any insights as to why this may be the case? What are the issues?

Line 315-316 Consider re-wording. Also note that there were only 15 plots of Fagus sylvatica! Can you make such a conclusion based on relatively few samples?

Line 348 "Other authors, although agreeing that canopy structural properties needed to be accounted for, suggested that a direct biochemical link between canopy N and reflectance data was not necessary to detect canopy N with reflectance data (Ollinger et al., 2013;Townsend et al., 2013)." What did the authors suggest was necessary?

Section 4.4 doesn't really come to any conclusions or suggest reasons for the PFT differences and so it is somewhat superfluous as it stands. Better to integrate this in a wider discussion or include some more detailed interpretation of the data.

Lines 359-3622. I do not follow this point here. What treatments were required and what "might reveal laborious" Consider re-wording.

---

## Author Comment (AC1) · 27 Nov 2017

*We would like to thank the reviewer for reviewing our manuscript.*

The manuscript shows an interesting study on the use of MERIS data to analyze empirical relationships between MTCI and ground measurements of forest canopy N content and concentration. Foliar N influences a variety of important ecosystem processes so it is clear the interest of exploring the capacity for remote detection of canopy N at regional scales from space-based platforms and the potential of new generation of sensors such as those included in the Copernicus program. However, direct estimation of N in fresh vegetation using remote sensing data is challenging due to its weak effect on leaf reflectance so the influence of structural properties of the canopy and other potential confounding factors related with the input data are key issues to be explored.

The paper is well written and also well-structured and the research questions addressed are relevant and clearly fall within the scope of Biogeosciences.

*Thank you for your nice comment.*

However, my main concern about this work is that, at some point, the paper could be read as a search for correlations without a thoughtful discussion on the different **confounding factors** that could potentially affect to the observed relationship between satellite and ground data and how these factors could impact the results.

*We understand your concern about this paper being a search for correlations and would like to stress that we did a directed search rather than a random search. Remote sensing of canopy N (especially handheld and airborne) has already been extensively investigated and vegetation indices based on the red-edge region, on which MTCI is based, have been repeatedly used (Schlemmer et al., 2013;Li et al., 2014;Cho et al., 2013;Clevers and Gitelson, 2013;Dash and Curran, 2004). In this study, we want to extend on the existing analyses by including spaceborne remote sensing. We will explain this in the introduction part of the manuscript. Next, we will also address the possible confounding factors and their effects on the MTCI canopy N relationship in the discussion part of the manuscript. The potential confounding factors include e.g. biomass, canopy structure, LAI, as well as geomorphological variables (Sardans et al., 2011;Sardans et al., 2016).*

A key element in this study is related with the intrinsic limitations of the input data: spatial (1) and temporal mismatch (2) but also, for example, the method used to scale from leaf to canopy N using field sampling strategies (3).

*1) The reviewer is correct, since the beginning of this project we were aware of the limitation of the dataset included. We chose to use the data from the Catalonia National Forest Inventory because it includes many plots that are well spread over the forested region of Catalonia. The spatial mismatch has been addressed by resampling both the MTCI product and the canopy N ground measurements to the same and lower spatial resolution. Then, we analyzed the relationship between both dataset, taking the spatial discrepancy into account. The results showed that the*

correlation between the resampled canopy N and MTCI were significant regardless of the resampled pixel size. Moreover, an analysis investigated the influence of the spatial resolution on the remote sensing of canopy nitrogen. They could show that, even though the percentage of explained variance was reduced by going from high spatial resolution product to a low spatial resolution one (500 m), it was still possible to observe significant relationship between coarse spatial resolution remote sensing data and ground measurements (Lepine et al., 2016). This is mentioned in the discussion part of the manuscript in the section 4.5 (Line 362 – 365).

2) We addressed the temporal mismatch by averaging the MTCI product by month over the 10 years acquisition period, and selecting only the summer months, i.e. May-October, which corresponds to the growing season. By doing this, we decrease the influence of annual anomaly on the results. Moreover, the different selection criteria applied on the dataset, ensured that the plots that had undergone a land cover change were removed from the analysis. The consequence of this is that among the 846 plots included in the analysis, 625 were measured between 2000 and 2001. This is presented in the table 1. Finally, as you suggested in a later comment, an analysis of the inter-annual variation of canopy N data will be included in the manuscript.

| Year | 1988 | 1989 | 1990 | 1991 | 1992 | 1993 | 1994 | 1995 | 2000 | 2001 |
|---|---|---|---|---|---|---|---|---|---|---|
| Number of plot measured | 8 | 47 | 46 | 35 | 44 | 29 | 9 | 3 | 304 | 321 |

**Table 1. Number of plots included in the analysis by sampling year.**

3) To scale from leaf to canopy N, we use the leaf N value averaged of three individuals trees as the plot canopy N value. This methodology, i.e. using leaf N concentration averaged over several individuals as the plot level value, is common (Schlerf et al., 2010). In our study 96% of the plots were monospecific and 4% of the plots contained only two species, therefore we did not weight the average by the species abundance (Smith and Martin, 2001;Townsend et al., 2003;McNeil et al., 2007).

The paragraph about the leaf sampling method has been changed to stress that most of the plots were monospecific (Line 155):

*A proportion of 96% of the plots included in this analysis were monospecific. 4% of the plots had two codominant species. For these plots, two leaf samples were collected, one for each of the codominant species found on the plots.*

In this work, allometric equations are used to relate the diameter of the branches to the leaves dry weight in order to estimate canopy N content. It would be interesting to discuss the accuracy of this method compared to others proposed in the literature to estimate canopy foliar mass per species at the stand level.

Thank you for your comment. When analyzing this further, we found that the information provided in the original version of the manuscript about biomass calculation was incorrect. The foliar biomass data were calculated using allometric equations based on the diameter at breast height (DBH). The DBH was measured for all the trees present on the plot. This information is provided in two articles that also include data from the Catalonian National Forest Inventory (Vilà et al., 2003;Sardans and Peñuelas, 2015).

The paragraph in the canopy N data section 2.2.1 was changed accordingly (Line 148):

*Along with the canopy N[%] data, we used foliar biomass data (g m-2) acquired during the same forest inventory (n = 2286). The foliar biomass data were obtained for each plot from allometric*

*equations relating the diameter at breast height to the leave dry weight. These allometric equations were species specific (Sardans et al. (2015), Vila et al. (2003), Table in supplementary information).*

It would be also interesting to know what is the inter-annual variation of N (ground measurements) in the study region in order to evaluate how this can affect to the discrepancy between timing of ground and satellite data.

We agree with you, the inter-annual variation of the ground measurements of canopy N is indeed essential due to the temporal discrepancy between our two datasets. As we have a large datasets covering the complete sampling period, studying the inter-annual evolution of the canopy N ground measurements would be possible. We will include this analysis in the revised manuscript.

Another important issue in this work is the lack of assessment of robustness of empirical models applied using either independent data or statistical techniques (bootstrap). This may be critical when the relationships found could depend on the covariance with other variables as is typically the case in the canopy N estimation from remote sensing.

Thank you, we agree with your comment. In order to assess the robustness of the relationships between MTCI and canopy N, a leave-one-out cross validation could be calculated for each of the relationships presented in the analysis. This would yield a Root Mean Square Error value that would give information about the prediction error of these relationships. This additional analysis will be included in the revised manuscript.

Finally, I also miss in the discussion how the authors consider the results could be potentially useful for monitoring canopy N at regional scale considering the strength of the relationships found and the estimation errors (not analyzed in the paper).

Thank you for your comment. The goal of this case-study analysis was to explore the feasibility of canopy N detection at regional scale using MTCI. Although the relationships are modest, our study contributes to the ongoing discussion about how to map canopy N over larger area, which could also lead to canopy N monitoring possibilities. This will be explained in the discussion part of the manuscript. We will also calculate the prediction intervals of canopy N data.

Specific comments addressing particular scientific/technical/formal issues follow:

Page 5 line 139. Complementary o alternative reference on methodology applied?

The explanation on the allometric relationship has been changed (Line 137):

*Along with the canopy N[%] data, we used foliar biomass data (g m-2) acquired during the same forest inventory (n = 2286). The foliar biomass data were obtained for each plot from allometric equations relating the diameter at breast height to the leave dry weight. These allometric equations were species specific (Sardans et al. (2015), Vila et al. (2003), Table in supplementary information).*

Page 5 line 143. Correct : : :.foliar biomass (N g per square meter: : :.

This has been changed in the text:
*foliar biomass (dry matter g per square meter of ground area, g $m^{-2}$)*

Page 5 line 153. Reword to clarify content and avoid repetitions

The sentence was clarified in the text:

*A proportion of 96% of the plots included in this analysis were monospecific and had a single dominant tree species. There were 30 plots with two codominant species. For these plots, two leaf samples were collected, one for each of the codominant species found on the plots.*

Page 6 line 180. Why the MERIS 300m full resolution product was not used instead?

Thank you for your question. We indeed first looked at using the MERIS 300 m full resolution reflectance images. These images were not used for our analysis for several reasons. The 300 m full resolution reflectance images available from the ESA are not corrected for cloud cover and atmospheric influences. Moreover, there is no temporally averaged product available at full resolution. This means that one image of the 300 m full resolution reflectance data is available every three to four days from 2002 until 2012. Each of the images included in this analysis would thus need to be atmospherically corrected (365/4 *10 ~ 912 images). This would have been very time intensive.

In this context, the MTCI 1 km level 3 product presented several advantages. It is a readily usable product that has been corrected for atmospheric influences and cloud cover and was monthly averaged. The availability of the MTCI monthly product made it possible for us to relate the ground canopy N measurements to 10 years monthly averaged without involving time consuming images processing. We believe that this way we could decrease part of the uncertainty of relating ground measurement to any daily remotely sensed reflectance value measured several years later. Finally, MTCI product is available for the extent of the Catalonia region in one single image, while the MERIS full resolution product can sometimes only partly cover the region and therefore each image would have had to be selected individually.

This will be more clearly explained in the Material and Method part, section 2.2.2 "MTCI product" of the revised manuscript (Line 159).

Page 7 lines 197-199. What about other land cover changes as those caused by forest fires (quite frequent in the study region), where they investigated and filtered?

The land cover changes caused by forest fires were not investigated in a separate way. As Globcover 2009 the land cover map includes a sparse vegetation class, which we believe is how the vegetation appears after a forest fire, the change due to forest fire should be accounted for when excluding sparse vegetation class from the analysis.

Page 7 sections 2.3.2 and 2.3.3. Would be interesting to know the number of plots per pixel (average, min and max) at the different spatial resolutions.

Thank you, we agree. The number of plots per resampled pixel size are shown in the table 2. This table will be added to the manuscript in the Result section 3.2 "Relationship between MTCI and canopy N data at lower spatial resolution".

Table 2. Mean, minimum, maximum and standard deviation of the number of plots per pixel by the pixel spatial resolution (km).

| Pixel spatial resolution (km) | average number of plots/pixel | minimum number of plots/pixel | maximum number of plots/pixel | standard deviation of the number of plots/pixel |
|---|---|---|---|---|
| 5 | 1.5 | 1 | 6 | 0.8 |
| 10 | 2.3 | 1 | 11 | 1.5 |
| 15 | 3.2 | 1 | 15 | 2.6 |
| 20 | 4.5 | 1 | 22 | 4 |

Page 8 line 238. Foliar biomass is used in the calculation of canopy N content so the correlation is obviously strongest

Thank you, we agree with your comment. This was not intended to be understood as a new finding but we rather wished to be fully explicit about the correlation between the variables. The original sentence was replaced by (Line 238):

*The correlation between each pair of variables was significant and the correlation between canopy $N_{[area]}$ and foliar biomass was strongest (r = 0.88). This result was expected as the foliar biomass was included in the $N_{[area]}$ calculation.*

Page 9 line 254. Higher instead of lower
This has been changed.

Page 9 line 269. R2 for Quercus ilex?
The r2 value for *Quercus ilex* plots has been added in the text:

*The relationship between MTCI and canopy N[area] was also investigated for 10 individual species and one of them showed significant relationships: Quercus ilex (r2 = 0.10, n = 160).*

Page 9 section 4.1. Could the authors elaborate here on how this could affect to the regional estimation of canopy N using new generation Sentinel-2 and 3 with improved spatial resolutions?

Due to the higher spatial resolution of the MSI sensor onboard Sentinel 2 and the bands well positioned in the red edge region, remote sensing of canopy N at regional scale might be promising. However, a pre-processed product similar to the MTCI time series should first be made available to reproduce the methodology applied in this study. This has been addressed in the discussion section 4.5 "Perspective for larger scale applications" (Line 353):

*In this context, the new sensors OLCI, onboard Sentinel 3 satellite, and especially MSI, onboard Sentinel 2 satellite, might also be promising due to their higher spatial resolution, from 10 to 60 m for Sentinel 2. They have bands well positioned to compute the MTCI vegetation index. Although the OLCI Terrestrial Chlorophyll Index (OTCI), the successor of the MTCI for the OLCI sensor, is already included in the OLCI level 2b reflectance image, no level 3 product similar to the MTCI time series used in this analysis, i.e. mosaicked over larger areas and temporally averaged, is available yet.*

Page 11 line 315. Any hypothesis on the stronger relationship found for DBF plots? Further investigation on the proportion of the variance explained by other potential confounding factors would be desirable (same in lines 329 and 341)

Thank you for your comment. We will address the effects of the potential confounder on the relationship, among which biomass and canopy structure are related to the different PFTs, in the discussion part of the revised manuscript.

Page 11 lines 332-335. This has been already stated in the results sections. This apply for other paragraphs in this section, authors should avoid to repeat the results and focus on the discussion.

Thank you, we agree that repeating this information several times might be unnecessary. In this instance, we wanted to remind the reader what we are going to address in the next paragraph.

Page 12 lines 152-153. I would recommend to include the analysis in this paper using information acquired in the forest inventory used for the study.

Thank you for your comment. We agree that using additional data besides canopy N and foliar biomass would make the analysis stronger. However, only biomass and foliar concentration was measured during the forest inventory. Additional physiological data related to the forest plots is thus not available.

260

Page 20 FIGURE 1. Please clarify if the plots represented in the map are all the forest inventory plots (2300?) or 1075 (after temporal and spatial filtering) or 846/841 finally used in the analysis. I would recommend including only the plots used in the analysis.

265

Thank you, the number of plots represented in the figure (n = 846) has been added to the figure caption (Line 577). The number "1075" plots was mistake from a former version of the manuscript and has been changed where it appeared in the text.

270   *Figure 1. Map showing the forest plots (n = 846) location in the region of Catalonia, north eastern Spain. DBF = Deciduous Broadleaf Forest, EBF = Evergreen Broadleaf Forest, ENF = Evergreen Needleleaf Forest, mixed = mixed forest.*

References:

275   Cho, M. A., Ramoelo, A., Debba, P., Mutanga, O., Mathieu, R., van Deventer, H., and Ndlovu, N.: Assessing the effects of subtropical forest fragmentation on leaf nitrogen distribution using remote sensing data, Landscape Ecology, 28, 1479-1491, doi:10.1007/s10980-013-9908-7, 2013.

Clevers, J. G. P. W., and Gitelson, A. A.: Remote estimation of crop and grass chlorophyll and nitrogen content using red-edge bands on sentinel-2 and-3, International Journal of Applied Earth Observation and Geoinformation, 23, 344-351, doi:10.1016/j.jag.2012.10.008, 2013.

280   Dash, J., and Curran, P. J.: The MERIS terrestrial chlorophyll index, International Journal of Remote Sensing, 25, 5403-5413, doi:10.1080/0143116042000274015, 2004.

Lepine, L. C., Ollinger, S. V., Ouimette, A. P., and Martin, M. E.: Examining spectral reflectance features related to foliar nitrogen in forests: Implications for broad-scale nitrogen mapping, Remote Sensing of Environment, 173, 174-186,

285   doi:10.1016/j.rse.2015.11.028, 2016.

Li, F., Miao, Y., Feng, G., Yuan, F., Yue, S., Gao, X., Liu, Y., Liu, B., Ustin, S. L., and Chen, X.: Improving estimation of summer maize nitrogen status with red edge-based spectral vegetation indices, Field Crops Research, 157, 111-123, doi:10.1016/j.fcr.2013.12.018, 2014.

McNeil, B. E., Read, J. M., and Driscoll, C. T.: Foliar nitrogen responses to elevated atmospheric nitrogen deposition in nine

290   temperate forest canopy species, Environmental Science and Technology, 41, 5191-5197, 2007.

Sardans, J., Rivas-Ubach, A., and Peñuelas, J.: Factors affecting nutrient concentration and stoichiometry of forest trees in Catalonia (NE Spain), Forest Ecology and Management, 262, 2024-2034, doi:10.1016/j.foreco.2011.08.019, 2011.

Sardans, J., and Peñuelas, J.: Trees increase their P: N ratio with size, Global Ecology and Biogeography, 24, 147-156, 10.1111/geb.12231, 2015.

295   Sardans, J., Alonso, R., Carnicer, J., Fernández-Martínez, M., Vivanco, M. G., and Peñuelas, J.: Factors influencing the foliar elemental composition and stoichiometry in forest trees in Spain, Perspectives in Plant Ecology, Evolution and Systematics, 18, 52-69, 10.1016/j.ppees.2016.01.001, 2016.

Schlemmer, M., Gitelson, A., Schepers, J., Ferguson, R., Peng, Y., Shanahan, J., and Rundquist, D.: Remote estimation of nitrogen and chlorophyll contents in maize at leaf and canopy levels, International Journal of Applied Earth Observation and

300   Geoinformation, 25, 47-54, doi:10.1016/j.jag.2013.04.003, 2013.

Schlerf, M., Atzberger, C., Hill, J., Buddenbaum, H., Werner, W., and Schüler, G.: Retrieval of chlorophyll and nitrogen in Norway spruce (Picea abies L. Karst.) using imaging spectroscopy, International Journal of Applied Earth Observation and Geoinformation, 12, 17-26, doi:10.1016/j.jag.2009.08.006, 2010.

Smith, M. L., and Martin, M. E.: A plot-based method for rapid estimation of forest canopy chemistry, Canadian Journal of

305   Forest Research, 31, 549-555, 2001.

Townsend, P. A., Foster, J. R., Chastain Jr, R. A., and Currie, W. S.: Application of imaging spectroscopy to mapping canopy nitrogen in the forest of the central Appalachian mountains using hyperion and AVIRIS, IEEE Transactions on Geoscience and Remote Sensing, 41, 1347-1354, doi:10.1109/TGRS.2003.813205, 2003.

Vilà, M., Vayreda, J., Gracia, C., and Ibáñez, J. J.: Does tree diversity increase wood production in pine forests?, Oecologia,

310   135, 299-303, 2003.

---

## Author Comment (AC2) · 27 Nov 2017

*We would like to thank the reviewer for reviewing our manuscript.*

General comments: The paper aims to investigate the potential of using MTCI to map regional variations in canopy nitrogen (N). The study uses field measurements of canopy N for a large number of forest plots situated across Catalonia to derive empirical relationships between N and MTCI data across a range of spatial resolutions (1 - 20 km). The study also aims to identify the influence of plant functional type on the observed relationships. Whilst the premise of the work may be interesting, there are a number of questions and comments, some of which are fairly fundamental, which I feel need addressing before this manuscript can be considered for publication. The comments are provided in the hope that they may help improve the manuscript and its subsequent impact.

Specific comments: I am not entirely convinced of the justification for reducing the spatial resolution of the MTCI data. Why degrade the 1 km product? The MERIS sensor on board ENVISAT is no longer operational (which the author's should note). The authors do note that a variation of the MTCI can be calculated from Sentinel-2 but this is a sensor with a higher spatial resolution then MERIS so what is the justification for making the data worse? Especially since the forest plots were substantially smaller than the original 1 km pixel size in the first instance. Averaging 6 m plots over a 1 km grid would "reduce small-scale variations (line 279)" so why 5, 10, 15 and 20 km also? Without this information the paper appears to be more of an academic exercise as opposed to addressing a tangible issue.

*Thank you for your comment. This issue is indeed important as it addresses the spatial discrepancy between the two datasets, i.e. the forest plots used for the ground measurements (6 m) and the MTCI pixel size (1 km). Studying the relationship between canopy N and MTCI at a lower, degraded spatial resolution was done to overcome this spatial discrepancy and to study the relationships between our variables independently of the initial spatial discrepancy. This step also allowed us to study the influence of the spatial resolution of the MTCI pixel. The results showed that the relationships between the variable were not strongly affected by the resampling factor.*

*Moreover, we are afraid that there might have been some confusion. In section 4.1, we say that averaging 6 m plots over a lower spatial resolution, i.e. 5, 10, 15 and 20 km (and not 1 km), would reduce small scale variations. To make this more clear, we added the resampled pixel size in the text in section 4.1 (Line 278 - 282) as well as in the objective in the Introduction and Material and Methods sections were we felt it was lacking.*

*Finally, we agree with you that the fact that the MERIS sensor came to an end in 2012 is an essential information linked to our analysis. This was also noted in the original version of the manuscript (Line 166):*
*While the ESA ENVISAT satellite mission producing MERIS data came to an end in 2012*

One of the main justifications for the study is that "limited research has been conducted to sense canopy N in Mediterranean ecosystems and even more so in Mediterranean forests", yet there is no discussion of the importance of these ecosystems, or their N content. More information should be included to justify the significance of this sentence.

Thank you for your comment. Remote sensing of canopy N has not been done a lot in Mediterranean forest. We will include justification about the spatial and ecological importance, especially regarding species diversity (Vilà-Cabrera et al., 2018), of Mediterranean forests in the introduction part of the manuscript. We will also mention that there is a lack of studies and supporting data for Mediterranean ecosystems in global vegetation models studies (Line 40 – 41).

More information is required on how the forest plot data are deemed suitable for comparison with the MTCI data. There are several questions here:

1. Is the year of data collection an issue for the correlation? Perhaps colouring the points in figure 4 based on year of in situ collection may be useful e.g. were there any climatically anomalous years that could have influenced the MTCI relationships?

Thank you for your suggestion. In the figure 4, the plot ground measurements were averaged by pixel (20 km) and sampling month (over 10 years). This means that on the same pixel, the plots were measured during the same month and located at maximum 20 km from each other. However, these plots might have been measured during different years. The Material and Methods section 2.3.2 " Relationship between MTCI and canopy N data at lower spatial resolution" (Line 200) has been edited to make it more clear.
Colouring the points based on the sampling year might be an option, however, many of these points are likely to be the results of the average over several years. In table 1, we present the numbers of plots by sampling year. In the revised manuscript, we will also present the average number of year per pixel size.

| Year | 1988 | 1989 | 1990 | 1991 | 1992 | 1993 | 1994 | 1995 | 2000 | 2001 |
|---|---|---|---|---|---|---|---|---|---|---|
| Number of plots measured | 8 | 47 | 46 | 35 | 44 | 29 | 9 | 3 | 304 | 321 |

Table 1. Number of plots included in the analysis by sampling year.

2. How well do the 6m forest data plots represent the 1, 5, 15 and 20 km grid scales? There isn't any information as to how many points were included in each grid square when the data were resampled at each resolution. What was the distribution of values (mean, SD)?

Thank you for your comment. This is indeed important if we wish to evaluate the effectiveness of the resampling method to overcome the initial difference in support size. The table 2 shows the number of plots for each resampled pixel size. This table will also be included in the revised manuscript.

Table 2. Mean, minimum, maximum and standard deviation of the number of plots per pixel by the pixel spatial resolution (km).

| Pixel spatial resolution (km) | average number of plots/pixel | minimum number of plots/pixel | maximum number of plots/pixel | standard deviation of the number of plots/pixel |
|---|---|---|---|---|
| 5 | 1.5 | 1 | 6 | 0.8 |
| 10 | 2.3 | 1 | 11 | 1.5 |
| 15 | 3.2 | 1 | 15 | 2.6 |
| 20 | 4.5 | 1 | 22 | 4 |

3. Can homogenous species plots be observed from satellite imagery at 5 – 20 km resolution? Surely the plots are going to be mixed species at this scale?

Thank you for your comment. Indeed, the plots are likely to be mixed species and mixed PFT too. This is the reason why the analysis by species and PFT was not carried out at this step of the analysis. Similar to the two previous question, we will present a table showing the average number of PFT and species per resampled pixel size in the revised manuscript.

The results presented, whilst statistically significant have quite low r2 values, which indicates that the precision with which N can be predicted will be low, even though there is a statistically significant relationship between the two variables. The authors do not comment on this but I think they should as this has practical implications for their suggested approach. It would be useful for the authors to suggest possible reasons why the reported statistically significant regressions are only explain 20 - 30% of the variation at best.

Thank you for your comment. The obtained r2 are indeed low, between 0.10 and 0.40. We would like to stress that other studies report similar or sometimes, lower r2, for analyses conducted at higher spatial resolution and in more controlled conditions (Cho et al., 2013;Wang et al., 2016). The obtained results were compared with existing literature in the section 4.2.1 "Canopy N concentration detection". This section was modified to stress the differences in spatial resolution (Line 288 - 296):

*The performance of the MTCI vegetation index to detect canopy N[%] in Mediterranean vegetation was similar to the results obtained from previous studies using spaceborne MTCI at higher spatial resolution. For example, using MTCI computed from the spaceborne RapidEye sensor at 5 m spatial resolution, it was possible to detect canopy N[%] in grassland savannah and sub-tropical forest with similar coefficients of determination, r2 = 0.35 and r2 = 0.52, respectively (Ramoelo et al., 2012;Cho et al., 2013). However, while there is a consensus regarding MTCI ability for in situ leaf or canopy N[%] detection in a variety of crops using handheld spectrometers (Tian et al., 2011;Li et al., 2014), there is no general agreement about MTCI ability for canopy N[%] detection across vegetation and sensor types at larger scales. For example, MTCI computed from airborne data at 3 m spatial resolution could not be related to canopy N[%] from a mixed temperate forest (Wang et al., 2016). In this context our finding brings new insight into MTCI N[%] sensing capabilities at a much coarser spatial resolution (1 km) compared to what has been done before.*

Moreover, we will address the potential confounding variables of the relationship between MTCI and canopy N in the discussion part of the revised manuscript. These confounding variables include biomass, canopy structure, LAI as well as climatic and geomorphological variables (Sardans et al., 2011;Sardans et al., 2016).

The goal of this case-study analysis was to explore the feasibility of canopy N detection at regional scale using MTCI. Although the statistical relationship are modest, the results provide spatio-temporal indicators of canopy N and we think that this analysis brings a valuable information in the ongoing discussion about the feasibility of sensing canopy N over larger spatial extent.

The authors indicate that these r2 values are somewhat lower than MODIS so why not just use MODIS?
Thank you for your question. We did not use MODIS because our goal was to test the relationship for the MTCI vegetation index. Vegetation indices products available for MODIS are NDVI and EVI, which have showed lower correlation with canopy N compared to MTCI due to saturation problem at high N concentration (Schlemmer et al., 2013;Pacheco-Labrador et al., 2014).
Moreover, the study we referred to in the discussion (Line 363 - 365), did in indeed get higher r2 using MODIS images. Their methodology was different as there was no temporal

discrepancy between their ground measurements and the satellite images acquisition. They worked with 7 x 7km MODIS tiles, while the MERIS MTCI level 3 product is available from the ESA for the extend of the whole region (and actually even Europe) in one single image.

5    Technical corrections:
The first sentence of the abstract is quite long. Consider fragmenting and re-wording to improve impact.

Thank you, the sentence has been changed (Line 10):
10   *Canopy nitrogen (N) concentration and content are linked to several vegetation processes at leaf and canopy levels.  Therefore, canopy N concentration is a state variable in global vegetation models with coupled carbon (C) and N cycles.*

Line 11 and throughout – Data "is" should be changed to data "are" since data are Plural
15   This has been changed.

Line 13 etc. – The abstract should include some justification as to why the work is important. This could be more clearly explained in the abstract as opposed to simply saying x did this and we are doing that. The key question is why?
20   This will be added

Line 31: Delete "," after processes This has been changed.
Line 35: Insert "," after (N g m-2) This has been changed.
Line 48: Delete "Currently" This was changed. This has been changed.
25   Line 49: Insert "," either side of from and sensors This has been changed.
Line 48 – 52: This is a very long sentence. Consider fragmenting.
This has been changed.

Line 53: No need for a new paragraph. This has been changed.
30

Line 68: "were aimed" is an odd choice of words. Consider re-wording
This has been changed.
*most studies were carried out in agricultural crops using MTCI values computed from in situ hyperspectral reflectance data*
35   Line74: Do the authors mean "a few studies" or "few studies"? It's not clear as no references are referred to.
We mean here "few studies". A reference has been added.

Line 83: "stated that the NIR – canopy N relationship was not necessarily spurious
40   as plant traits have been known to covary along the leaf economic spectrum" This statement needs further explanation. What is meant by the leaf economic spectrum?
This will be explained in more details

Line 89: "MTCI time series could be applied to estimate canopy N at a larger scale" Be
45   careful with the terms scale here. Do you mean over a larger spatial extent?
This has been changed.

Line 106: Suggests that there are 1075 forest plots but line 123 suggest that there are 2300 and in line 2017 there are 846 plots. Were some removed from the sample?
50   Thank you for noticing this mistake. On Line 106, 846 plots should have been written in place of 1075. This has been changed. The 2300 plots (Line 125) refers to the original number of plots included in the forest inventory before applying the selection criteria explained in the Material and Methods section (Line 216 – 218).

55   Line 110: What are the re-sampled resolutions and what is the justification for this?

*Thank you for your comment. The resampled resolution are now clearly indicated in the text (Line 108):*

*Next, both data sets are resampled to the same, lower, spatial resolutions, i.e. 5 km, 10 k, 15 km and 20 km, in order to overcome the initial spatial discrepancy between MTCI spatial resolution (1 km) and the size of the forest plots (6 m).*

Line 117: duplicate word "create": This has been changed.

Line 150: "Several (up to two times) " does not make sense. Several suggests three or more. Consider re-wording.

Thank you for your comment. This has been changed:
*There were 30 plots with two codominant species. For these plots, two leaf samples were collected, one for each of the codominant species found on the plots.*

Line 200: MTCI was not re-sampled as the product was already a 1 km product.

We agree with you that the initial spatial resolution of the MTCI product is 1 km. In the manuscript this is called the "higher spatial resolution". However, in our study we first analyze the relationship between MTCI and canopy N data after resampling both dataset to lower a spatial resolution (section 2.3.2 Relationship between MTCI and canopy N data at lower spatial resolution", Line 199). This was done to overcome the initial spatial discrepancy between the two datasets. To make this more clear, the resampled spatial resolution was added (line 201):

*2.3.2   Relationship between MTCI and canopy N data at lower spatial resolution*
*In a first step, the relationships between MTCI and canopy N data values were investigated after resampling both datasets to the same, lower, spatial resolution. The resampled spatial resolutions were 5 km, 10 km, 15 km, and 20 km.  This was done because of the initial difference in support size between MTCI spatial resolution and the forest plots size (i.e. 1 km and 6 m, respectively). This enabled us to investigate the relationships between MTCI and canopy N data independently of differences in initial support size.*

Line 303-204: "This enabled us to investigate the relationships between MTCI and canopy N data independently of differences in initial support size." I don't entirely agree. Just because they now match on a spatial grid does not mean that the difference in sampling support size no longer matters. The crucial point is how well do the 6 m forest data represent the 1 km grid scale? Anything can be re-sampled. Whether it makes sense to do so is a different question.

Thank you for your comment. We chose to include the resampling analysis in our study due to the initial spatial discrepancy between the two datasets used, i.e. the ground measurements (6 m plots) and the MTCI pixel (1 km). By resampling both dataset to a lower and equal spatial resolution, we wanted to study the relationship between the two variables when the spatial discrepancy was accounted for. As you mention it, this process is dependent on spatial representativity of the plots found on the new resampled pixels. This has been addressed on the page 2 and 3 of this response. We looked at the number of plots per resampled pixel, the number of different species and PFT per resampled pixel as well as the number of different sampling years per resampled pixel. Moreover, Globcover 2009 landcover map was used to exclude from the resampling calculations the pixels that did not classify as natural vegetation. This was done to address the patchiness of the vegetation as explained on Line 204 – 206:

*Tthe Globcover 2009 land cover map was used to exclude from the resampling computation the MTCI pixels located on land surface without natural vegetation cover. As for the forest plots, MTCI pixels*

*whose land cover class corresponded to rainfed cropland, mosaic between croplands and natural vegetation, sparse vegetation or artificial surfaces were excluded from the upscaling analysis*

Section 2.3.3. It seems a bit odd to investigate relationships at a lower resolution before you investigate it at the original spatial resolution.

Thank you for your comment. We choose to study the relationship at the lower spatial resolution before because we wanted to explore the relationship at higher spatial resolution in more details, i.e. by also PFT and species into account. At lower spatial resolution, this information about PFT and species is lost due to the resampling process.

Line 215: Refer to section numbers as opposed to "explained above"
The section number has been added.

Lines 219 and 220: delete the word "then" This has been changed.

Line 223: "The spatial analyses were done with the PCRaster software" It is not clear what spatial analyses were "done". Consider re-wording.

This has been added (Line 223):
*Resampling both datasets as well as linking the plots to the MTCI pixels was done with the PCRaster software*

Figure3: I am not sure what the purpose of this figure is since some of the variables being correlated are actually included in the calculation of others e.g. biomass and N concentration are both used to calculate N content – they are bound to be correlated. Hence line 238 is not really a finding.

This figure was included in the manuscript to summarize the information about the forest dataset. We also wished to be explicit about the correlation between the variables included in the analysis. The line 238 was not meant to be understood as a new finding but rather a statement about the correlation existing between the canopy N content and the biomass. This figure also show how the variables distribution are skewed.
The original sentence was replaced by (Line 238):
*The correlation between each pair of variables was significant and the correlation between canopy $N_{[area]}$ and foliar biomass was strongest (r = 0.88). This result was expected as the foliar biomass was included in the $N_{[area]}$ calculation.*

Line 282: I don't understand what this sentence means I'm afraid "This shows that, when the influence of the discrepancy between the original datasets was taken into account, MTCI and canopy N data were linked" what discrepancies were observed?

This sentence referred to the spatial discrepancy between the spatial resolution of the MTCI (1 km) and the forest plots (6m). The sentence has been rephrased.

Line 294 "there is no general agreement about MTCI ability for canopy N[%] detection across vegetation and sensor types" Can the authors bring any insights as to why this may be the case? What are the issues?

Thank you. The issue is that across different studies that investigate remote sensing of canopy N at larger scale, i.e. larger than with a handheld spectrometer, the prediction accuracy of the result is highly variable. When using a RapidEye at 5m resolution the prediction is similar to what we obtain. While the results obtained by Wang, 2016, even though at very high spatial resolution (3 m) were not significant. As we could think that the coarse spatial resolution might be a big obstacle to sense canopy N, our results showed that

even though the spatial resolution was comparatively low (min 1km) we still get significant results. This thus adds to the discussion about canopy N remote sensing.

The paragraph has been edited to stress these distinctions (Line 291 – 303):
*The overall relationship between MTCI and canopy N[%] at 1 km spatial resolution for all the forest plots (n = 846) was significant and the r2 value was equal to 0.32 (Fig. 5). This result showed that canopy N[%] could be related to MTCI in Mediterranean forests. The performance of the MTCI vegetation index to detect canopy N[%] in Mediterranean vegetation was similar to the results obtained from previous studies using spaceborne MTCI at higher spatial resolution. For example, using MTCI computed from the spaceborne RapidEye sensor at 5 m spatial resolution, it was possible to detect canopy N[%] in grassland savannah and sub-tropical forest with similar coefficients of determination, r2 = 0.35 and r2 = 0.52, respectively (Ramoelo et al., 2012;Cho et al., 2013). However, while there is a consensus regarding MTCI ability for in situ leaf or canopy N[%] detection in a variety of crops using handheld spectrometers (Tian et al., 2011;Li et al., 2014), there is no general agreement about MTCI ability for canopy N[%] detection across vegetation and sensor types at larger scales. For example, MTCI computed from airborne data at 3 m spatial resolution could not be related to canopy N[%] from a mixed temperate forest (Wang et al., 2016). In this context our finding brings new insight into MTCI N[%] sensing capabilities at a much coarser spatial resolution (1 km) compared to what has been done before.*

Line 315-316 Consider re-wording. Also note that there were only 15 plots of Fagus sylvatica! Can you make such a conclusion based on relatively few samples?

Thank you. The sentence has been changed.
*Moreover, when studied separately, the results observed for Fagus sylvatica plots (n = 15) were consistent with the stronger relationship observed for DBF plots.*

Moreover, we agree that compared to the general size of our dataset, 15 Beech plots is relatively small subset but it provides a first indication. Moreover, many studies studying canopy N detection include very few samples in total. For example in a mixed temperate forest, Wang et al. (2016) included 26 plots (30 x 30 m) in total. In 2008, Huber et al. studied the remote sensing of canopy in a temperate forest using 28 plots (50 x 50 m) in total. In an arid shrubland, Mitchell et al. (2012) studied 35 plots (7 x 7m). These examples concern remote sensing of canopy N in general, i.e. they do not necessarily include MTCI, nor vegetation indices and use remote sensing sensors with high spatial resolution. Nonetheless, this can still give you an impression that 15 plots is not so uncommon.

Line 348 "Other authors, although agreeing that canopy structural properties needed to be accounted for, suggested that a direct biochemical link between canopy N and reflectance data was not necessary to detect canopy N with reflectance data (Ollinger et al., 2013;Townsend et al., 2013)." What did the authors suggest was necessary?

Thank you. Ollinger et al. (2008) used overall reflectance in the NIR and found a correlation with canopy N in boreal forest. Knyazikhin et al. (2013) argued that this relationship was spurious and resulted solely from differences in canopy structures linked to differences in PFT. Ollinger et al. (2013) and Townsend et al. (2013) argued that the observed relationship was not the result of a direct biochemical mechanism between nitrogen and incoming radiation but rather of an indirect link between nitrogen and plant structure, which would result from adaptive processes. We will modify the existing paragraph and add this information to the revised manuscript.

Section 4.4 doesn't really come to any conclusions or suggest reasons for the PFT differences and so it is somewhat superfluous as it stands. Better to integrate this in a wider discussion or include some more detailed interpretation of the data.

Thank you for your suggestion. We will integrate the differences induced by the PFT in a wider discussion about the possible confounding factors that might influence the relationship between MTCI and canopy N. These confounding factors include biomass, canopy structure and climatic variables.

Lines 359-362. I do not follow this point here. What treatments were required and what "might reveal laborious" Consider re-wording.

Thank you. This sentence refers to the different treatments applied to images obtained with imaging spectroscopy at high spatial resolution with airborne or spaceborne sensors. These images need to be corrected for the influence of the atmosphere and clouds (atmospheric correction). Moreover depending on the initial sensor swath width as well as the size of the region to investigate, the images might need to be mosaicked into an image covering a larger area than the initial image acquired by the sensor. Depending on the time period for the ground measurements, the remote sensing images might also need to be temporally averaged. These treatments are similar to the one that would need to done to the MODIS images, as described on page 4 of this response.

This has been added to the manuscript (Line 359):

*However, due to the different treatments required as well as the limited swath width associated with the high spatial resolution (from 3 m to 30 m for Hyspex airborne and Hyperion spaceborne sensors, respectively, Wang et al., 2016;Smith et al., 2003), applying imaging spectroscopy at a broader scale might reveal laborious. Depending on the sensors as well as on the extent of the study area, this might involve correcting the acquired images for atmospheric influences and cloud cover as well as combining several images into a larger scale image.*

References:

Bartlett, M. K., Ollinger, S. V., Hollinger, D. Y., Wicklein, H. F., and Richardson, A. D.: Canopy-scale relationships between foliar nitrogen and albedo are not observed in leaf reflectance and transmittance within temperate deciduous tree species, Botany, 89, 491-497, 2011.

Cho, M. A., Ramoelo, A., Debba, P., Mutanga, O., Mathieu, R., van Deventer, H., and Ndlovu, N.: Assessing the effects of subtropical forest fragmentation on leaf nitrogen distribution using remote sensing data, Landscape Ecology, 28, 1479-1491, doi:10.1007/s10980-013-9908-7, 2013.

Huber, S., Kneubühler, M., Psomas, A., Itten, K., and Zimmermann, N. E.: Estimating foliar biochemistry from hyperspectral data in mixed forest canopy, Forest Ecology and Management, 256, 491-501, doi:10.1016/j.foreco.2008.05.011, 2008.

Knyazikhin, Y., Schull, M. A., Stenberg, P., Mõttus, M., Rautiainen, M., Yang, Y., Marshak, A., Latorre Carmona, P., Kaufmann, R. K., Lewis, P., Disney, M. I., Vanderbilt, V., Davis, A. B., Baret, F., Jacquemoud, S., Lyapustin, A., and Myneni, R. B.: Hyperspectral remote sensing of foliar nitrogen content, Proceedings of the National Academy of Sciences, 110, E185–E192, 10.1073/pnas.1210196109, 2013.

Mitchell, J. J., Glenn, N. F., Sankey, T. T., Derryberry, D. R., and Germino, M. J.: Remote sensing of sagebrush canopy nitrogen, Remote Sensing of Environment, 124, 217-223, 2012.

Ollinger, S. V., Richardson, A. D., Martin, M. E., Hollinger, D. Y., Frolking, S. E., Reich, P. B., Plourde, L. C., Katul, G. G., Munger, J. W., Oren, R., Smith, M. L., Paw U, K. T., Bolsta, P. V., Cook, B. D., Day, M. C., Martin, T. A., Monson, R. K., and Schmid, H. P.: Canopy nitrogen, carbon assimilation, and albedo in temperate and boreal forests: Functional relations and potential climate feedbacks, Proceedings of the National Academy of Sciences of the United States of America, 105, 19336-19341, doi:10.1073/pnas.0810021105., 2008.

Ollinger, S. V., Reich, P. B., Frolking, S., Lepine, L. C., Hollinger, D. Y., and Richardson, A. D.: Nitrogen cycling, forest canopy reflectance, and emergent properties of ecosystems, Proceedings of the National Academy of Sciences, 110, E2437, 10.1073/pnas.1304176110, 2013.

Pacheco-Labrador, J., González-Cascón, R., Pilar Martín, M., and Riaño, D.: Understanding the optical responses of leaf nitrogen in mediterranean holm oak (Quercus ilex) using field spectroscopy, International Journal of Applied Earth Observation and Geoinformation, 26, 105-118, doi:10.1016/j.jag.2013.05.013, 2014.

Sardans, J., Rivas-Ubach, A., and Peñuelas, J.: Factors affecting nutrient concentration and stoichiometry of forest trees in Catalonia (NE Spain), Forest Ecology and Management, 262, 2024-2034, doi:10.1016/j.foreco.2011.08.019, 2011.

Sardans, J., Alonso, R., Carnicer, J., Fernández-Martínez, M., Vivanco, M. G., and Peñuelas, J.: Factors influencing the foliar elemental composition and stoichiometry in forest trees in Spain, Perspectives in Plant Ecology, Evolution and Systematics, 18, 52-69, 10.1016/j.ppees.2016.01.001, 2016.

Schlemmer, M., Gitelson, A., Schepers, J., Ferguson, R., Peng, Y., Shanahan, J., and Rundquist, D.: Remote estimation of nitrogen and chlorophyll contents in maize at leaf and canopy levels, International Journal of Applied Earth Observation and Geoinformation, 25, 47-54, doi:10.1016/j.jag.2013.04.003, 2013.

Townsend, P. A., Serbin, S. P., Kruger, E. L., and Gamon, J. A.: Disentangling the contribution of biological and physical properties of leaves and canopies in imaging spectroscopy data, Proceedings of the National Academy of Sciences of the United States of America, 110, 10.1073/pnas.1300952110, 2013.

Vilà-Cabrera, A., Coll, L., Martínez-Vilalta, J., and Retana, J.: Forest management for adaptation to climate change in the Mediterranean basin: A synthesis of evidence, Forest Ecology and Management, 407, 16-22, 10.1016/j.foreco.2017.10.021, 2018.

Wang, Z., Wang, T., Darvishzadeh, R., Skidmore, A. K., Jones, S., Suarez, L., Woodgate, W., Heiden, U., Heurich, M., and Hearne, J.: Vegetation indices for mapping canopy foliar nitrogen in a mixed temperate forest, Remote Sensing, 8, doi:10.3390/rs8060491, 2016.

---

## Author Response (AR1)

*We would like to thank the reviewer for reviewing our manuscript.*

The manuscript shows an interesting study on the use of MERIS data to analyse empirical relationships between MTCI and ground measurements of forest canopy N content and concentration. Foliar N influences a variety of important ecosystem processes so it is clear the interest of exploring the capacity for remote detection of canopy N at regional scales from space-based platforms and the potential of new generation of sensors such as those included in the Copernicus program. However, direct estimation of N in fresh vegetation using remote sensing data is challenging due to its weak effect on leaf reflectance so the influence of structural properties of the canopy and other potential confounding factors related with the input data are key issues to be explored.

The paper is well written and also well-structured and the research questions addressed are relevant and clearly fall within the scope of Biogeosciences.

*Thank you for your nice comment.*

However, my main concern about this work is that, at some point, the paper could be read as a search for correlations without a thoughtful discussion on the different **confounding factors** that could potentially affect to the observed relationship between satellite and ground data and how these factors could impact the results.

*We understand your concern about this paper being a search for correlations and would like to stress that we did a directed search rather than a random search. Remote sensing of canopy N (especially handheld and airborne) has already been extensively investigated and vegetation indices based on the red-edge region, on which MTCI is based, have been repeatedly used (Schlemmer et al., 2013;Li et al., 2014;Cho et al., 2013;Clevers and Gitelson, 2013;Dash and Curran, 2004). This can be explained by the link between foliar nitrogen and chlorophyll content. This is mentioned in the Introduction part of the manuscript (Line 78 – 81):*

*Detection of foliage N status with vegetation indices is attributed to the strong link between foliar nitrogen and chlorophyll content (Schlemmer et al., 2013) and is often based on the NIR and red-edge region of the spectrum, hence similar to the ones used for chlorophyll detection (Filella and Penuelas, 1994;Dash and Curran, 2004;Clevers and Gitelson, 2013).*

*In this study, we want to extend on the existing analyses by including spaceborne remote sensing. In the introduction part of the manuscript, we gave more details about the complexity of the reflectance signal and we clarified why canopy structure was found to be a confounding factor for remote sensing of canopy N at canopy level (Line 82 – 93):*

*At canopy level, however, spectral reflectance is a complex function of vegetation cover, plant activity, water content, illumination angle, viewing angle and atmospheric composition (Kumar et al., 2006) and it is not straightforward to disentangle*

*the influence of nitrogen from the other contributions in the spectra. It is thus not clear how the relationships observed at the leaf level translate at the canopy level. The mechanisms possibly modifying the remote detection of foliage N status at the canopy scale are still not clearly understood (Ollinger, 2011). High correlation between canopy N and both NIR reflectance and albedo has been reported in boreal forests (Ollinger et al., 2008). However, the mechanism behind these findings is still controversial. Knyazikhin et al. (2013) argued that the observed correlation solely resulted from canopy structural differences between broad and needleleaf forests and was thus spurious. Other authors, although agreeing that canopy structure was a confounding factor to account for, stated that the NIR – canopy N relationship was not necessarily spurious and stemmed from an association between canopy N and structural traits (Ollinger et al., 2013;Townsend et al., 2013). Canopy traits are interrelated (Wright et al., 2004) and have been known to covary due to evolutionary convergence, as stated by Ollinger (2011).*

In the discussion part of the manuscript, Section 4.4 "Possible confounding factors of the MTCI canopy N relationship", we discussed the possible confounding factors that could affect the MTCI – canopy N relationship, including canopy structure, biomass, and climatic conditions (Line 392 – 411):

*The relationships between MTCI and both canopy N[%] and canopy N[area] were influenced by the PFT of the plots. The relationship between MTCI and canopy N[%] was stronger for DBF and mixed plots compared to EBF and ENF plots while the opposite was true for the MTCI-canopy N[area] relationship. In the ongoing discussion about the mechanisms underlying the remote detection of canopy N, some authors argued that the difference in structural properties between different PFTs was a confounding factor of the observed relationship between canopy N and remote sensing data, rendering it spurious (Knyazikhin et al., 2013). Other authors, , suggested that the role of canopy structure as confounding factor can be explained by an indirect association between canopy N and canopy structure resulting from convergent adaptive processes (Ollinger et al., 2013;Townsend et al., 2013). In this context, our analysis showed that the PFTs of the plots had an influence on the MTCI canopy N relationship in a specific type of ecosystem, namely Mediterranean forests. Other confounding factors associated with N availability that might affect the observed relationship possibly include biomass, biomass allocation, leaf area index (LAI), water availability, soil type, etc. The data from the forest inventory used in this analysis, i.e. the Catalonian National Forest Inventory, were extensively studied, showing that water availability was the most limiting factor in this region. Water availability was positively correlated with both the N[area] and N[%] in leaves, as well as with foliar and total above-ground biomass through MAP (Sardans, 2011, Sardans, 2013). The MAP also influenced the PFT distribution as DBF plots were located in wetter areas than EBF plots, which were found in wetter sites than ENF plots. Regarding the influence of PFT on the foliar biomass, DBF plots had on average 45% less foliar biomass than EBF or ENF plots (Sardans, 2013). This shows that canopy N[%] and canopy N[area] were interrelated to biomass, PFT and MAP.*

A key element in this study is related with the intrinsic limitations of the input data: spatial (1) and temporal mismatch (2) but also, for example, the method used to scale from leaf to canopy N using field sampling strategies (3).

1) The reviewer is correct, since the beginning of this project we were aware of the limitation of the dataset included. We chose to use the data from the Catalonia National Forest Inventory because it includes many plots that are well spread over the forested region of Catalonia. The spatial mismatch has been addressed (in our original and revised manuscripts) by resampling both the MTCI product and the canopy N ground measurements to the same and lower spatial resolution. Then, we analysed the relationship between both datasets, taking the spatial discrepancy into account. The results showed that the correlation between the resampled canopy N and MTCI were significant regardless of the resampled pixel size, as described in the result part section 3.2 "Relationship between MTCI and canopy N data at lower spatial resolution". Moreover, a previous analysis investigated the influence of the spatial resolution on the remote sensing of canopy nitrogen. They could show that, even though the proportion of explained variance was reduced by going from high spatial resolution product to a low spatial resolution one (500 m), it was still possible to observe significant relationship between coarse spatial resolution remote sensing data and ground measurements (Lepine et al., 2016). This is mentioned in the discussion part of the manuscript in Section 4.5 (Line 424 – 426).

2) We addressed (in our original and revised manuscript) the temporal mismatch by averaging the MTCI product by month over the 10 years acquisition period, and selecting only the summer months, i.e. May-October, which corresponds to the growing season. By doing this, we decrease the influence of annual anomaly on the results. Moreover, the different selection criteria applied on the dataset, ensured that the plots that had undergone a land cover change were removed from the analysis. The consequence of this is that among the 846 plots included in the analysis, 625 were measured between 2000 and 2001, i.e. closer in time to the MTCI acquisition period (2002 – 2012), as shown in Table 1 below. Additionally, in response to a later comment in your review report, an analysis of the inter-annual variation of canopy N data has been included in the revised manuscript (figure A1, p.4 of this rebuttal).

| Year | 1988 | 1989 | 1990 | 1991 | 1992 | 1993 | 1994 | 1995 | 2000 | 2001 |
|---|---|---|---|---|---|---|---|---|---|---|
| Number of plot measured | 8 | 47 | 46 | 35 | 44 | 29 | 9 | 3 | 304 | 321 |

**Table 1. Number of plots included in the analysis by sampling year.**

3) To scale from leaf to canopy N, we use the leaf N value averaged over three individual trees as the plot canopy N value. This methodology, i.e. using leaf N concentration averaged over several individuals as the plot level value, is common (Schlerf et al., 2010). In our study 96% of the plots were monospecific and 4% of the plots contained only two species, therefore we did not weight the average by the species abundance (Smith and Martin, 2001;Townsend et al., 2003;McNeil et al., 2007).

The paragraph about the leaf sampling method (Section 2.2.1 "Canopy N data") has been changed to stress that most of the plots were monospecific (Line 146 - 148):

*A proportion of 96% of the plots included in this analysis were monospecific (Sardans, 2015). 4% of the plots (n = 30) had two codominant species. For these plots, two leaf samples were collected, one for each of the codominant species found on the plots.*

We also explain how we scaled up from leaf to canopy level (Line 152 - 154):
*To scale from leaf to canopy level, we used the leaf nitrogen concentration averaged over three individuals as the plot level value (Schlerf, 2010). We did not weight the average by species abundance (Smith and Martin, 2001) as only 4% of the plots had two different species.*

In this work, allometric equations are used to relate the diameter of the branches to the leaves dry weight in order to estimate canopy N content. It would be interesting to discuss the accuracy of this method compared to others proposed in the literature to estimate canopy foliar mass per species at the stand level.

Thank you for your comment. When analysing this further, we found that the information provided in the original version of the manuscript about biomass calculation was incorrect. The foliar biomass data were calculated using allometric equations based on the diameter at breast height (DBH). The DBH was measured for all the trees present on the plot. This information is provided in two articles that also include data from the Catalonian National Forest Inventory (Vilà et al., 2003;Sardans et al., 2015).

The paragraph in the canopy N data Section 2.2.1 was changed accordingly (Line 155 - 158):

*Along with the canopy N[%] data, we used foliar biomass data (g m-2) acquired during the same forest inventory (n = 2286). The foliar biomass data were obtained for each plot from allometric equations relating the diameter at breast height to the leave dry weight. These allometric equations were species specific (Sardans et al. (2015), Table A1).*

The table with the allometric relationships was included in the Appendix Section (Table A 1):

**Table A 1. Allometric relationships between foliar biomass and DBH for the different species included in this analysis. DBH = Diameter at breast height (cm).**

| Species | Foliar biomass $= a \cdot DBH^b$ | | | |
|---|---|---|---|---|
| | a | b | n | $r^2$ |
| *Castanea sativa* | 0.032 | 1.669 | 86 | 0.49 |
| *Fagus sylvatica* | 0.026 | 1.546 | 285 | 0.66 |
| *Pinus halepensis* | 0.037 | 1.656 | 2420 | 0.65 |
| *Pinus nigra* | 0.022 | 1.870 | 1641 | 0.65 |
| *Pinus pinaster* | 0.034 | 1.848 | 169 | 0.67 |
| *Pinus pinea* | 0.014 | 2.029 | 335 | 0.72 |
| *Pinus sylvestris* | 0.036 | 1.651 | 2755 | 0.66 |
| *Pinus uncinata* | 0.087 | 1.410 | 770 | 0.62 |
| *Quercus canariensis* | 0.120 | 1.322 | 36 | 0.57 |
| *Quercus faginea* | 0.197 | 0.943 | 170 | 0.40 |
| *Quercus humilis* | 0.047 | 1.462 | 595 | 0.59 |
| *Quercus cerrioides* | 0.023 | 1.805 | 138 | 0.73 |
| *Quercus ilex* | 0.063 | 1.576 | 2151 | 0.60 |
| *Quercus petraea* | 0.014 | 1.888 | 121 | 0.73 |
| *Quercus suber* | 0.026 | 1.446 | 314 | 0.55 |

It would be also interesting to know what is the inter-annual variation of N (ground measurements) in the study region in order to evaluate how this can affect to the discrepancy between timing of ground and satellite data.

We agree with you, the inter-annual variation of the ground measurements of canopy N is indeed essential due to the temporal discrepancy between our two datasets. As we have a large datasets covering the complete sampling period, studying the inter-annual evolution of the canopy N ground measurements was possible. We have mentioned this analysis in the revised manuscript (Line 222 – 224):

*The inter-annual variation of canopy $N_{[\%]}$ data was analysed for each month included in the analysis to ensure that the ground data could be related with MTCI data (Figure A 1).*

We have included the inter-annual graph in the Appendix part of the manuscript:

[Figure]

*Figure A 1. Inter-annual variation of canopy N[%] (%N) for each month included in the analysis. The numbers 6 – 10 (right side of the figure, row numbers) refer to the month of June, July, August, September and October, respectively. DBF = Deciduous Broadleaf Forest, EBF = Evergreen Broadleaf Forest, ENF = Evergreen Needleleaf Forest, mixed = mixed forest. Each point represents an observation at a forest plot. Note that the forest plots were not sampled multiple times, hence the inter-annual variation encompasses both temporal variation and spatial variation.*

It is important to note that the forest plots were not sampled multiple times. Hence, this inter-annual variation encompasses both temporal and spatial variation and it is not possible to distinguish temporal variation alone. Figure A1 shows that trends in canopy N values over the years are rather small relative to the variation of canopy N values within a year, which includes both spatial and in between month variation. Thus, the temporal discrepancy between both datasets as well as averaging MTCI values over 10 years may have influenced the found correlations between MTCI and canopy N, but we expect the effect to be limited.

Another important issue in this work is the lack of assessment of robustness of empirical models applied using either independent data or statistical techniques (bootstrap). This may be critical when the relationships found could depend on the covariance with other variables as is typically the case in the canopy N estimation from remote sensing.

Thank you, we agree with your comment. In order to assess the robustness of the relationships between MTCI and canopy N, a leave-one-out cross validation was calculated for each of the relationships presented in the analysis. This yielded a Relative Root Mean Square Error (RRMSEcv) value giving information about the prediction error of these relationships.

The calculation of the RRMSEcv was included in the material and method part, Section 2.3.4 Statistical analysis (Line 258 – 263):

*We calculated the Relative Root Mean Square Error of cross-validation (RRMSEcv, %) using the leave-one-out cross validation method (Clevers and Gitelson, 2013). Its calculation is presented in Eq. (3) following (Yao et al., 2010):*

$$RRMSEcv = \sqrt{\frac{1}{n} \times \sum_{i=1}^{n} (P_i - O_i)^2} \times \frac{100}{\overline{O_t}} \qquad (3)$$

*where $P_i$ represents the predicted value, $O_i$, the observed value, $\overline{O_i}$ the mean of all observed value and n the total number of measurement.*

The results are showed in the Table 3 and 4 of the revised manuscript, Section 3.3.1 and 3.3.2:

*Table 2. Observed linear regression equations between the MERIS Terrestrial Chlorophyll Index (MTCI) (-) and canopy nitrogen concentration ($CN_{[\%]}$, %N) for different subgroups. Number of plots (n), determination coefficient ($r^2$), p-value and Relative Root Mean Square Error of cross-validation (RRMEcv). PFT = Plant Functional type, DBF = Deciduous Broadleaf Forest, EBF = Evergreen Broadleaf Forest, ENF = Evergreen Needleleaf Forest, mixed = mixed forest.*

| group | n | linear regression | 95% confidence interval intercept | 95% confidence interval slope | $r^2$ | p-value | RRMSEcv |
|---|---|---|---|---|---|---|---|
| overall | 846 | MTCI = 2.18 + 0.79 log($CN_{[\%]}$) | [2.15, 2.20] | [0.71, 0.87] | 0.32 | < 0.000 | 17.0 |
| DBF | 80 | MTCI = 2.07 + 0.95 log($CN_{[\%]}$) | [1.78, 2.36] | [0.59, 1.32] | 0.25 | < 0.000 | 12.7 |
| EBF | 186 | MTCI = 2.39 + 0.29 log($CN_{[\%]}$) | [2.31, 2.48] | [0.04, 0.54] | 0.03 | 0.021 | 12.4 |
| ENF | 564 | MTCI = 2.13 + 0.61 log($CN_{[\%]}$) | [2.10, 2.17] | [0.46, 0.76] | 0.10 | < 0.000 | 19.2 |
| mixed | 16 | MTCI = 2.05 + 1.35 log($CN_{[\%]}$) | [1.63, 2.46] | [0.53, 2.17] | 0.47 | 0.003 | 12.4 |

*Table 3. Observed linear regressions equations between the MERIS Terrestrial Chlorophyll Index (MTCI) (-) and canopy nitrogen content ($CN_{[area]}$, g $m^{-2}$) for different subgroups. Number of plots (n), determination coefficient ($r^2$), p-value and Relative Root Mean Square Error of cross-validation (RRMEcv). PFT = Plant Functional type, DBF = Deciduous Broadleaf Forest, EBF = Evergreen Broadleaf Forest, ENF = Evergreen Needleleaf Forest, mixed = mixed forest.*

| group | n | linear regression | 95% confidence interval intercept | 95% confidence interval slope | $r^2$ | p-value | RRMSEcv |
|---|---|---|---|---|---|---|---|
| Overall | 841 | MTCI = 2.08 + 0.20 log($CN_{[area]}$) | [2.04, 2.12] | [0.17, 0.23] | 0.17 | <0.000 | 18.7 |
| DBF | 80 | MTCI = 2.72 + 0.06 log($CN_{[area]}$) | [2.58, 2.87] | [-0.04, 0.15] | 0.02 | 0.263 | 14.7 |
| EBF | 186 | MTCI = 2.39 + 0.07 log($CN_{[area]}$) | [2.32, 2.46] | [0.02, 0.12] | 0.04 | 0.005 | 12.4 |
| ENF | 563 | MTCI = 1.94 + 0.20 log($CN_{[area]}$) | [1.91, 1.99] | [0.17, 0.24] | 0.2 | <0.000 | 18.2 |
| mixed | 12 | MTCI = 2.43 + 0.34 log($CN_{[area]}$) | [2.05, 2.82] | [-0.26, 0.95] | 0.14 | 0.236 | 12.8 |

We also mentioned the RRMSEcv results in the text of the Section 3.3.1 (Line 300 – 301):

*The results showed that the linear regression between MTCI and canopy N[%] for the whole dataset (n = 846) was highly significant (p<0.000) and had an $r^2$ value of 0.32 and a RRMSEcv value of 18.7 % (Table 3).*

Finally, I also miss in the discussion how the authors consider the results could be potentially useful for monitoring canopy N at regional scale considering the strength of the relationships found and the estimation errors (not analyzed in the paper).

Thank you for your comment. The goal of this case-study analysis was to explore the feasibility of canopy N detection at regional scale using MTCI. Although the relationships are modest, our study contributes to the ongoing discussion about how

to remotely sense canopy N over larger area, which could also lead to canopy N monitoring possibilities. This has been explained in the discussion part of the manuscript, Section 4.5 "Perspectives for future applications" (Line 427 – 441):

*In this context, the methodology applied in this article could be a valuable alternative to explore canopy N detection at larger scale. Using published data from an extensive field plot inventory, we were able to relate both canopy $N_{[\%]}$ and canopy $N_{[area]}$ to MTCI at different spatial resolutions. Although the relationships found were modest, our study contributes to the ongoing discussion about how to remotely sense canopy N over larger area. As MTCI time series are readily and almost globally available, it could eventually be possible to assess our approach at a broader scale in different types of biomes. The results obtained for DBF species and Fagus sylvatica in particular suggest that this method may be efficient at estimating canopy N in temperate forests. If the strength of the relationship between MTCI and canopy N can further be improved, this could lead to canopy N monitoring possibilities at regional scale. In this context, the new sensors OLCI, onboard Sentinel 3 satellite, and especially MSI, onboard Sentinel 2 satellite might be promising due to their higher spatial resolution, from 10 to 60 m for Sentinel 2. They have bands well positioned to compute the MTCI vegetation index. Although the OLCI Terrestrial Chlorophyll Index (OTCI), the successor of the MTCI for the OLCI sensor, is already included in the OLCI level 2b reflectance image, no level 3 product (mosaicked over larger areas and temporally averaged hence similar to the MTCI time series used in this analysis) is available yet. In addition to more detailed remote sensing data, adding additional ground based canopy N observations to the regression models could better constrain these models as well. It would in particular be promising to use canopy N data over larger areas and for more diverse and globally distributed vegetation types.*

Regarding your suggestion to include estimation errors, we have calculated the prediction intervals for the MTCI – canopy N relationships. These are shown in the scatterplots Figure 5 and Figure 6 of the revised manuscript:

[Figure]

*Figure 5. Scatterplot and linear regression line between the MERIS Terrestrial Chlorophyll Index (MTCI) (-) and canopy nitrogen (N) concentration (canopy $N_{[\%]}$, %N) for (a) whole dataset (n = 846); (b) Deciduous Broadleaf Forest plots (DBF, n = 80); (c) Evergreen Broadleaf Forest plots (EBF, n = 186); (d) Evergreen Needleleaf Forest plots (ENF, n = 564); (e) mixed forest plots (n = 16). PFT = Plant functional type. The grey shading represents the prediction intervals (95 %) . Canopy $N_{[\%]}$ variable was log transformed to fulfil linear model assumptions.*

[Figure]

*Figure 6. Scatterplot and linear regression line between the MERIS Terrestrial Chlorophyll Index (MTCI) (-) and canopy N content (canopy N[area], g m-2) for (a) whole dataset (n = 841); (b) Deciduous Broadleaf Forest plots (DBF, n = 80); (c) Evergreen Broadleaf Forest plots (EBF, n = 186); (d) Evergreen Needleleaf Forest plots (ENF, n = 563); (e) mixed forest plots (n = 12). PFT = Plant functional type. The grey shading represents the prediction intervals (95 %). Canopy N[area] variable was log transformed to fulfil linear models assumptions.*

Specific comments addressing particular scientific/technical/formal issues follow:

Page 5 line 139. Complementary o alternative reference on methodology applied?

*The explanation on the allometric relationship has been changed (Line 155-):*

*Along with the canopy N[%] data, we used foliar biomass data (g m-2) acquired during the same forest inventory (n = 2286). The foliar biomass data were obtained for each plot from allometric equations relating the diameter at breast height to the leaves dry weight. The allometric equations were species specific (Sardans et al. (2015), Table A 1).*

Page 5 line 143. Correct : : :.foliar biomass (N g per square meter: : :.

*This has been changed in the text (Line 162):*
*foliar biomass (dry matter g per square meter of ground area, g m$^{-2}$)*

Page 5 line 153. Reword to clarify content and avoid repetitions

*The sentence was clarified in the text (Line 146 – 148):*
*A proportion of 96% of the plots included in this analysis were monospecific (Sardans et al., 2011). 4% of the plots (n = 30) had two codominant species. For these plots, two leaf samples were collected, one for each of the codominant species found on the plots.*

Page 6 line 180. Why the MERIS 300m full resolution product was not used instead?

Thank you for your question. We indeed first looked at using the MERIS 300 m full resolution reflectance images. These images were not used for our analysis for several reasons. The 300 m full resolution reflectance images available from the ESA are

not corrected for cloud cover and atmospheric influences. Moreover, there is no temporally averaged product available at full resolution. This means that one image of the 300 m full resolution reflectance data is available every three to four days from 2002 until 2012. Each of the images included in this analysis would thus need to be atmospherically corrected (365/4 *10 ~ 912 images). This would have been very time intensive.

In this context, the MTCI 1 km level 3 product presented several advantages. It is a readily usable product that has been corrected for atmospheric influences and cloud cover and was monthly averaged. The availability of the MTCI monthly product made it possible for us to relate the ground canopy N measurements to 10 years monthly averaged without involving time consuming images processing. We believe that this way we could decrease part of the uncertainty of relating ground measurement to any daily remotely sensed reflectance value measured several years later. Finally, the MTCI product is available for the extent of the Catalonia region in one single image, while the MERIS full resolution product can sometimes only partly cover the region and therefore each image would have had to be selected manually.

In the revised manuscript, this is more clearly explained in the Material and Methods part, Section 2.2.2 "MTCI product" of the revised manuscript (Line 200 - 211).

*MTCI level 3 imagery was obtained from the NERC Earth Observation Data Centre (NEODC, 2015) for the region of Catalonia between 2002 and 2012. The original data were provided by the European Space Agency and then processed by Airbus Defence and Space. The original MERIS reflectance images, following ENVISAT specifications, have a revisit time of three days and a spatial resolution of 300 m. Compared to the original reflectance images, the MTCI processed imagery has been corrected for atmospheric influences and cloud cover (Curran and Dash, 2005) and is available as an either weekly or monthly averaged product. The spatial resolution of the processed data is approximately 1 km. As there is no temporally averaged product available at full resolution, we chose to carry out this analysis with the MTCI monthly averaged imagery. This was done to decrease the uncertainty resulting from the use of single daily reflectance values. An MTCI 10 time series of 10 years is available almost globally. One MTCI monthly averaged imagery product covering the entire study area was obtained for every month between June 2002 and March 2012, except for October 2003, when no valid product was available.*

Page 7 lines 197-199. What about other land cover changes as those caused by forest fires (quite frequent in the study region), where they investigated and filtered?

The land cover changes caused by forest fires were not investigated in a separate way. As Globcover 2009 the land cover map includes a sparse vegetation class, which we believe is how the vegetation appears after a forest fire, the change due to forest fire should be accounted for when excluding sparse vegetation class from the analysis.

Page 7 Sections 2.3.2 and 2.3.3. Would be interesting to know the number of plots per pixel (average, min and max) at the different spatial resolutions.

Thank you, we agree. The number of plots per resampled pixel size are shown in the table 2 of the revised manuscript in the Result part Section 3.2 "Relationship between MTCI and canopy N data at lower spatial resolution" (Line 293 – 296):

*Table 2 shows the number of plots per pixel for different pixel sizes (km). As expected, the number of plots per pixel increased with the pixel size, with a mean of 4.1 plots at 20 km spatial resolution. The descriptive statistics of the number of different PFT, species and sampling years per pixel spatial resolution are provided in the Appendix (Table A2 – A4).*

*Table 4. Descriptive statistics of the number of plots per pixel, for different spatial resolutions (km, pixel length). min = minimum, max = maximum, mean = average, sd = standard deviation.*

| Spatial resolution (km) | Number of plots per pixel | | | |
| --- | --- | --- | --- | --- |
| | min | max | mean | sd |

| | | | | |
|---|---|---|---|---|
| 5 | 1 | 6 | 1.44 | 0.77 |
| 10 | 1 | 11 | 2.19 | 1.53 |
| 15 | 1 | 15 | 3.11 | 2.59 |
| 20 | 1 | 22 | 4.09 | 3.74 |

Page 8 line 238. Foliar biomass is used in the calculation of canopy N content so the correlation is obviously strongest

Thank you, we agree with your comment. This was not intended to be understood as a new finding but we rather wished to be fully explicit about the correlation between the variables. The original sentence was replaced by (Line 277 - 279):

*The correlation between each pair of variables was significant and the correlation between canopy $N_{[area]}$ and foliar biomass was strongest (r = 0.88). This result was expected as the foliar biomass was included in the $N_{[area]}$ calculation.*

Page 9 line 254. Higher instead of lower
This has been changed.

Page 9 line 269. R2 for Quercus ilex?
The r2 value for *Quercus ilex* plots has been added in the text (Line 315):

*The relationship between MTCI and canopy N[area] was also investigated for 10 individual species and one of them showed significant relationships: Quercus ilex (r2 = 0.10, n = 160).*

Page 9 Section 4.1. Could the authors elaborate here on how this could affect to the regional estimation of canopy N using new generation Sentinel-2 and 3 with improved spatial resolutions?

Due to the higher spatial resolution of the MSI sensor onboard Sentinel 2 and the bands well positioned in the red edge region, remote sensing of canopy N at regional scale might be promising. However, a pre-processed product similar to the MTCI time series should first be made available to reproduce the methodology applied in this study. This has been addressed in the discussion Section 4.5 "Perspective for larger scale applications" (Line 434-439):

*In this context, the new sensors OLCI, onboard Sentinel 3 satellite, and especially MSI, onboard Sentinel 2 satellite, might also be promising due to their higher spatial resolution, from 10 to 60 m for Sentinel 2. They have bands well positioned to compute the MTCI vegetation index. Although the OLCI Terrestrial Chlorophyll Index (OTCI), the successor of the MTCI for the OLCI sensor, is already included in the OLCI level 2b reflectance image, no level 3 product similar to the MTCI time series used in this analysis, i.e. mosaicked over larger areas and temporally averaged, is available yet.*

Page 11 line 315. Any hypothesis on the stronger relationship found for DBF plots? Further investigation on the proportion of the variance explained by other potential confounding factors would be desirable (same in lines 329 and 341)

The effects of the confounding factors, among which canopy structure and the differences between PFTS, on the MTCI-canopy N relationship was addressed in general in an extensive comment in the Section 4.4 "Possible confounding factors of the MTCI canopy N relationship" (Line 391 – 411):

*4.4 Possible confounding factors of the MTCI canopy N relationship*

*The relationships between MTCI and both canopy N[%] and canopy N[area] were influenced by the PFT of the plots. The relationship between MTCI and canopy N[%] was stronger for DBF and mixed plots compared to EBF and ENF plots while*

*the opposite was true for the MTCI-canopy N[area] relationship. In the ongoing discussion about the mechanisms underlying the remote detection of canopy N, some authors argued that the difference in structural properties between different PFTs was a confounding factor of the observed relationship between canopy N and remote sensing data, rendering it spurious (Knyazikhin et al., 2013). Other authors, , suggested that the role of canopy structure as confounding factor can be explained by an indirect association between canopy N and canopy structure resulting from convergent adaptive processes (Ollinger et al., 2013;Townsend et al., 2013). In this context, our analysis showed that the PFTs of the plots had an influence on the MTCI canopy N relationship in a specific type of ecosystem, namely Mediterranean forests. Other confounding factors associated with N availability that might affect the observed relationship possibly include biomass, biomass allocation, leaf area index (LAI), water availability, soil type, etc. The data from the forest inventory used in this analysis, i.e. the Catalonian National Forest Inventory, were extensively studied, showing that water availability was the most limiting factor in this region. Water availability was positively correlated with both the N[area] and N[%] in leaves, as well as with foliar and total above-ground biomass through MAP (Sardans, 2011, Sardans, 2013). The MAP also influenced the PFT distribution as DBF plots were located in wetter areas than EBF plots, which were found in wetter sites than ENF plots. Regarding the influence of PFT on the foliar biomass, DBF plots had on average 45% less foliar biomass than EBF or ENF plots (Sardans, 2013). This shows that canopy N[%] and canopy N[area] were interrelated to biomass, PFT and MAP.*

Page 11 lines 332-335. This has been already stated in the results Sections. This apply for other paragraphs in this Section, authors should avoid to repeat the results and focus on the discussion.

Thank you, we agree that repeating this information several times might be unnecessary. In this instance, we wanted to remind the reader what we are going to address in the next paragraph.

Page 12 lines 152-153. I would recommend to include the analysis in this paper using information acquired in the forest inventory used for the study.

Thank you for your comment. We agree that using additional data besides canopy N and foliar biomass would make the analysis stronger. However, only biomass and foliar concentration was measured during the forest inventory. Additional physiological data related to the forest plots is thus not available.

Page 20 FIGURE 1. Please clarify if the plots represented in the map are all the forest inventory plots (2300?) or 1075 (after temporal and spatial filtering) or 846/841 finally used in the analysis. I would recommend including only the plots used in the analysis.

Thank you, the number of plots represented in the figure (n = 846) has been added to the figure caption (Line 664). The number "1075" plots was mistake from a former version of the manuscript and has been changed where it appeared in the text.

*Figure 1. Map showing the forest plots (n = 846) location in the region of Catalonia, north eastern Spain. DBF = Deciduous Broadleaf Forest, EBF = Evergreen Broadleaf Forest, ENF = Evergreen Needleleaf Forest, mixed = mixed forest.*

General comments: The paper aims to investigate the potential of using MTCI to map regional variations in canopy nitrogen (N). The study uses field measurements of canopy N for a large number of forest plots situated across Catalonia to derive empirical relationships between N and MTCI data across a range of spatial resolutions (1 - 20 km). The study also aims to identify the influence of plant functional type on the observed relationships. Whilst the premise of the work may be interesting, there are a number of questions and comments, some of which are fairly fundamental, which I feel need addressing before this manuscript can be considered for publication. The comments are provided in the hope that they may help improve the manuscript and its subsequent impact.

Specific comments: I am not entirely convinced of the justification for reducing the spatial resolution of the MTCI data. Why degrade the 1 km product? The MERIS sensor on board ENVISAT is no longer operational (which the author's should note). The authors do note that a variation of the MTCI can be calculated from Sentinel-2 but this is a sensor with a higher spatial resolution then MERIS so what is the justification for making the data worse? Especially since the forest plots were substantially smaller than the original 1 km pixel size in the first instance. Averaging 6 m plots over a 1 km grid would "reduce small-scale variations (line 279)" so why 5, 10, 15 and 20 km also? Without this information the paper appears to be more of an academic exercise as opposed to addressing a tangible issue.

We are thankful that the reviewer finds the premise of the work interesting, and would like to use the opportunity to clarify some of the steps we took in the paper.
We indeed studied the relationship between canopy N and MTCI at different spatial resolutions. In our analysis, we used two datasets: MTCI, which has an original spatial resolution of 1 km, and canopy N ground measurements measured on 6 m diameter forest plots. In a first step, we looked at the relationship between MTCI and canopy N after resampling both datasets to the same, lower spatial resolutions, i.e. 5, 10, 15 and 20 km, removing small scale variability as values were averaged over the larger, new, resolution. The reason for doing this was to overcome the initial spatial discrepancy between our two datasets and to study the relationship between the two variables independently of the initial spatial discrepancy. This step also allowed us to study the influence of the spatial resolution on the relationship between MTCI and canopy N. The results as shown in Section 3.2 "Relationship between MTCI and canopy N data at lower spatial resolution" showed that the relationships between the variables was not strongly affected by the resampling factor. In a second step, we analysed the relationship between MTCI and canopy N at the original spatial resolution, i.e. we related each 6m plot to a 1 km MTCI pixel. During this step, we could analyse the MTCI – canopy N relationship in more detail by looking at the influence of the different PFTs on the relationship. However, during this step, the difference between MTCI and canopy N measurements in spatial resolution was retained and small-scale variation might have had an influence on the found relationship.

To make this clearer we modified the Introduction of the revised manuscript, Section 1 (Line 119 – 123), where we added the resampled pixel size:

*Next, both data sets are resampled to the same, lower, spatial resolutions, i.e. 5 km, 10 k, 15 km and 20 km, in order to overcome the initial spatial discrepancy between MTCI spatial resolution (1 km) and the size of the forest plots (6 m). Subsequently, we analyse the relationship between MTCI and both canopy N concentration and canopy N content variables, both at the resampled and initial spatial resolutions. The relationships at the initial spatial resolution are then stratified according to the PFT of the plots.*

We adjusted the text to better clarify the steps we took. In Section 4.1, we say that averaging multiple 6 m plots over a lower spatial resolution, i.e. 5, 10, 15 and 20 km (and not 1 plot in a 1 km pixel), would reduce small-scale variations. To make this more clear, we added the resampled pixel size in the text in section 4.1 (Line 324 - 330):

*By resampling both datasets to a lower spatial resolution, i.e. 5 km, 10 km, 15 km and 20 km, the obtained values were less impacted by small-scale variations because they were obtained by averaging several values over a larger area. The results showed that the relationship between MTCI and canopy N data was significant and consistent across all spatial resolutions investigated: 5 km, 10 km, 15 km and 20 km. This showed that, when the spatial discrepancy between the original datasets, i.e. 6 m and 1 km, was taken into account, MTCI and canopy N data were linked and that the MTCI-canopy N relationship was not strongly affected by the resampled spatial resolution.*

We also more clearly stated the resampled spatial resolutions in the Material and Methods part, Section 2.3.2 "Relationship between MTCI and canopy N data at lower spatial resolution" (Line 233 – 236):

*In a first step, the relationships between MTCI and canopy N data values were investigated after resampling both datasets to the same, lower, spatial resolution. The resampled spatial resolutions were 5 km, 10 km, 15 km, and 20 km. This was done because of the initial difference in support size between MTCI spatial resolution and the forest plots size (i.e. 1 km and 6 m, respectively).*

Finally, we agree with you that the fact that the MERIS sensor came to an end in 2012 is essential information linked to our analysis. This was also noted in the original version of the manuscript (Line 192):
*While the ESA ENVISAT satellite mission producing MERIS data came to an end in 2012*

One of the main justifications for the study is that "limited research has been conducted to sense canopy N in Mediterranean ecosystems and even more so in Mediterranean forests", yet there is no discussion of the importance of these ecosystems, or their N content. More information should be included to justify the significance of this sentence.

We have include justification about the ecological importance, especially regarding species diversity (Vilà-Cabrera et al., 2018), of Mediterranean forests in the introduction part of the manuscript (Line 99 – 102):

*Moreover, Mediterranean forests have specific functional characteristic due to their great forest ecosystems diversity, influenced by contrasting climatic and topographic conditions, and their high tree species richness (Vilà-Cabrera et al., 2018). However, to our knowledge, limited research has been conducted to sense canopy N in Mediterranean ecosystems (Serrano et al., 2002) and even more so in Mediterranean forests.*

More information is required on how the forest plot data are deemed suitable for comparison
with the MTCI data. There are several questions here:

1. Is the year of data collection an issue for the correlation? Perhaps colouring the points in figure 4 based on year of in situ collection may be useful e.g. were there any climatically anomalous years that could have influenced the MTCI relationships?

Thank you for your suggestion. In the figure 4, measurements at the vegetation plots were averaged by pixel (20 km) and sampling month (over 10 years). This means that within a particular (resampled) 20 km pixel, the plots were measured in the same month. However, these plots might have been measured in different years. The Material and Methods section 2.3.2 " Relationship between MTCI and canopy N data at lower spatial resolution" (Line 243) has been edited to make this more clear.

Moreover, we agree with you the year of sampling might have an influence on the results given the temporal discrepancy between our two datasets. As we have a large datasets covering the complete sampling period, studying the inter-annual evolution of the canopy N ground measurements was possible. We have mentioned this analysis in the revised manuscript (Line 222 – 224):

*The inter-annual variation of canopy $N_{[\%]}$ data was analysed for each month included in the analysis to ensure that the ground data could be related with MTCI data (Figure A 1).*

We have included the inter-annual graph in the Appendix part of the revised manuscript:

[Figure]

*Figure A1. Inter-annual variation of canopy $N_{[\%]}$ (%N) for each month included in the analysis. The numbers 6 – 10 (right side of the figure, row numbers) refer to the month of June, July, August, September and October, respectively. DBF = Deciduous Broadleaf Forest, EBF = Evergreen Broadleaf Forest, ENF = Evergreen Needleleaf Forest, mixed = mixed forest. Each point represents an observation at a forest plot. Note that the forest plots were not sampled multiple times, hence the inter-annual variation encompasses both temporal variation and spatial variation.*

It is important to note that the forest plots were not sampled multiple times. Hence, the inter-annual variation shown in Figure A1 encompasses both temporal and spatial variation and it is not possible to distinguish temporal variation alone. Figure A1 shows that trends in canopy N values over the years are rather small relative to the variation of canopy N values within a year, which includes both spatial and in between month variation. Thus, the temporal discrepancy between both datasets as well

as averaging MTCI values over 10 years may have influenced the found correlations between MTCI and canopy N, but we expect the effect to be limited.

2. How well do the 6m forest data plots represent the 1, 5, 15 and 20 km grid scales? There isn't any information as to how many points were included in each grid square when the data were resampled at each resolution. What was the distribution of values (mean, SD)?

Thank you for your comment. This is indeed important if we wish to evaluate the effectiveness of the resampling method to overcome the initial difference in support size. A table showing the descriptive statistics (min, max, mean and sd) of the number of plots per pixel, has been added to the manuscript in the Results part, section 3.2 "Relationship between MTCI and canopy N data at lower spatial resolution" (Line 293 - 296):

*Table 2 shows the number of plots per pixel resampled spatial resolution (km). As expected, the number of plots per pixel spatial resolution increased with the pixel spatial resolution, with a maximum of 22 plots and a mean of 4.1 plots at 20 km spatial resolution.*

*Table 2. Descriptive statistics of the number of plots per pixel, by pixel spatial resolution (km). min = minimum, max = maximum, mean = average, sd = standard deviation.*

| Spatial resolution (km) | Number of plots per pixel | | | |
|---|---|---|---|---|
| | min | max | mean | sd |
| 5 | 1 | 6 | 1.44 | 0.77 |
| 10 | 1 | 11 | 2.19 | 1.53 |
| 15 | 1 | 15 | 3.11 | 2.59 |
| 20 | 1 | 22 | 4.09 | 3.74 |

Even though the number of plots per pixel may be low, we obtained significant correlations for all spatial resolutions studied. Please note that we did not resample to 1 km, as 1 km is the original spatial resolution of the MTCI product.

3. Can homogenous species plots be observed from satellite imagery at 5 – 20 km resolution? Surely the plots are going to be mixed species at this scale?

Indeed, the plots are likely to be mixed species and mixed PFT too. This is the reason why the analysis by species and PFT was not carried out at this step of the analysis (where we resample both MTCI and plot measurements to the same resolution of 5-20 km), but only in the step where the original MTCI data were used (1 km resolution).

To show the mixing effect at the resampled resolutions, we have incorporated two tables showing the average number of different PFT and species per resampled pixel. These tables are included in the Appendix of the manuscript (Table A 2 and Table A 3 of the revised manuscript):

*Table A 2. Descriptive statistics of the number of plant functional types (PFT) per pixel, by pixel spatial resolution (km). min = minimum, max = maximum, mean = average, sd = standard deviation.*

| Spatial resolution (km) | Number of PFT per pixel | | | |
|---|---|---|---|---|
| | min | max | mean | sd |
| 5 | 1 | 3 | 1.08 | 0.29 |
| 10 | 1 | 4 | 1.22 | 0.48 |

| | | | | |
|---|---|---|---|---|
| 15 | 1 | 4 | 1.34 | 0.61 |
| 20 | 1 | 4 | 1.45 | 0.69 |

*Table A 3. Descriptive statistics of the number of species per pixel, by pixel spatial resolution (km). min = minimum, max = maximum, mean = average, sd = standard deviation.*

| Spatial resolution (km) | Number of species per pixel | | | |
|---|---|---|---|---|
| | min | max | mean | sd |
| 5 | 1 | 4 | 1.14 | 0.41 |
| 10 | 1 | 4 | 1.38 | 0.67 |
| 15 | 1 | 4 | 1.58 | 0.85 |
| 20 | 1 | 6 | 1.79 | 1.07 |

The results presented, whilst statistically significant have quite low r2 values, which indicates that the precision with which N can be predicted will be low, even though there is a statistically significant relationship between the two variables. The authors do not comment on this but I think they should as this has practical implications for their suggested approach.
It would be useful for the authors to suggest possible reasons why the reported statistically
significant regressions are only explain 20 - 30% of the variation at best.

Thank you for your comment. The obtained r2 are indeed low, between 0.10 and 0.40. We would like to stress that other studies report similar or sometimes lower r2, for analyses conducted at higher spatial resolution and in more controlled conditions (Cho et al., 2013;Wang et al., 2016). The obtained results were compared with existing literature in the section 4.2.1 "Canopy N concentration detection". This section was modified to stress the differences in spatial resolution (Line 335 - 344):

*The performance of the MTCI vegetation index to detect canopy N[%] in Mediterranean vegetation was similar to the results obtained from previous studies using spaceborne MTCI at higher spatial resolution. For example, using MTCI computed from the spaceborne RapidEye sensor at 5 m spatial resolution, it was possible to detect canopy N[%] in grassland savannah and sub-tropical forest with similar coefficients of determination, r2 = 0.35 and r2 = 0.52, respectively (Ramoelo et al., 2012;Cho et al., 2013). However, while there is a consensus regarding MTCI ability for in situ leaf or canopy N[%] detection in a variety of crops using handheld spectrometers (Tian et al., 2011;Li et al., 2014), there is no general agreement about MTCI ability for canopy N[%] detection across vegetation and sensor types at larger scales. For example, MTCI computed from airborne data at 3 m spatial resolution could not be related to canopy N[%] from a mixed temperate forest (Wang et al., 2016). In this context our finding brings new insight into MTCI N[%] sensing capabilities at a much coarser spatial resolution (1 km) compared to what has been done before.*
We agree with the reviewer that there is indeed considerable prediction uncertainty. In the revised manuscript we have added prediction intervals for the MTCI – canopy N relationships. These are shown in the scatterplots Figure 5 and Figure 6 of the revised manuscript:

[Figure]

*Figure 5. Scatterplot and linear regression line between the MERIS Terrestrial Chlorophyll Index (MTCI) (-) and canopy nitrogen (N) concentration (canopy N[%], %N) for (a) whole dataset (n = 846); (b) Deciduous Broadleaf Forest plots (DBF, n = 80); (c) Evergreen Broadleaf Forest plots (EBF, n = 186); (d) Evergreen Needleleaf Forest plots (ENF, n = 564); (e) mixed forest plots (n = 16). PFT = Plant functional type. The grey shading represents the prediction intervals (95 %) . Canopy N[%] variable was log transformed to fulfil linear model assumptions.*

[Figure]

*Figure 6. Scatterplot and linear regression line between the MERIS Terrestrial Chlorophyll Index (MTCI) (-) and canopy N content (canopy N[area], g m-2) for (a) whole dataset (n = 841); (b) Deciduous Broadleaf Forest plots (DBF, n = 80); (c) Evergreen Broadleaf Forest plots (EBF, n = 186); (d) Evergreen Needleleaf Forest plots (ENF, n = 563); (e) mixed forest plots (n = 12). PFT = Plant functional type. The grey shading represents the prediction intervals (95 %). Canopy N[area] variable was log transformed to fulfil linear models assumptions.*

The prediction intervals may be wide but still show that the MTCI has considerable information content for estimating canopy N. We would like to stress that the goal of this case-study analysis was to explore the feasibility of canopy N detection at regional scale using MTCI. Although the statistical relationships are modest, the results provide spatio-temporal indicators of canopy N and we believe that this analysis provides information that is valuable to the ongoing discussion about the feasibility of sensing canopy N over larger spatial extent. We expect that the statistical models can be improved in future research using both improved remote sensing data at higher spatial resolution and other plot data from different types of ecosystems. This has been mentioned in the discussion part of the manuscript, Section 4.5 "Perspectives for future applications" (Line 429 – 441):

*Although the relationships found were modest, our study contributes to the ongoing discussion about how to remotely sense canopy N over larger area. As MTCI time series are readily and almost globally available, it could eventually be possible to assess our approach at a broader scale in different types of biomes. The results obtained for DBF species and Fagus sylvatica in particular suggest that this method may be efficient at estimating canopy N in temperate forests. If the strength of the relationship between MTCI and canopy N can further be improved, this could lead to canopy N monitoring possibilities at regional scale. In this context, the new sensors OLCI, onboard Sentinel 3 satellite, and especially MSI, onboard Sentinel 2 satellite might be promising due to their higher spatial resolution, from 10 to 60 m for Sentinel 2. They have bands well positioned to compute the MTCI vegetation index. Although the OLCI Terrestrial Chlorophyll Index (OTCI), the successor of the MTCI for the OLCI sensor, is already included in the OLCI level 2b reflectance image, no level 3 product (mosaicked over larger areas and temporally averaged hence similar to the MTCI time series used in this analysis) is available yet. In addition to more detailed remote sensing data, adding additional ground based canopy N observations to the regression models could better constrain these models as well. It would in particular be promising to use canopy N data over larger scale areas and for more diverse and globally distributed vegetation types.*

The authors indicate that these r2 values are somewhat lower than MODIS so why not just use MODIS?
Thank you for your question. We did not use MODIS because our goal was to test the relationship for the MTCI vegetation index. Vegetation indices products available for MODIS are NDVI and EVI, which have showed lower correlation with canopy N compared to MTCI due to saturation problem at high N concentration (Schlemmer et al., 2013;Pacheco-Labrador et al., 2014).
Moreover, the study we referred to in the discussion (Line 424 - 426), did indeed get higher r2 using MODIS images. Their methodology was different as there was no temporal discrepancy between their ground measurements and the satellite image acquisition. They worked with 7 x 7 km MODIS tiles, while the MERIS MTCI level 3 product is available from the ESA for the extend of the whole region (and actually even Europe) in one single image.

Technical corrections:
The first sentence of the abstract is quite long. Consider fragmenting and re-wording to improve impact.

Thank you, the sentence has been changed (Line 10):
*Canopy nitrogen (N) concentration and content are linked to several vegetation processes at leaf and canopy levels. Therefore, canopy N concentration is a state variable in global vegetation models with coupled carbon (C) and N cycles.*

Line 12 and throughout – Data "is" should be changed to data "are" since data are Plural
This has been changed.

Line 13 etc. – The abstract should include some justification as to why the work is important. This could be more clearly explained in the abstract as opposed to simply saying x did this and we are doing that. The key question is why?
We added a sentence to specify the opportunity to use vegetation indices to detect canopy N data at larger scale (Line 14):

*Vegetation indices could be a valuable tool to detect canopy N concentration and canopy N content at larger scale.*

Line 32: Delete "," after processes This has been changed.
Line 36: Insert "," after (N g m-2) This has been changed.
Line 49: Delete "Currently". This has been changed.
Line 49: Insert "," either side of from and sensors This has been changed.
Line 48 – 52: This is a very long sentence. Consider fragmenting.
This has been changed (Line 49 – 52):
*Imaging spectrometry has proven efficient in improving N sensing capabilities at the local scale. Imaging spectrometry images are acquired from either airborne of spaceborne sensors and are analysed with different methods, including partial least squares regression (PLS), continuum removal, spectral unmixing or vegetation indices.*

Line 53: No need for a new paragraph. This has been changed.

Line 71: "were aimed" is an odd choice of words. Consider re-wording
This has been changed:
*most studies were carried out in agricultural crops using MTCI values computed from in situ hyperspectral reflectance data*

Line 74: Do the authors mean "a few studies" or "few studies"? It's not clear as no
references are referred to.
We mean here "few studies". A reference has been added (Line 76 – 78):
*Remote detection of foliage N status has been extensively studied at the leaf scale (Hansen and Schjoerring, 2003;Ferwerda et al., 2005;Li et al., 2014) and few studies have investigated the processes underlying the relationships between vegetation indices and foliar N (Pacheco-Labrador et al., 2014).*

Line 83: "stated that the NIR – canopy N relationship was not necessarily spurious
as plant traits have been known to covary along the leaf economic spectrum" This
statement needs further explanation. What is meant by the leaf economic spectrum?

We have modified this sentence to make it clearer (Line 89 – 93):
*Other authors, although agreeing that canopy structure was a confounding factor to account for, stated that the NIR – canopy N relationship was not necessarily spurious and stemmed from an association between canopy N and structural traits (Ollinger et al., 2013;Townsend et al., 2013). Canopy traits are interrelated (Wright et al., 2004) and have been known to covary due to evolutionary convergence, as stated by Ollinger (2011).*

We also have expended the previous paragraph to include more detailed explanations on the complexity of the reflectance signal (Line 82 – 85):

*At canopy level, however, spectral reflectance is a complex function of vegetation cover, plant activity, water content, illumination angle, viewing angle and atmospheric composition (Kumar et al., 2006) and it is not straightforward to disentangle the influence of nitrogen from the other contributions in the spectra. It is thus not clear how the relationships observed at the leaf level translate at the canopy level. The mechanisms possibly modifying the remote detection of foliage N status at the canopy scale are still not clearly understood (Ollinger, 2011).*

Line 89: "MTCI time series could be applied to estimate canopy N at a larger scale" Be
careful with the terms scale here. Do you mean over a larger spatial extent?
This has been changed (Line 98 - 99):
*Due to its almost global coverage, MTCI time series could be applied to estimate canopy N over a larger spatial extent.*

Line 106: Suggests that there are 1075 forest plots but line 123 suggest that there are 2300 and in line 2017 there are 846 plots. Were some removed from the sample?

Thank you for noticing this mistake. On Line 116, 846 plots should have been written in place of 1075. This has been changed. The 2300 plots (Line 137) refers to the original number of plots included in the forest inventory before applying the selection criteria explained in the Material and Methods section (Line 251 – 253).

Line 110: What are the re-sampled resolutions and what is the justification for this?

Thank you for your comment. The resampled resolution are now clearly indicated in the text (Line 119 - 121):
*Next, both data sets are resampled to the same, lower, spatial resolutions, i.e. 5 km, 10 k, 15 km and 20 km, in order to overcome the initial spatial discrepancy between MTCI spatial resolution (1 km) and the size of the forest plots (6 m).*

Line 117: duplicate word "create": This has been changed.

Line 150: "Several (up to two times) " does not make sense. Several suggests three or more. Consider re-wording.

Thank you for your comment. This has been changed (Line 146 – 148):
*A proportion of 96% of the plots included in this analysis were monospecific (Sardans et al., 2011). 4% of the plots (n = 30) had two codominant species. For these plots, two leaf samples were collected, one for each of the codominant species found on the plots.*

Line 200: MTCI was not re-sampled as the product was already a 1 km product.

We agree with you that the initial spatial resolution of the MTCI product is 1 km. In the manuscript this is called the "higher spatial resolution". However, in our study we first analyse the relationship between MTCI and canopy N data after resampling both datasets to a lower spatial resolution (section 2.3.2 Relationship between MTCI and canopy N data at lower spatial resolution", Line 231). This was done to overcome the initial resolution discrepancy between the two datasets. To make this more clear, the resampled spatial resolution was added (Line 232 - 237):

*2.3.2    Relationship between MTCI and canopy N data at lower spatial resolution*
*In a first step, the relationships between MTCI and canopy N data values were investigated after resampling both datasets to the same, lower, spatial resolution. The resampled spatial resolutions were 5 km, 10 km, 15 km, and 20 km. This was done because of the initial difference in support size between MTCI spatial resolution and the forest plots size (i.e. 1 km and 6 m, respectively). This enabled us to investigate the relationships between MTCI and canopy N data independently of differences in initial support size.*

Line 303-204: "This enabled us to investigate the relationships between MTCI and canopy N data independently of differences in initial support size." I don't entirely agree. Just because they now match on a spatial grid does not mean that the difference in sampling support size no longer matters. The crucial point is how well do the 6 m forest data represent the 1 km grid scale? Anything can be re-sampled. Whether it makes sense to do so is a different question.

Thank you for your comment. We chose to include the resampling analysis in our study due to the initial spatial discrepancy between the two datasets used, i.e. the ground measurements (6 m plots) and the MTCI pixel (1 km). By resampling both dataset to a lower and equal spatial resolution, i.e. 5 km, 10 km, 15 km, and 20 km, we wanted to study the relationship between the two variables when the spatial discrepancy was accounted for. The statistical basis of our approach is that we bring both sources of information (field sampling and MTCI values) to the same support size (representative area). By averaging out point samples (plot observations) within this support size, we calculate the expectation (mean) of the canopy N at that support size. By resampling the MTCI values to that same support size, the obtained result consist of an expectation

(mean) of the MTCI at that support size. We then regressed the expected canopy N values (at the new support size) against the expected MTCI values (at the new support size). As you mention it, this process is dependent on the spatial representability of the plots within the support size. This has been addressed on the page 3 and 4 of this response. We looked at the number of plots per resampled pixel, the number of different species and PFT per resampled pixel as well as the number of different sampling years per resampled pixel. The number of plots per resampled pixel may be low, but in our opinion, is still sufficient to calculate the value representative for the resampled pixel. In our opinion, this is the best we can do to make the data sets the same regarding support.

Section 2.3.3. It seems a bit odd to investigate relationships at a lower resolution before you investigate it at the original spatial resolution.

Thank you for your comment. We choose to study the relationship at the lower spatial resolution before because we wanted to explore the relationship at higher spatial resolution in more details, i.e. by also PFT and species into account. At lower spatial resolution, this information about PFT and species is lost due to the resampling process.

Line 215: Refer to section numbers as opposed to "explained above"
The section number has been added.

Lines 219 and 220: delete the word "then" This has been changed.

Line 223: "The spatial analyses were done with the PCRaster software" It is not clear what spatial analyses were "done". Consider re-wording.

This has been added (Line 263 - 264):
*Resampling both datasets as well as linking the plots to the MTCI pixels was done with the PCRaster software*

Figure3: I am not sure what the purpose of this figure is since some of the variables being correlated are actually included in the calculation of others e.g. biomass and N concentration are both used to calculate N content – they are bound to be correlated. Hence line 238 is not really a finding.

This figure was included in the manuscript to summarize the information about the forest plots dataset. We also wished to be explicit about the correlation between the variables included in the analysis. Line 238 was not meant to be understood as a new finding but rather a statement about the correlation existing between the canopy N content and the biomass. This figure also shows the skewness of the variables. The original sentence was replaced by (Line 277 - 279):

*The correlation between each pair of variables was significant and the correlation between canopy $N_{[area]}$ and foliar biomass was strongest (r = 0.88). This result was expected as the foliar biomass was included in the $N_{[area]}$ calculation.*

Line 282: I don't understand what this sentence means I'm afraid "This shows that, when the influence of the discrepancy between the original datasets was taken into account, MTCI and canopy N data were linked" what discrepancies were observed?

This sentence referred to the spatial discrepancy between the spatial resolution of the MTCI (1 km) and the forest plots (6m). The sentence has been rephrased (Line 328– 330):

*This showed that, when the influence of the spatial discrepancy between the original datasets, i.e. 6 m and 1 km, was taken into account, MTCI and canopy N data were linked and that the MTCI-canopy N relationship was not strongly affected by the resampled spatial resolution.*

Line 294 "there is no general agreement about MTCI ability for canopy N[%] detection across vegetation and sensor types" Can the authors bring any insights as to why this may be the case? What are the issues?

Thank you. The issue is that across different studies that investigate remote sensing of canopy N at larger scale, i.e. larger than with a handheld spectrometer, the prediction accuracy of the result is highly variable. When using RapidEye at 5m resolution the prediction is similar to what we obtain, while the results obtained by Wang, 2016, even though at very high spatial resolution (3 m) were not significant. As one would maybe expect that the coarse spatial resolution might be a big obstacle to sense canopy N, our results showed that even though the spatial resolution was comparatively low (min 1km) we still get significant results. This thus adds to the discussion about canopy N remote sensing.

The paragraph has been edited to stress these distinctions (Line 333 – 344):
*The overall relationship between MTCI and canopy N[%] at 1 km spatial resolution for all the forest plots (n = 846) was significant and the r2 value was equal to 0.32 (Fig. 5). This result showed that canopy N[%] could be related to MTCI in Mediterranean forests. The performance of the MTCI vegetation index to detect canopy N[%] in Mediterranean vegetation was similar to the results obtained from previous studies using spaceborne MTCI at higher spatial resolution. For example, using MTCI computed from the spaceborne RapidEye sensor at 5 m spatial resolution, it was possible to detect canopy N[%] in grassland savannah and sub-tropical forest with similar coefficients of determination, r2 = 0.35 and r2 = 0.52, respectively (Ramoelo et al., 2012;Cho et al., 2013). However, while there is a consensus regarding MTCI ability for in situ leaf or canopy N[%] detection in a variety of crops using handheld spectrometers (Tian et al., 2011;Li et al., 2014), there is no general agreement about MTCI ability for canopy N[%] detection across vegetation and sensor types at larger scales. For example, MTCI computed from airborne data at 3 m spatial resolution could not be related to canopy N[%] from a mixed temperate forest (Wang et al., 2016). In this context our finding brings new insight into MTCI N[%] sensing capabilities at a much coarser spatial resolution (1 km) compared to what has been done before.*

Line 315-316 Consider re-wording. Also note that there were only 15 plots of Fagus sylvatica! Can you make such a conclusion based on relatively few samples?

Thank you. The sentence has been changed (Line 363 – 364):
*Moreover, when studied separately, the results observed for Fagus sylvatica plots (n = 15) were consistent with the stronger relationship observed for DBF plots.*

We agree that compared to the general size of our dataset, 15 Beech plots is relatively small subset but it provides a first indication. Furthermore, many studies into canopy N detection include very few samples in total. For example, in a mixed temperate forest, Wang et al. (2016) included 26 plots (30 x 30 m) in total. In 2008, Huber et al . studied the remote sensing of canopy in a temperate forest using 28 plots (50 x 50 m) in total. In an arid shrubland, Mitchell et al . (2012) studied 35 plots (7 x 7m). These examples concern remote sensing of canopy N in general, i.e. they do not necessarily include MTCI, nor vegetation indices and use remote sensing sensors with high spatial resolution. Nonetheless, this can still give you an impression that 15 plots is not so uncommon.

Line 348 "Other authors, although agreeing that canopy structural properties needed to be accounted for, suggested that a direct biochemical link between canopy N and reflectance data was not necessary to detect canopy N with reflectance data (Ollinger et al., 2013;Townsend et al., 2013)." What did the authors suggest was necessary?

Thank you. Ollinger et al. (2008) used overall reflectance in the NIR and found a correlation with canopy N in boreal forest. Knyazikhin et al. (2013) argued that this relationship was spurious and resulted solely from differences in canopy structures linked to differences in PFT. Ollinger et al. (2013) and Townsend et al. (2013) argued that the observed relationship was not the result of a direct biochemical mechanism between nitrogen and incoming radiation but rather of an indirect link between

nitrogen and plant structure, which would result from adaptive processes. We have modified the paragraph to add this information to the revised manuscript (Line 396 – 401):

*Other authors, suggested that the role of canopy structure as confounding factor can be explained by an indirect association between canopy N and canopy structure resulting from convergent adaptive processes (Ollinger et al., 2013;Townsend et al., 2013).*

Section 4.4 doesn't really come to any conclusions or suggest reasons for the PFT differences and so it is somewhat superfluous as it stands. Better to integrate this in a wider discussion or include some more detailed interpretation of the data.

Thank you for your suggestion. We have integrated the differences induced by the PFT in a wider discussion about the possible confounding factors that might influence the relationship between MTCI and canopy N. These confounding factors include biomass, canopy structure and climatic variables (Line 391 – 411):

*4.4 Possible confounding factors of the MTCI canopy N relationship*

*The relationships between MTCI and both canopy N[%] and canopy N[area] were influenced by the PFT of the plots. The relationship between MTCI and canopy N[%] was stronger for DBF and mixed plots compared to EBF and ENF plots while the opposite was true for the MTCI-canopy N[area] relationship. In the ongoing discussion about the mechanisms underlying the remote detection of canopy N, some authors argued that the difference in structural properties between different PFTs was a confounding factor of the observed relationship between canopy N and remote sensing data, rendering it spurious (Knyazikhin et al., 2013). Other authors, suggested that the role of canopy structure as confounding factor can be explained by an indirect association between canopy N and canopy structure resulting from convergent adaptive processes (Ollinger et al., 2013;Townsend et al., 2013). In this context, our analysis showed that the PFTs of the plots had an influence on the MTCI canopy N relationship in a specific type of ecosystem, namely Mediterranean forests. Other confounding factors associated with N availability that might affect the observed relationship possibly include biomass, biomass allocation, leaf area index (LAI), water availability, soil type, etc. The data from the forest inventory used in this analysis, i.e. the Catalonian National Forest Inventory, were extensively studied, showing that water availability was the most limiting factor in this region. Water availability was positively correlated with both the N[area] and N[%] in leaves, as well as with foliar and total above-ground biomass through MAP (Sardans, 2011, Sardans, 2013). The MAP also influenced the PFT distribution as DBF plots were located in wetter areas than EBF plots, which were found in wetter sites than ENF plots. Regarding the influence of PFT on the foliar biomass, DBF plots had on average 45% less foliar biomass than EBF or ENF plots (Sardans, 2013). This shows that canopy N[%] and canopy N[area] were interrelated to biomass, PFT and MAP.*

Lines 359-362. I do not follow this point here. What treatments were required and what "might reveal laborious" Consider re-wording.

Thank you. This sentence refers to the different treatments applied to images obtained with imaging spectroscopy at high spatial resolution with airborne or spaceborne sensors. These images need to be corrected for the influence of the atmosphere and clouds (atmospheric correction). In addition, depending on the initial sensor swath width as well as the size of the region to investigate, the images may need to be mosaicked into an image covering a larger area than the initial image acquired by the sensor. Depending on the time period for the ground measurements, the remote sensing images may also need to be temporally averaged.
This has been added to the manuscript (Line 418 - 423):

*However, due to the different treatments required as well as the limited swath width associated with the high spatial resolution (from 3 m to 30 m for Hyspex airborne and Hyperion spaceborne sensors, respectively (Wang et al., 2016;Smith et al., 2003), applying imaging spectrometry at a broader scale might reveal laborious. Depending on the sensors as well as on the extent of the study area, this might involve correcting the acquired images for atmospheric influences and cloud cover as well as combining several images into a larger scale image.*

[revised manuscript text omitted]

---

## Referee Report (RR1)

I would like to thank you the authors for the meticulous review and response to the comments on the first manuscript version. In this new version authors have addressed most of the issues raised during the interactive revision process including additional information that helps to understand methods and results, as is the case for the temporal analysis of the canopy N field measurements and the improved statistical processes. However I still see some weak points in the paper:

1. In my opinion authors have not jet provided a proper justification on the usefulness of the statistical analysis using resampled MTCI images to lower spatial resolution. In fact, there is a kind of contradiction in the manuscript between this analysis and the information provided by the authors in the introduction and discussion about the future potential of canopy N estimation form RS using new generation of sensor with improved spatial resolution. I would find the analysis useful if the authors wanted to demonstrate that sensors with lower spatial resolution can be potentially used to obtain global estimations of canopy N, but, as this is not the case, I would find more convenient to undertake an analysis that allows to demonstrate the sensitivity of the statistical relationships found to the field data ( sample size and distribution). This is an important issue raised by the authors in the discussion (section4-5). Ground canopy N observations are necessary to calibrate and validate models at regional-global scales. In this context, an interesting (and I would say feasible) output of this work could be a sensitivity analysis on the model performance according to field data availability.

2. I still miss in the discussion a more "quantitative" consideration on the potential of the results obtained to feed global vegetation models. Authors argue in their response that their study contributes to the ongoing discussion on canopy N estimations on larger areas using RS but this is, in my opinion, a quite diffuse argument. I would expect a more detailed discussion on how much the estimations should be improved to provide useful input to those models (what is the uncertainty in canopy N that can be considered acceptable for the models? And specifically for Mediterranean environments?)

3. In the discussion authors compare their results (in terms of r2) with other works were similar relationships have been found between canopy N and vegetation indices but they do not mention that other studies do not include the temporal dimension. Temporal variability of vegetation due to phenology should not be ignored when estimates are based in secondary relationships as is the case with N vs vegetation indices and, therefore, studies that including or not this temporal dimension are not fully comparable.

I have also some comments addressing technical/formal issues referred to manuscript version 3:

Abstract line 12. Remote sensing and vegetation indices are not excluding terms, I would recommend rephrasing.

Abstract line 19. I would say "original" instead of "initial higher"

Section 2.2.1 Authors mention that "all foliar cohorts in the canopy were included in the leaf sample" but, was the % of new-old leaves in the crown taken into account during the sampling or the data processing? The N content can greatly differ depending on the leaf age so, in certain phonological periods this need to be considered to obtain an accurate estimation of canopy N.

Section 2.2.2. I think authors should mention here Sentinel-3 OLCI sensor as the most direct inheritor of MERIS ENVISAT.

Section 2.3.1 line 207. It is not clear why you need to resample the landcover map to the MTCI images resolution. If I properly understand you just want to identify and mask the field plots that changed from forest to other non forest covers. If so, you would just mask those field plots located in a landcover map pixel classified as those covers excluded from the analysis.

Line 222. In the title of this section and all through the manuscript I recommend to replace "initial higher" by "original 1Km" spatial resolution.

Line 281. Authors mean here statistically significant?

Line 288. P-value of this relationship?

Lines 407-408. Consider rephrasing to avoid repetition (addition…adding..additional)

Figure 1. I would recommend to add a couple of zoom windows showing the MERIS MTCI 1 km grid on areas with high and low density of field sampling points.

---

## Author Response (AR2)

**Structure of the author's response**

Please note that the line numbers indicated in the replies correspond to the line numbers in the track\_changed version of the manuscript

I would like to thank you the authors for the meticulous review and response to the comments on the first manuscript version. In this new version authors have addressed most of the issues raised during the interactive revision process including additional information that helps to understand methods and results, as is the case for the temporal analysis of the canopy N field measurements and the improved statistical processes. However I still see some weak points in the paper:

**Thank you for your kind words**

1. In my opinion authors have not jet provided a proper justification on the usefulness of the statistical analysis using resampled MTCI images to lower spatial resolution. In fact, there is a kind of contradiction in the manuscript between this analysis and the information provided by the authors in the introduction and discussion about the future potential of canopy N estimation form RS using new generation of sensor with improved spatial resolution. I would find the analysis useful if the authors wanted to demonstrate that sensors with lower spatial resolution can be potentially used to obtain global estimations of canopy N, but, as this is not the case, I would find more convenient to undertake an analysis that allows to demonstrate the sensitivity of the statistical relationships found to the field data (sample size and distribution). This is an important issue raised by the authors in the discussion (section4-5). Ground canopy N observations are necessary to calibrate and validate models at regional-global scales. In this context, an interesting (and I would say feasible) output of this work could be a sensitivity analysis on the model performance according to field data availability.

All analyses in the manuscript use MTCI data at the original 1 km resolution. In Sections 2.3.3 and 3.2, however, we included an additional analysis using resampled MTCI data. We would like to explain here why we did this additional analysis where both data sets (MTCI and forest plots) were resampled to the same lower spatial resolutions. The objective of our study is not to prove the usefulness of the resampled MTCI images, but rather to investigate the relationship between the MTCI product (1 km) and canopy N from forest plots at regional scale. We realize that our analysis is based on data at different spatial scale. We therefore resampled (in the additional analysis mentioned above) the datasets to the same support size to be able to study the relationship between MTCI and canopy N independently of the initial difference in support size. When the relationship between our datasets is still present. Given the similarity in the relationship between canopy N values and MTCI values at the original and lower spatial resolutions, the confidence in the relationship is increased.

This was not clearly explained in the manuscript. In the revised manuscript, a paragraph was added in the Material and Methods part, section 2.3.2 (Line 242 – 250):

This was done because of the initial difference in support size between MTCI spatial resolution and the forest plots size (i.e. 1 km and 6 m, respectively). This enabled us to investigate the relationships between MTCI and canopy N data when the spatial discrepancy was accounted for independently of differences in initial support size. The statistical basis of this approach is that we bring both datasets (forest plots and MTCI values) to the same support size or representative area (Bierkens, 2000). By

averaging out forest plot values within this support size, we calculate the mean of the canopy N value at that support size. By resampling the MTCI values to that same support size, the obtained result consist of a mean of the MTCI value at that support size. We then regressed the expected canopy N values (at the new support size) against the expected MTCI values (at the new support size).

We do not see a contradiction between resampling both our datasets to the same support size and our statement that the advent of sensors with high spatial resolution is expected to improve the observed relationship. Given that the sample size of the forest plots (6 m) will unlikely be larger in the future, using higher spatial resolution data is the only way to decrease the initial scale discrepancy between the forests plots size and the original pixel size.

We think that a sensitivity analysis of the global vegetation model performance according to field data density and distribution is important to assess the validity of the model output, however, it is outside of the scope of our study and study objectives, mainly because we do not have the enormous amount of plot data that would be required to do a proper sensitivity analysis. To still take your comment into account, we have now mentioned this aspect regarding model validation in the revised version of the manuscript in the future perspective section 4.5 (Line 459 – 461):

Obtaining reliable ground based canopy N data over larger areas and for diverse and globally distributed vegetation types would also be necessary to calibrate and validate global vegetation models, as the model performance will depend on the ground data availability and distribution.

2. I still miss in the discussion a more "quantitative" consideration on the potential of the results obtained to feed global vegetation models. Authors argue in their response that their study contributes to the ongoing discussion on canopy N estimations on larger areas using RS but this is, in my opinion, a quite diffuse argument. I would expect a more detailed discussion on how much the estimations should be improved to provide useful input to those models (what is the uncertainty in canopy N that can be considered acceptable for the models? And specifically for Mediterranean environments?)

We have included a consideration about how foliar nitrogen prediction in the model LPJ-Guess could benefit from canopy N estimates from remote sensing in section 4.5 (Line 461 – 465): *Remotely sensed canopy N estimates would support calibration of such models. In a recent study the global vegetation model LPJ-Guess was able to simulate the differences in foliar nitrogen between different PFTs but not within one PFT (Fleischer et al., 2015). In this context, improving remotely sensed canopy N estimates for homogeneous vegetation types would be a beneficial development for such models.*

3. In the discussion authors compare their results (in terms of r2) with other works were similar relationships have been found between canopy N and vegetation indices but they do not mention that other studies do not include the temporal dimension. Temporal variability of vegetation due to phenology should not be ignored when estimates are based in secondary relationships as is the case with N vs vegetation indices and, therefore, studies that including or not this temporal dimension are not fully comparable.

We agree that phenology is important and should not be ignored. To address the temporal dimension, we 1) analyzed the influence of the temporal discrepancy between the plot sampling campaign and the period of MTCI acquisition data in the inter-annual variation of canopy N (Fig. A1). The graph shows that this inter-annual variation is not strong. 2) Forest plots are linked to a 10 year average of MTCI values measured during the same month, i.e. plots measured in July are linked to a 10 years average of MTCI values measured at the same location in July. In this case, the influence of phenology is thus present in both the plot data and the remote sensing data. We think that the main influence of using data from a forest inventory that was carried out during the whole growing season over several years is that the range of canopy N values included is larger.

Regarding the studies we reference and compare our results to, the canopy at the forest sites are indeed sampled once or during a short period (Cho et al., 2013;Ramoelo et al., 2012;Wang et al., 2016), the studies carried out in crops (Tian et al., 2011;Li et al., 2014) were sampled during the whole growing season over several years.

A mention was added to the manuscript, section 4.2.1 (Line 358- 360):

In these comparisons, it should be taken into account that most previous studies were based on a short sampling campaign while our study incorporates canopy N data from a forest inventory that was carried out during the entire growing season and therefore includes differences in phenology.

I have also some comments addressing technical/formal issues referred to manuscript version 3:

**Abstract line 12. Remote sensing and vegetation indices are not excluding terms, I would recommend rephrasing.**

This was replaced by: "Remotely sensed vegetation indices"

Abstract line 19. I would say "original" instead of "initial higher" This was replaced at the mentioned occurrence and later in text

Section 2.2.1 Authors mention that "all foliar cohorts in the canopy were included in the leaf sample" but, was the % of new-old leaves in the crown taken into account during the sampling or the data processing? The N content can greatly differ depending on the leaf age so, in certain phonological periods this need to be considered to obtain an accurate estimation of canopy N.

The % of new-old leaves in the crown was not taken into account. During the sampling campaign, all the foliar cohorts were pooled together in the same sample and this percentage was not recorded. As all the foliar cohorts present on the selected sampled branches are included, we expect that the measurements still represent the plot canopy N value with acceptable accuracy.

Section 2.2.2. I think authors should mention here Sentinel-3 OLCI sensor as the most direct inheritor of MERIS ENVISAT.

This was edited in the text, section 2.2.2. (Line 200 – 205):

While the ESA ENVISAT satellite mission producing MERIS data came to an end in 2012, MERIS products and MTCI in particular are still relevant because the new ESA Sentinel-2 and Sentinel-3 satellite missions haves improved band settings compared to those of MERIS. and increased the spatial resolution to 20 m MTCI can be calculated from Sentinel-2 reflectance data with increased spatial resolution to 20 m (Drusch et al., 2012). The Sentinel-3 mission also releases a level 2 chlorophyll product, the OLCI Terrestrial Chlorophyll Index (OTCI), which calculation is directly based on MTCI. OTCI continues the time series already available for MTCI (Dash and Vuolo, 2010;Vuolo et al., 2012).

Section 2.3.1 line 207. It is not clear why you need to resample the landcover map to the MTCI images resolution. If I properly understand you just want to identify and mask the field plots that changed from forest to other non forest covers. If so, you would just mask those field plots located in a landcover map pixel classified as those covers excluded from the analysis.

We decided to resample the Globcover landcover map (with an original resolution of 300 m) to MTCI original spatial resolution of 1 km before using it as a selection criterion for our forest plots to be on the safe side regarding plot selection and leave out plots located on heterogeneous MTCI pixels. For example, if one plot was located in the only 300 m natural vegetation area of the 1 km MTCI pixel, e.g. a small forest patch surrounded by agricultural crops, the MTCI pixel value would also be influenced by the non-forested area surrounding the plot. Resampling the Globcover landcover map was carried out using the majority option. That way, by using the resampled landcover map (1 km) as the selection criterion, we make sure that the plot located in the isolated vegetation patch is excluded. Also, the number of plots selected for both the analysis with and without resampling of MTCI pixels, i.e. the analysis at the original spatial resolution of 1 km and the analysis at the resampled spatial resolution, is equal. The difference in the results cannot be attributed to the difference in the plots selected.

However, when the analysis is conducted using the original landcover map instead of the resampled landcover map, the number of plot selected is almost the same (n = 866 instead of n = 846) and the relationships observed between the variables are almost not affected and we thus propose to leave out these results as they provide negligible additional information that could be relevant for the interpretation of our results.

Line 222. In the title of this section and all through the manuscript I recommend to replace "initial higher" by "original 1Km" spatial resolution. This has been done.

Line 281. Authors mean here statistically significant? Yes, this has been replaced

**Line 288. P-value of this relationship?**

The p-value was added: *Quercus ilex (r2 = 0.10, p-value

Figure 1. Map showing the forest plots (n = 846) location in the region of Catalonia, north eastern Spain. Two zoom windows are included showing the density of the plots, one with high density and one with low density, relatively to the MTCI 1 km pixel grid. DBF = Deciduous Broadleaf Forest, EBF = Evergreen Broadleaf Forest, ENF = Evergreen Needleleaf Forest, mixed = mixed forest.

**References**

Cho, M. A., Ramoelo, A., Debba, P., Mutanga, O., Mathieu, R., van Deventer, H., and Ndlovu, N.: Assessing the effects of subtropical forest fragmentation on leaf nitrogen distribution using remote sensing data, Landscape Ecology, 28, 1479-1491, doi:10.1007/s10980-013-9908-7, 2013.

Li, F., Miao, Y., Feng, G., Yuan, F., Yue, S., Gao, X., Liu, Y., Liu, B., Ustin, S. L., and Chen, X.: Improving estimation of summer maize nitrogen status with red edge-based spectral vegetation indices, Field Crops Research, 157, 111-123, doi:10.1016/j.fcr.2013.12.018, 2014.

Ramoelo, A., Skidmore, A. K., Cho, M. A., Schlerf, M., Mathieu, R., and Heitkönig, I. M. A.: Regional estimation of savanna grass nitrogen using the red-edge band of the spaceborne rapideye sensor, International Journal of Applied Earth Observation and Geoinformation, 19, 151-162, doi:10.1016/j.jag.2012.05.009, 2012.

Tian, Y. C., Yao, X., Yang, J., Cao, W. X., Hannaway, D. B., and Zhu, Y.: Assessing newly developed and published vegetation indices for estimating rice leaf nitrogen concentration with ground- and space-based hyperspectral reflectance, Field Crops Research, 120, 299-310, doi:10.1016/j.fcr.2010.11.002, 2011.

Wang, Z., Wang, T., Darvishzadeh, R., Skidmore, A. K., Jones, S., Suarez, L., Woodgate, W., Heiden, U., Heurich, M., and Hearne, J.: Vegetation indices for mapping canopy foliar nitrogen in a mixed temperate forest, Remote Sensing, 8, doi:10.3390/rs8060491, 2016.

**Associate Editor Decision: Publish subject to minor revisions (review by editor) (01 Mar 2018) by Sönke Zaehle**

Comments to the Author:

Dear authors,

my apologies for the delay in coming to a decision. I failed to find a second reviewer, therefore I reviewed the manuscript myself. I believe that the manuscript can be publishable in Biogeosciences, if you decided to further revise the manuscript according to the suggestions and comments by reviewer #1 and my comments below. Best wishes,

Sönke

While I find that the manuscript has improved, I still see need for further improvements Major comments:

My main worry with this manuscript is still that the authors state that they show their relationships to be robust against spatial upscaling, while they leave out the critical scale jump from the plot level to the 1km resolution level. I don't think that this invalidates the results of the study per se, but I do believe that the design of this study is unsuitable to make claims about the scalability of the results, because for this one would need to address the scaling from the proximity of the forest plots to the 1km as well.

All analyses in the manuscript use MTCI data at the original 1 km resolution. In Sections 2.3.3 and 3.2, however, we included an additional analysis using resampled MTCI data. We would like to explain here why did this additional analysis where both data sets (MTCI and plot samples) were resampled to the same lower spatial resolutions. The objective of our analysis was not to make claims about the scalability of the results. Our main objective was to study the relationship between the MTCI time series (1 km) and canopy N data from forest plots (6 m). As our two datasets present a difference in scale, in the additional analysis mentioned above, we resampled the two datasets to the same support size to be able to study the relationship between MTCI and canopy N independently of the initial difference in support size. The results show that the correlation between MTCI and canopy N after resampling is not strongly influenced by the resampling. Given the similarity in the relationship between canopy N values and MTCI values at the original and lower spatial resolutions, the confidence in the relationship is increased.

This was not clearly explained in the manuscript. In the revised manuscript, a paragraph was added in the Material and Methods part to make it more clear, section 2.3.2 (Line 242 – 250):

This was done because of the initial difference in support size between MTCI spatial resolution and the forest plots size (i.e. 1 km and 6 m, respectively). This enabled us to investigate the relationships between MTCI and canopy N data when the spatial discrepancy was accounted for. The statistical basis of this approach is that we bring both datasets (forest plots and MTCI values) to the same support size or representative area (Bierkens et al., 2000). By averaging out forest plot values within this support size, we calculate the mean of the canopy N value at that support size. By resampling the MTCI values to that same support size, the obtained result consist of a mean of the MTCI value at that support size. We then regressed the expected canopy N values (at the new support size) against the expected MTCI values (at the new support size).

We have also downscaled the claims about the scalability of the results. In the result part, section 3.2, we have replaced (Line 298 – 299):

This was done to investigate the relationship between MTCI and canopy N data independently of difference in support size

By:

This was done to investigate the relationship between MTCI and canopy N data when the initial spatial discrepancy between the two datasets was accounted for.

In the discussion part section 4.1, we have removed part of the text where the claims were too strong (Line 342 – 344):

This showed that, when the spatial discrepancy between the original datasets, i.e. 6 m and 1 km, was taken into account, MTCI and canopy N data were linked and that the MTCI-canopy N relationship was not strongly affected by the resampled spatial resolution.

The use of the word detection is inappropriate. The paper demonstrates a sometimes significant log-linear correlation between these two variables, but does not attempt to disentangle the possible signal from canopy nitrogen from confounding factors, it does therefore not allow for a detection of canopy N trends.

The word "detection" was replaced when it appeared in text when describing the results we obtained (but not when describing results obtained by others, e.g. in the Introduction part).

The results of the manuscript are presented in a misleading fashion, because it states that (L221) that MTCI and canopy values are related via linear regression, whereas infact the authors use a log-linear regression. This needs to be made clear at every instance (for ease of writing possible by introducing two new symbols referring to the log-transformed canopy values), since this affects the interpretation of the regression (i.e. the connection is not linear as written in the text) as well of the r2 value. I would also expect a reasoning as to why the authors believe that the use of a log-linear relationship between MTCI and canopy N is to be expected.

The mentions to linear regression have been converted to "log-linear regression" when it appeared in text.

We use a log normal relationship because the canopy N concentration show outliers at higher values, as shown by its distribution (Figure 3 in the manuscript). Several phenomena in nature show outliers towards higher values, compared to lower values, in particular when bounded to values > 0 (Limpert et al., 2001).

---

## Author Response (AR3)

Comments to the Author:

Dear auhors,

many thanks for the revisions. I think this is largely fine now, but I would like to ask you to include a short version of your response to the below comment into the manuscript. You can also revert your response to my comment on L154 - that seems to have been an oversight on my part.

Best wishes,

Sönke

Section 2.3.1 line 207. It is not clear why you need to resample the landcover map to the MTCI images resolution. If I properly understand you just want to identify and mask the field plots that changed from forest to other non forest covers. If so, you would just mask those field plots located in a landcover map pixel classified as those covers excluded from the analysis.

We decided to resample the Globcover landcover map (with an original resolution of 300 m) to MTCI original spatial resolution of 1 km before using it as a selection criterion for our forest plots to be on the safe side regarding plot selection and leave out plots located on heterogeneous MTCI pixels. For example, if one plot was located in the only 300 m natural vegetation area of the 1 km MTCI pixel, e.g. a small forest patch surrounded by agricultural crops, the MTCI pixel value would also be influenced by the non-forested area surrounding the plot. Resampling the Globcover landcover map was carried out using the majority option. That way, by using the resampled landcover map (1 km) as the selection criterion, we make sure that the plot located in the isolated vegetation patch is excluded. Also, the number of plots selected for both the analysis with and without resampling of MTCI pixels, i.e. the analysis at the original spatial resolution of 1 km and the analysis at the resampled spatial resolution, is equal. The difference in the results cannot be attributed to the difference in the plots selected.

However, when the analysis is conducted using the original landcover map instead of the resampled landcover map, the number of plot selected is almost the same (n = 866 instead of n = 846) and the relationships observed between the variables are almost not affected and we thus propose to leave out these results as they provide negligible additional information that could be relevant for the interpretation of our results.

Dear editor,

Thank you for you answer.

Following your recommendation, the following comment was added to the manuscript (Line 261 – 264):

[revised manuscript text omitted]